

# Processes influencing near-surface heat transfer in Greenland's ablation zone

Benjamin H. Hills[1,2], Joel T. Harper[2], Toby W. Meierbachtol[2], Jesse V. Johnson[3], Neil F. Humphrey[4], Patrick J. Wright[5,2]

[1]Department of Earth and Space Sciences, University of Washington, Seattle, Washington, USA
[2]Department of Geosciences, University of Montana, Missoula, Montana, USA
[3]Department of Computer Science, University of Montana, Missoula, Montana, USA
[4]Department of Geology and Geophysics, University of Wyoming, Laramie, Wyoming, USA
[5]Inversion Labs LLC, Wilson, Wyoming, USA

*Correspondence to*: Benjamin H. Hills (bhills@uw.edu)

**Abstract.** To assess the influence of various mechanisms of heat transfer on the near-surface ice of Greenland's ablation zone, we incorporate highly resolved measurements of ice temperature into thermal modeling experiments. Seven separate temperature strings were installed at three different field sites, each with between 17 and 32 sensors and extending up to 20 m below the surface. In one string, temperatures were measured every 30 minutes, and the record is continuous for more than three years. We use these measured ice temperatures to constrain modeling analyses focused on four isolated processes to assess the relative importance of each to the near-surface ice temperature: 1) the moving boundary of an ablating surface, 2) thermal insulation by snow, 3) radiative energy input, and 4) temperature gradients below the seasonally active near-surface layer. In addition to these four processes, transient heating events were observed in two of the temperature strings. Despite no observations of meltwater pathways to the subsurface, these heating events are likely the refreezing of liquid water below 5-10 m of cold ice. Together with subsurface refreezing, the five heat transfer mechanisms presented here account for measured differences of up to 3°C between the ice temperature at the depth where annual temperature variability is dissipated and the mean annual air temperature. Thus, in Greenland's ablation zone, the mean annual air temperature cannot be used to predict the near-surface ice temperature, as is commonly assumed.



## 1 Introduction

Ice sheets are coupled to the atmosphere at their upper surface through an exchange of mass and energy. Understanding this coupling is important for knowing the ice sheet's surface mass balance and its associated contribution to sea level rise. In particular, the Greenland Ice Sheet (GrIS) has shown a change toward a more

negative surface mass balance, which constitutes at least half of its contribution to recent sea level rise (van den Broeke et al., 2009; Enderlin et al., 2014). In high melt regions, the surface ice temperature is important to ablation processes such as water storage/runoff and albedo modifications associated with the surface cryoconite layer (Wharton et al., 1985). The ice surface temperature also acts as an essential boundary condition for the transfer of heat into deeper ice below. In order to constrain the rate of ice melting, and more generally to understand the

mechanisms which move energy between the ice and climate systems, we must understand the processes that control near-surface heat transfer in bare ice.

Heat transfer at the ice surface is dominated by thermal diffusion from the overlying air. Seasonal air temperature oscillations are diminished with depth into the ice, until they are negligible (i.e. ~1%) at a 'depth of zero annual amplitude' (van Everdingen, 1998). The exact location of this depth is dependent on the thermal diffusivity of the

material through which heat is conducted as well as the period of oscillations (Carslaw and Jaeger, 1959; pp. 64-70). In theory, the temperature at the depth of zero annual amplitude, a value we will call $T_0$, is constant and equal to the mean annual air temperature. In snow and ice, the depth of zero annual amplitude is approximately 10 and 15 m respectively (Hooke, 1976). For this reason, studies in the cryosphere often use $T_0$ as a proxy for the mean air temperature, drilling to 10 or more meters to measure the snow or ice temperature at that depth (e.g. Loewe, 1970).

In places where heat transfer is purely diffusive, the snow or ice is homogeneous, and the climate forcing is constant, $T_0$ is a good approximation for the mean air temperature. However, prior studies have shown that, in many areas of glaciers and ice sheets, the relationship between air and ice temperatures can be substantially altered by additional heat transfer processes. For example, in the percolation zone, infiltration and refreezing of surface meltwater warm the subsurface (Humphrey et al., 2012; Müller, 1976). Studies have also revealed ice anomalously

warmed by 5°C or more in the ablation zone (Hooke et al., 1983; Meierbachtol et al., 2015), but the mechanisms for this are unclear.

Hooke et al. (1983) explored the impacts of several heat transfer processes within near-surface ice at Storglaciären and the Barnes Ice Cap. They focused on the wintertime snowpack which acts as insulation to cold air temperatures, but is permeable to meltwater percolation. Their results showed that the average ice temperature at and below the

equilibrium line of those glaciers tends to be higher than the mean annual air temperature, and they attributed this mainly to snow insulation because the strength of their measured offset was correlated to the thickness of the snowpack.

In this study, we expand the analysis of Hooke et al. (1983) and turn focus to the GrIS ablation zone with near-surface temperature profiles from seven locations. We use those temperature measurements in conjunction with a

one-dimensional heat transfer model to assess heat transfer processes in this area. The processes which make the



ablation zone different from other areas of the GrIS are, first, that the ice surface spends much of the summer period pinned at the melting point, despite slightly higher air temperatures. Next, high ablation rates counter emerging ice flow, removing the ice surface and exposing deeper ice, along with its heat content, to the surface. Further, winter snow insulates the ice from cold winter temperatures, and the absorption of solar radiation results in heating and

melting of ice. Finally, surface melt can move through open fractures, carrying latent heat with it to deeper and colder ice, and upon refreezing, the meltwater warms that ice below the surface.

Each of these processes can be simulated numerically. However, a quantitative comparison of the competing processes can only be derived through observation. Confidently constraining the role of each of these processes requires in situ measurements with both high time and space resolution, and records that span hours to seasons.

**2 Field Site and Instrumentation**

Field observations used in this study are from three sites in western Greenland (Figure 1). Each site is named by its location with respect to the terminus of Isunnguata Sermia, a land-terminating outlet glacier. The furthest inland site, 46-km, is ~30 km below the equilibrium line altitude which is at about 1500 m elevation (van de Wal et al., 2012), so all sites are well within the ablation zone and ablation rates are high (2-3 m/yr). Solar radiation in the summer

creates a layer of interconnected cryoconite holes at the ice surface, and water moving through that cryoconite layer converges into surface streams. There are no large supraglacial lakes in the immediate area of any site; all streams eventually drain from the surface through moulins. A series of dark folded layers emerge to the ice surface in this region of the ice sheet (Wientjes & Oerlemans, 2010).

At each field site, boreholes for temperature instrumentation were drilled from the surface to 20 m depth using hot-

water methods. In total, seven strings of temperature sensors were installed – one at both sites 27-km and 46-km in 2011, followed by five at site 33-km between 2014 and 2016. Strings are named by the year they were installed. Each consists of between 17 and 32 sensors spaced at 0.5-3.0 m along the cable (Table 1). In 2011 and 2014, thermistors were used as temperature sensors. The thermistors have measurement resolution of 0.02°C and accuracy of about 0.5°C after accounting for drift (Humphrey et al., 2012). In subsequent years, we used a digital temperature

sensor (model DS18B20 from Maxim Integrated Products, Inc.). This sensor has resolution 0.0625°C and about the same accuracy as the thermistors. To increase accuracy, each sensor was lab-calibrated in a 0°C bath, and field-calibrated during borehole freeze-in.

Meteorological variables were measured at each field site as well, using standard Campbell Scientific products. In this study, we use the near-surface air temperature (Vaisala HMP60), the net radiative heat flux over all wavelengths

shorter than 100 μm (Kipp and Zonen NR Lite), and the change in surface elevation measured with a sonic ranger (Campbell SR50A). Two variables are taken from the sonic ranger, cumulative ablation during the melt season and changes in snow depth during the winter. The meteorological station with all the above instrumentation was mounted on a fixed pole frozen in the ice. The measurement frequency for meteorological data varies from ten minutes to an hour, but all data are collapsed to a daily mean for input to a heat transfer model.



In addition to ice temperature and meteorological measurements, investigations of the subsurface were completed at site 33-km with a borehole video camera, and with a high-frequency ground-penetrating radar survey (supplementary). These investigations were carried out in pursuit of what we think may have been subsurface fractures that are not expressed at the ice surface (described in section 5.2). With five temperature sensor strings, a meteorological station, and the subsurface investigations, site 33-km is by far the most heavily instrumented of the three sites. For that reason, measurements from this site serve as the foundation for a model case study presented in section 4.

## 3 Results

### 3.1 Observed Ice Temperature

Near-surface ice temperatures were measured through time in seven shallow boreholes at three different field sites (Figure 2). Although hot-water drilling methods temporarily warm ice near the instrumentation, the ice around these shallow boreholes cools to its original temperature within days to weeks. The measured temperatures are spatially variable between sites, with a $T_0$ of -3.2 at 27-km, -8.6 at 46-km, and from -9.7°C to -8.1 at 33-km. In all cases, measured $T_0$ values are warmer than the mean annual air temperature. Temperature gradients at 20 m are also variable, typically being between -0.1 and 0.0°C/m but +0.1°C/m at the 27-km field site. As expected, variability in the temperature gradients at the bottom of the profiles measured here correlate well with the observations of deeper temperature profiles measured at each site (Harrington et al., 2015; Hills et al., 2017).

Even the five temperature profiles measured at site 33-km exhibit some amount of spatial variability. Three temperature strings, T-15a, b, and c, are all similar, having strong negative temperature gradients (ranging from -0.1 to -0.05°C/m), and cold $T_0$ temperatures (-9.5°C). Close to the surface, these three temperature strings appear to be rather cold compared to the others. However, those strings failed in May 2017 and did not yield a full year of data. The missing summer period explains the mean cold bias near the surface. T-16 is the only string that did not reach 20 m. This short string extended to 9.5 m depth, and measured the smallest range in temperatures throughout a season with the coldest surface temperatures not even reaching -15°C. In terms of mean temperature, T-16 is similar to T-14, having a small negative temperature gradient and warm temperatures in comparison to those of T-15. Based on these observations, spatial variability in near-surface ice temperature at site 33-km is controlled on the scale of hundreds of meters. Proximal observations from the three nearby T-15 strings are similar to one another, but greater variability is observed when including the more distant strings, T-14 and T-16.

Closer inspection of the measured temperature record through time reveals the transient nature of near-surface ice temperature (Figure 3). As expected, these data show a strong seasonal oscillation near the surface. During the melt season, the ice surface quickly drops after near-surface ice is warmed to the melting point. Just below the surface, the winter cold wave persists for several weeks into the summer season. For this particular string, T-14, transient heating events were observed during the melt seasons. Similar events were observed in the T-16 string (Figure 4), but not in any other. The events range in magnitude, but in one instance ice is warmed from -10°C to -2°C in 2



hours. We can only speculate on the origins of these events, and address this below in section 5.2.

### 3.2 Meteorological Data

Meteorological data from site 33-km were observed over three years (Figure 5). Air temperatures are normally at or above the melting temperature during the summer, but fall to below -30°C in winter months. The measured ablation
rate is on the order of 2-3 m/yr and maximum snow accumulation is only up to about 0.5 m. Net shortwave radiation is less than zero in the winter (net outgoing because of thermal emission in the infrared wavelengths) but over 100 W/m$^2$ (daily mean) on some days in the summer.

The mean air temperature over the entire measurement period at site 33-km (-10.5°C) is cold in comparison to measured ice temperatures at that site (Figure 2; T-14, T-15, and T-16). This warm bias in the near-surface ice
temperature is also observed at sites 27-km and 46-km, where ice is warmer than the measured air temperature and significantly warmer than the reference from a regional climate model (Meierbachtol et al., 2015). Interestingly, we measure almost no winter snowpack at sites 27-km and 46-km due to low precipitation and strong winds. Our observations are thus in contradiction to the inferences made by Hooke et al. (1983), who said that the offset between air and ice temperature is primarily a result of snow insulation.

Overall, the three years for which meteorological data were collected are significantly different. The 2014-15 winter was particularly cold, bringing the mean air temperature of that year more than a degree lower than the other two seasons. Snow accumulation was approximately doubled that winter in comparison to the other two. Also, the summer melt season is longer in 2016 than in 2015. In comparison with past trends from a nearby meteorological station, IMAU s6, the second year is more typical for this area (Fausto et al., 2009). To model a representative
season, data from that second year (July 2015 to July 2016) were chosen as annual input for the model case study.

## 4 Analysis

Our objective is now to investigate how various processes active in Greenland's ablation zone influence $T_0$. In order for model results to achieve fidelity, inputs and parameters need to be representative of actual conditions. We therefore use the observational data above to constrain the modeling experiments. Our modeling is focused at field
site 33-km, where we have the most data for constraining the problem.

### 4.1 Model Formulation

The foundation for quantifying impacts of near-surface heat-transfer processes is a one-dimensional thermodynamic model. Our model uses measured meteorological variables as the surface boundary condition and simulates ice temperature to 20 m, a depth chosen for consistency with measured data. The ice temperature at the depth of zero
annual amplitude, $T_0$, is output from the bottom of the domain for each model experiment and used as a metric to compare net temperature changes between simulations. The model does not, nor is it meant to, simulate the surface mass balance.



We implement an Eulerian framework, treating the $z$ dimension as depth from a moving surface boundary so that emerging ice is moving through the domain and is removed when it melts at $z = 0$. We use a finite element model with a first-order linear element. For a seamless representation of energy across the water/ice phase boundary, we implement an advection-diffusion enthalpy formulation (i.e. Aschwanden et al., 2012; Brinkerhoff and Johnson, 2013),

$$(\partial_t + w\partial_z)H = \partial_z(\alpha\partial_z H) + {\phi}/{\rho_i} \qquad (1)$$

Here, $\partial$ is a partial derivative, $t$ is time, $w$ is the vertical ice velocity with respect to the lowering ice surface, $z$ is depth, $H$ is specific enthalpy, $\alpha$ is the ice diffusivity, $\phi$ is any added energy sources, and $\rho_i$ is the density of ice. The diffusivity term is enthalpy-dependent,

$$\alpha(H) = \begin{cases} {k_i}/{\rho_i C_p} & cold, H < H_m \\ {v}/{\rho_i} & temperate, H > H_m \end{cases} \qquad (2)$$

where $k_i$ is the thermal conductivity of ice which we assume is constant over the small temperature range in this study (~25°C), $C_p$ is the specific heat capacity which is again assumed constant, $v$ is the moisture diffusivity in temperate ice, and $H_m$ is the reference enthalpy at the melting point (all constants are shown in Table 2). Aschwanden et al. (2012) include a thermally diffusive component in temperate ice (i.e. $k_i\partial_z^2 T_m(P)$). However, since we consider only near-surface ice, where pressures ($P$) are low, this term reduces to zero. Using this formulation, energy moves by a sensible heat flux in cold ice and a latent heat flux in temperate ice. We assume that the latent heat flux, prescribed by a temperate ice diffusivity ($v/\rho_i$), is an order of magnitude smaller than the cold ice diffusivity ($k_i/\rho_i C_p$). We argue that this is representative of the near-surface ice when cold ice is impermeable to meltwater.

The desired model output is ice temperature. It has been argued that temperature is related to enthalpy through a continuous function, where the transition between cold and temperate ice is smooth over some 'cold-temperate transition surface' (Lüthi et al., 2002). On the other hand, we argue that cold ice is completely impermeable to water except in open fractures (which we do not include in these simulations), so we use a stepwise transition,

$$T(H) = \begin{cases} {(H - H_m)}/{C_p} + T_m & cold \\ T_m & temperate \end{cases} \qquad (3)$$

Additional enthalpy above the reference increases the water content in ice,

$$\omega(H) = \begin{cases} 0 & cold \\ {(H - H_m)}/{L_f} & temperate \end{cases} \qquad (4)$$

where $L_f$ is the latent heat of fusion. If enough energy is added to ice that its temperature would exceed the melting point, excess energy goes to melting. In our case study, we limit the water content based on field observations of




water accumulation in the layer of rotten ice and cryoconite holes. This rotten ice layer extends to approximately 20 cm deep and as an upper limit accumulates a maximum 50% liquid water. Therefore, we limit the water content,

$$0.0 \leq \omega \leq 0.5 \tag{5}$$

The two boundary conditions are 1) fixed to the air temperature at the surface,

$$T(\text{surface}, t) = T_{air} \tag{6}$$

and 2) free at the bottom of the domain,

$$\frac{\partial T}{\partial z}\bigg|_{bottom} = 0.0 \tag{7}$$

The surface boundary condition is updated at each time step to match the measured air temperature. The bottom boundary condition is fixed in time. This bottom boundary condition is also changed for some model experiments to test the influence of a temperature gradient at the bottom of the domain (section 4.2.4).

## 4.2 Experiments

Four separate model experiments are run, each with a new process incorporated into the model physics, and guided
by observational data. All simulations use the enthalpy formulation above rather than temperature in order to track the internal energy of the ice/water mixtures that are prevalent in the ablation zone. Each experiment is referenced to an initial control run which reflects simple thermal diffusion of the measured air temperature in the absence of any additional heat transfer processes. Meteorological data are input where needed for an associated process in the model. These data are clipped to one full year and input at the surface boundary in an annual cycle. The model is run
with a one-day time step until ice temperature at the bottom of the domain converges to a steady temperature. A description of each of the model experiments follows below. These experiments build on one another, so each new experiment incorporates the physics of all previously discussed processes.

### 4.2.1 Ablation

The first experiment simulates motion of the ablating surface. While the control run is performed with no advective
transport (i.e. $w = 0$), in this experiment we incorporate advection by setting the vertical velocity equal to measurements of the changing surface elevation through time. When ice is melted the ice surface location drops. Because the vertical coordinate, $z$, in the model domain is treated as a distance from this moving surface, ablation brings simulated ice closer to the surface boundary. Hence, the simulated ice velocity is assigned to the ablation rate (except in the opposite direction) for this first model experiment. This ablation rate is calculated as a forward
difference of the measured surface lowering.



### 4.2.2 Snow Insulation

The second experiment incorporates measured snow accumulation, which thermally insulates the ice from the air. The upper boundary condition is assigned to the snow surface, whose location changes in time. Diffusion through the snowpack is then simulated as an extension of the ice domain but with different physical properties. The thermal conductivity of snow (Calonne et al., 2011),

$$k_s = 2.5 * 10^{-6}\rho_s{}^2 - 1.23 * 10^{-4}\rho_s + 0.024 \tag{8}$$

is dependent on snow density, $\rho_s$, for which we use a constant value, 300 kg/m$^3$. We treat the specific heat capacity of snow to be the same as ice (Yen, 1981).

### 4.2.3 Radiative Energy

The third model experiment incorporates an energy source from the net solar radiation measured at the surface. Energy from radiation is absorbed in the ice and is transferred to thermal energy and to ice melting (van den Broeke et al., 2008). We assume that all this radiative energy is absorbed in the uppermost 20 cm, the rotten cryoconite layer, and if snow is present the melt production immediately drains to that cryoconite layer. When the net radiation is negative (wintertime) we assume that it is controlling the air temperature, so it is already accommodated in our simulation; thus, the radiative energy input is ignored in the negative case. This radiative source term, $\phi$, is incorporated into equation (1) at each time step.

While some models treat the absorption of radiation in snow/ice more explicitly with a spectrally-dependent Beer-Lambert Law (e.g. Brandt and Warren, 1993), we argue that it is reasonable to assume that all wavelengths are absorbed near the surface over the length scales that we consider. The only documented value that we know of for an absorption coefficient in the cryoconite layer is 28 m$^{-1}$ (Lliboutry, 1965) which is close to that of snow (Perovich, 2007). If the properties are truly similar to that of snow, about 90 percent of the energy is absorbed in the uppermost 20 cm (Warren, 1982). Moreover, we argue that this is precisely the reason that the cryoconite layer only extends to a limited depth: it is a result of where radiative energy causes melting.

### 4.2.4 20-m Temperature Gradient

Finally, in the fourth model experiment we change the boundary condition at the bottom of the domain. The free boundary is changed to a Neumann boundary with a gradient of -0.05°C/m., a value that approximately matches the measured gradient in our near-surface temperature measurements at field site 33-km. Importantly, this simulated gradient also matches that of the upper ~100 m of ice in our measurements of deep temperature profiles (Hills et al., 2017). In this case, the advective energy flux is upward, but the temperature gradient is negative, bringing colder ice to the surface.





### 4.3 Model Results

The control model run of simple thermal diffusion predicts that ice temperature damps to approximately the mean annual air temperature of the study year (-9.9°C) by about 15 m below the ice surface. This result is in agreement with the analytical solution (Carslaw and Jaeger, 1959), but slightly different from the mean air temperature because the air can exceed the melting temperature in the summer while the ice cannot. For each model treatment, 1-4, the incorporation of an additional physical process changes the ice temperature. Differences between model runs are compared using $T_0$ at 20 m. Again, the model experiments are progressive, so each new experiment includes the processes from all previous experiments. Key results from each experiment are as follows (Figure 6):

1. Diffusion alone results in $T_0 = -9.9°C$, whereas observed temperatures range from -9.7°C to -8.1 at the 33-km field site.
2. Because the ablation rate is strongest in the summer, the effect of incorporating ablation is to counteract the diffusion of warm summer air temperatures. The result is a net cooling of $T_0$ from experiment (1) by -0.92°C.
3. Snow on the ice surface insulates the ice from the air temperature. In the winter, snow insulation keeps the ice warmer than the cold air, but with warm air temperatures in the spring it has the opposite effect. Because snow quickly melts in the springtime, the net effect of snow insulation is substantially more warming than cooling. $T_0$ for this experiment is +0.78°C warmer than the previous.
4. Radiative energy input mainly controls melting (van den Broeke et al., 2008), but incorporating this process does warm $T_0$ by +0.52°C.
5. Imposing a 20-m temperature gradient consistent with observation dramatically changes $T_0$ by -2.5°C

Both ablation and the deep temperature gradient have a cooling effect on near-surface ice temperature. On the other hand, snow and radiative energy input have a warming effect. For this case study, the first three processes together result in almost no net change so that the modeled $T_0$ is close to the observed mean air temperature (Figure 6d). However, inclusion of the deep temperature gradient has a strong cooling effect on the simulated temperatures, bringing $T_0$ far from the mean measured air temperature. In summary, measured ice temperatures are consistently warmer than both the measured air temperature and any simulated ice temperature (Figure 7).

### 5 Discussion

Our observations show that measurements of near-surface ice in the ablation zone of western Greenland can be significantly warmer than would be predicted by diffusive heat exchange with the atmosphere. This is in agreement with past observations collected in other ablation zones (e.g. Hooke et al., 1983). With four experiments in a numerical model that progressively incorporate more physical complexity, we are unable to precisely match independent model output to observations. Our measurement and model output consistently point toward a disconnect between air and ice temperatures in the GrIS ablation zone, with ice temperatures being consistently warmer than the air.





### 5.1 Ablation-Diffusion

The strongest result from our model case study was a drop in $T_0$ by 2.5°C associated with the negative 20-m temperature gradient. While it was important to test this scenario for one case, the temperature gradient we used was somewhat arbitrary. In reality, the 20-m temperature gradient is widely variable from one site to another and even

within one site (Figure 2). Interestingly, full ice thickness temperature profiles show similar temperature gradients, both positive and negative, that persist for many hundreds of meters toward the bed (Harrington et al., 2015; Hills et al., 2017).

Our model could have tested additional temperature gradients, and those of the opposite (positive) sign likely would have fit our measured $T_0$ much better. However, we argue that this would have been a simple model tuning exercise

to match data, whereas our purpose is to elucidate the relationship between near-surface ice temperature and ablation/emergence. Furthermore, the ice temperature gradient should theoretically be negative in the ablation zone. Consider that fast horizontal velocities (~100 m/yr) advect cold ice from the divide to the ablation zone, and the air temperature lapse rate couples with the relatively steep surface gradients so that the surface warms rapidly toward the terminus. These conditions lead to a vertical temperature gradient below the ice surface that is negative (Hooke,

2005; pp. 131-135), as in our model example. The one exception is in the case of deep latent heating in a crevasse field (Harrington et al., 2015; sites S3 and S4) where the deep ice temperature would be warmer than the mean air temperature rather than colder.

Importantly, our results demonstrate the effect of the 20-m temperature gradient is coupled to that of surface lowering. With respect to the surface, the temperature gradient below is advected upward as ice melts. There is a

competition between surface lowering and diffusion of atmospheric energy into the ice; as near-surface ice is warmed, it can be removed quickly and a new boundary set. Therefore, our conceptualization of temperature in the near-surface ice of the ablation zone should not be a seasonally oscillating upper boundary with purely diffusive heat transfer (Carslaw and Jaeger, 1959), but one with advection and diffusion (Logan & Zlotnik, 1995; Paterson, 1972). This conceptualization is unique to the ablation zone because of its high ablation rates, which are at least an order of

magnitude larger than other areas.

The implications for the disconnect between air and ice temperature are that the near-surface active layer in the ablation zone is small (i.e. less than 15 m) and could be skewed toward the 20-m temperature gradient. Therefore, the surface boundary condition has a much weaker control on diffusion in ice well below the surface than it would in other areas of the ice sheet. Additionally, melting dynamics are complicated by the 20-m temperature gradient.

Under these conditions, it is no surprise that we see spatial variability in near-surface ice temperature even within one field site. That variability is simply an expression of the deeper ice temperature variations which are hypothesized to exist from variations in vertical advection (Hills et al., 2017), and do not have time to completely diffuse away before they are exposed at the surface.



### 5.2 Subsurface Refreezing

In two temperature strings we observe transient heating, the largest case being as much as 8°C in 2 hours between ~3-8 m below the ice surface. We argue that the most likely energy source for such events is latent heat because the events are transient, they are spatially discrete, and they are generated from depth. This implies that the heating observations are due to refreezing of liquid water in cold ice. Similar refreezing events have been observed in firn (Humphrey et al., 2012), where they are not only important for ice temperature but could also imply a large storage reservoir for surface meltwater (Harper et al., 2012). However, unlike firn, solid ice is impermeable unless fractures are present.

While water-filled cavities have been observed in cold ice (e.g. Paterson and Savage, 1970) and Fountain et al. (2005) suggest that fractures provide the main pathway for liquid water to move through temperate ice, a mechanism for sudden water movement to depth in the ablation zone of Greenland is not obvious. Fractures are the most likely pathway, but to move water through cold ice they would need to be large enough that water moves quickly and does not instantaneously refreeze. A 1-mm wide crack in ice that is -10°C freezes shut in about 45 seconds (Alley et al., 2005; eq. 8). Assuming that there is a hydraulic potential gradient to drive water flow, that amount of time is long enough for small volumes of water to move at least 5-10 m through cold ice.

The field site at 33-km has no visible open crevasses at the surface, but does have occasional mm aperture cracks. Nevertheless, independent field observations in this area including hole drainage during hot-water drilling, ground-penetrating radar, and borehole video observations, all pointing to the existence of subsurface air-filled and open fractures with apertures of up to a few cm (supplementary). We suggest that these features occasionally move water to ~10 m below the ice surface, where it refreezes and warms the ice as we have observed. That they are open at depth, but are narrow or non-existent at the surface, could be linked to the colder ice at depth and its stiffer rheology. On rare occasions, we argue that the aperture of the fractures open wider to the surface, where there is copious water stored in the cryoconite layer. While the events seem to happen in the springtime and it would be tempting to assert that fracture opening coincides with speedup, our measurements of surface velocity at these sites show that this is not always the case.

Latent heating is an obvious candidate for the source of 'extra' heat that we observe in our temperature strings relative to simulations. Our data show that refreezing in subsurface fractures has the potential to warm ice substantially over short periods of time, and apparently this can occur in places where crevasses are not readily observed at the surface. However, without a more thorough investigation, we have no evidence to show that these refreezing events are more than local anomalies. Of our seven near-surface temperature strings, only T-14 and T-16 demonstrated refreezing events, and so we are not confident that they are temporally or spatially ubiquitous. Other possibilities for the warm bias would either a strong warm event in previous years such as in 2010 (van As et al., 2012), or possibly deeper latent heating from an upstream crevasse field. In those cases, a positive 20-m temperature gradient would promote warming near the surface; however, in this area full-depth temperature profiles do not show deeper ice to be anomalously warmed except in one localized case (Hills et al., 2017).





## 6 Conclusion

We observe the temperature of ice at the depth of zero annual amplitude, $T_0$, in Greenland's ablation zone to be measurably different than the mean annual air temperature. These findings contradict predictions from purely diffusive heat transport, but are not surprising considering the processes which impact heat transfer in the ablation

zone. High ablation rates in this area indicate that ice temperatures below ~15 m reflect the temperature of deep ice that is emerging to the surface, confirming that the ice does not have time to equilibrate with the atmosphere. In other words, ice flow brings cold ice to the surface at a faster rate than heat from the atmosphere can diffuse into the ablating surface. The coupling between rapid ablation and the spatial variability in deep ice temperature implies there will always be a disconnect between air and ice temperatures. Additionally, we observe infrequent refreezing

events below 5-10 m of cold ice. Meltwater is likely moving to that depth through subsurface fractures that are not obviously visible at the surface.

In analyzing a series of processes that control near-surface ice temperature, we find that some of them lead to colder ice, and others to warmer, but most are strong enough to dramatically alter the ice temperature from the purely diffusive case. With rapid ablation, a spatially variable temperature field, and subsurface refreezing events, $T_0$ in the

ablation zone should not be expected to match the air temperature. That our measurements are consistently warmer, could simply be due to the limited number of observations we have, but latent heat additions are clearly measured and could be common in near-surface ice of the western Greenland ablation zone.

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



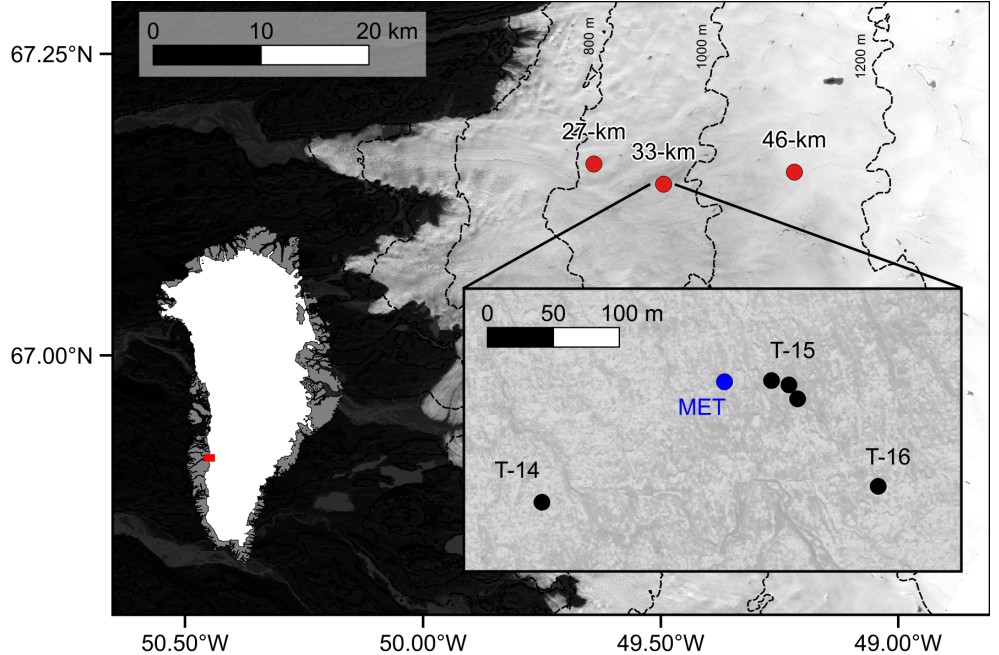

**Figure 1: A site map from southwest Greenland with field sites (red) named by their location with respect to the outlet terminus of Isunnguata Sermia. The inset shows locations of near-surface temperature strings (black) named by the year they were installed and a meteorological station (blue). Surface elevation contours are shown at 200-m spacing (Howat et al., 2014).**




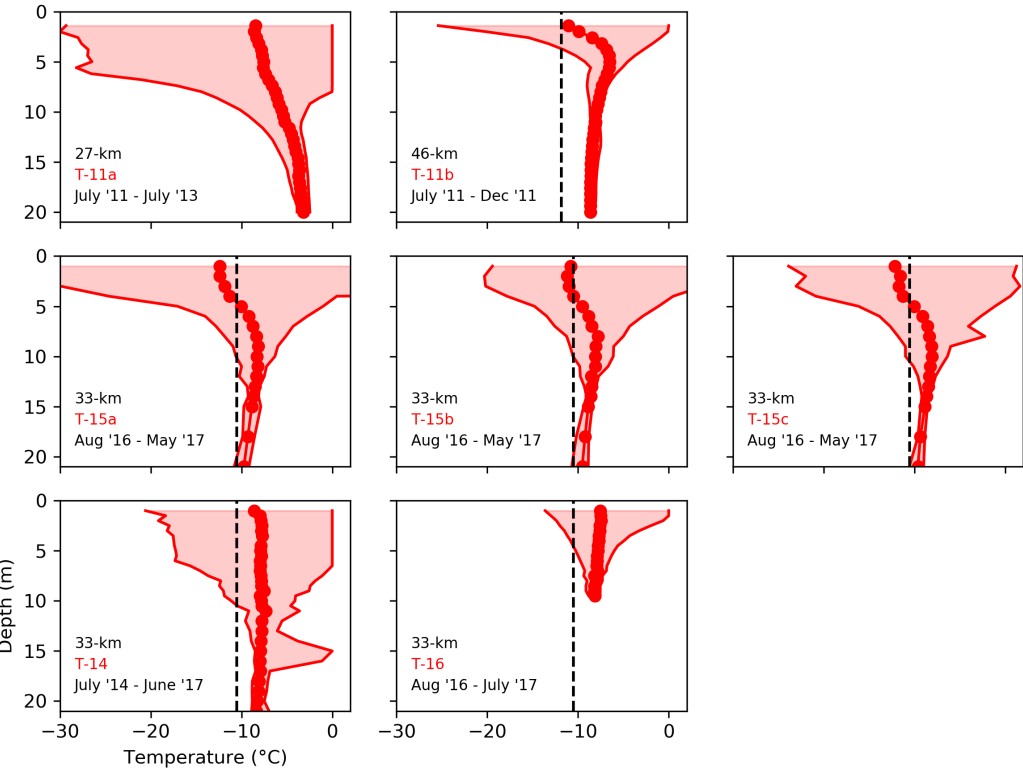

**Figure 2: Near-surface ice temperature measurements from seven strings: T-11a, T-11b, T-15a, T-15b, T-15c, T-14, and T-16. For each, the shaded region shows the range of measured temperatures over the entire measurement period, and the red dots indicate a mean value for each sensor. For field sites at which the air temperature was measured for at least a full year, a dashed line shows the mean air temperature.**





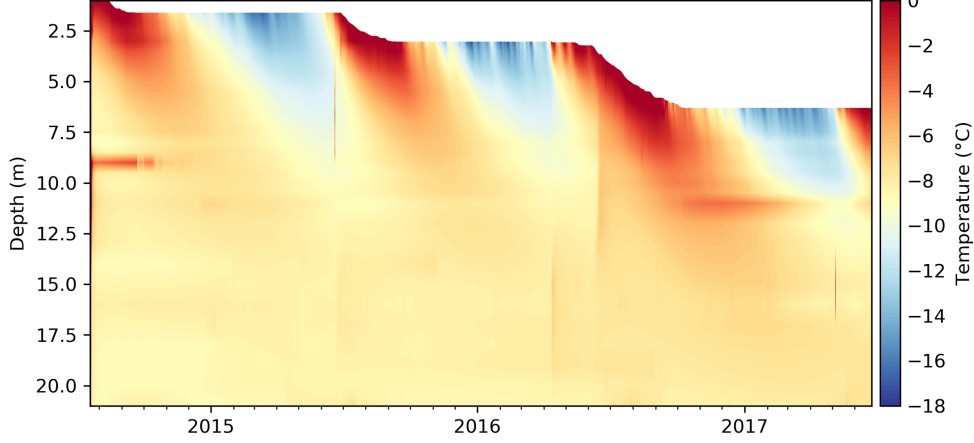

**Figure 3: Three years of ice temperature measurements from the T-14 string. While this string was initially installed to 20 m depth, the sensors closest to the surface melt out as the surface drops. Here, ablation measurements (corresponding to Figure 5b) are plotted as a white mask so that measurements from sensors laying at the surface are hidden.**




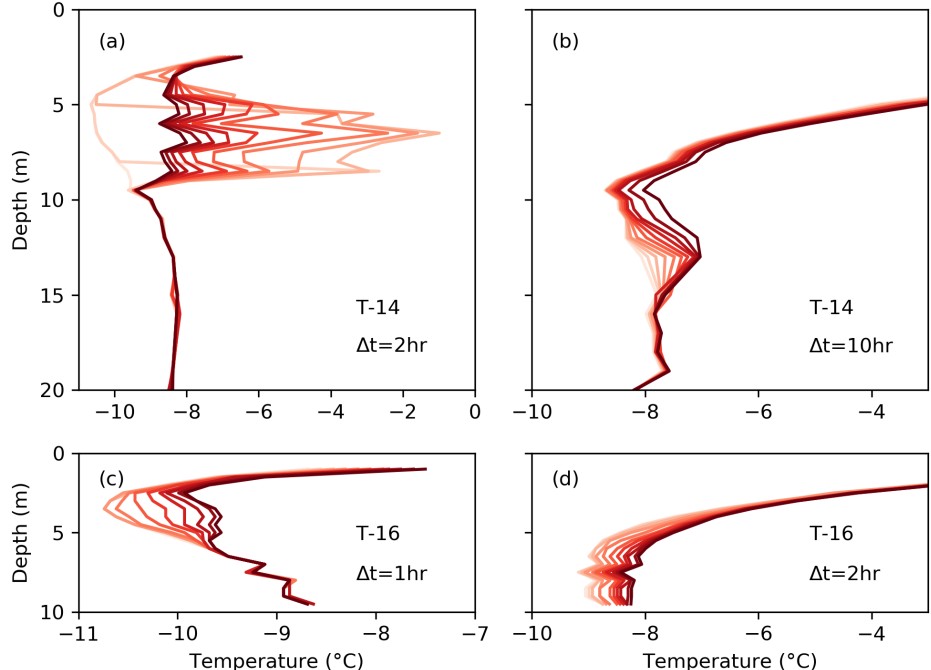

Figure 4: Heating events within the ice temperature record from two separate strings at 33-km. Profiles are displayed as a series through time with lighter being earlier and darker being later. The time step between profiles is a) 2 hours, b) 10 hours, c) 1 hour, and d) 2 hours.



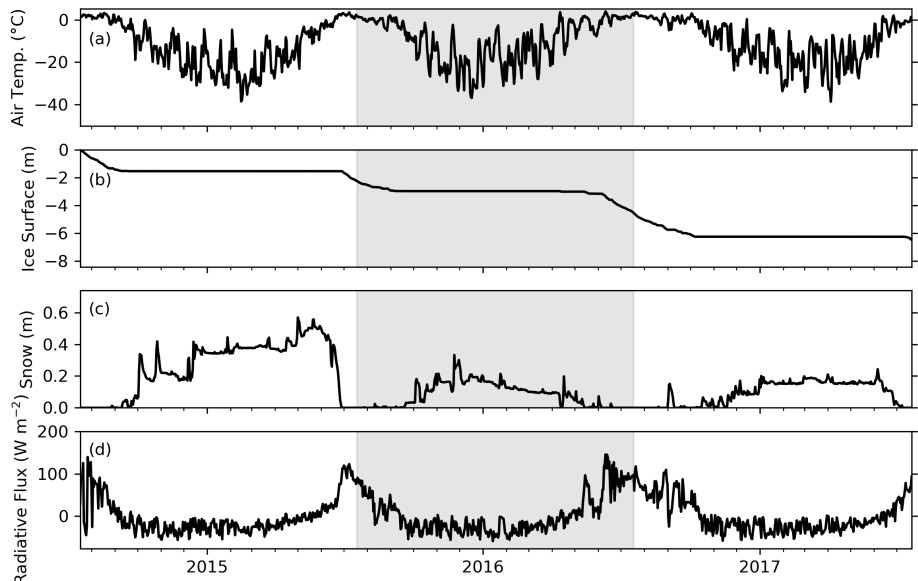

**Figure 5: Meteorological data from 33-km over three years including a) air temperature, b) ice surface location, c) snow depth, and d) net shortwave radiation. All data are plotted as a daily mean. The shaded region encloses the time period that is used for the model case study.**





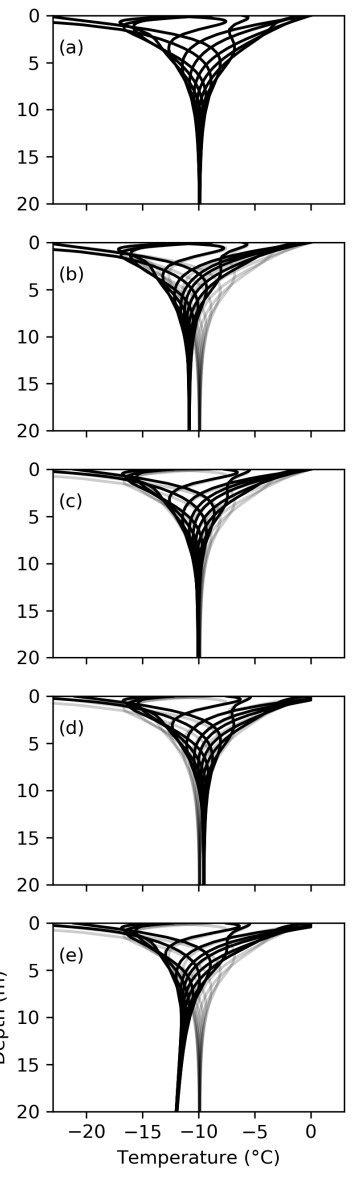

**Figure 6: Model results for six separate simulations. In each case, twelve simulated temperature profiles are shown from throughout the yearlong period, and transparent control results are displayed for comparison. Processes are from top to bottom: a) control model run of pure diffusion, b) ablation, c) snow insulation, d) radiative energy input, and finally e) 20-m temperature gradient.**



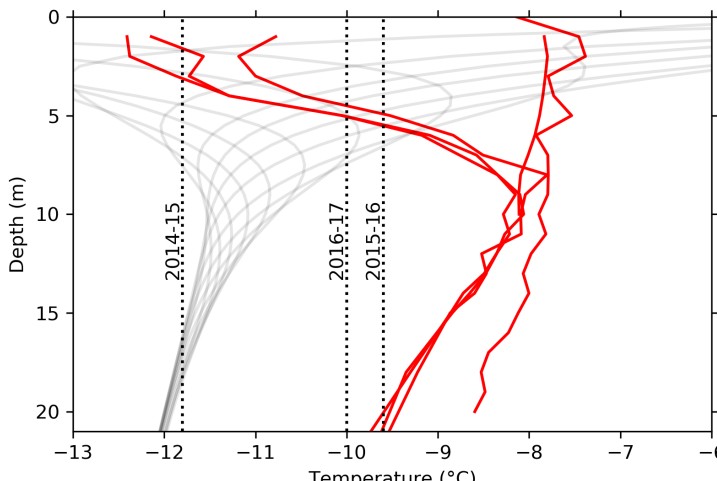

**Figure 7: A comparison of model output (transparent) and data from 33-km, including mean ice temperatures (red) and mean annual air temperatures for three seasons (black dashed). The measured temperatures are plotted differently from Figure 2. Instead of fixed sensor locations, the depth here is plotted at a distance relative to a melting surface (the same way as the model results). Note that three of the temperature strings failed before running for an entire year. Those three strings are biased cold near the surface.**



**Table 1: Temperature Strings**

| String Name | Data Time Period | Time Step (hr) | Sensor | # of Sensors | Sensor Spacing (m) | Latitude | Longitude |
|---|---|---|---|---|---|---|---|
| T-11a | 7/5/11 – 7/15/13 | 3 | Thermistor | 32 | 0.6 | 67.19518 | -49.71952 |
| T-11b | 7/11/11 – 12/17/11 | 3 | Thermistor | 32 | 0.6 | 67.20155 | -49.28906 |
| T-14 | 7/18/14 – 6/23/17 | 0.5 | Thermistor | 31 | < 11 m deep – 0.5<br>> 11 m deep – 1.0 | 67.18127 | -49.56982 |
| T-15a | 8/17/16 - 5/20/17 | 0.25 | DS18B20 | 17 | < 15 m deep – 1.0<br>> 15 m deep – 3.0 | 67.18211 | -49.56827 |
| T-15b | 8/17/16 - 5/20/17 | 0.25 | DS18B20 | 17 | < 15 m deep – 1.0<br>> 15 m deep – 3.0 | 67.18205 | -49.56806 |
| T-15c | 8/17/16 - 5/20/17 | 0.25 | DS18B20 | 17 | < 15 m deep – 1.0<br>> 15 m deep – 3.0 | 67.18211 | -49.56848 |
| T-16 | 8/17/16 - 7/22/17 | 0.25 | DS18B20 | 18 | 0.5 | 67.18147 | -49.57025 |

5    **Table 2: Constants**

| Variable | Symbol | Value | Units | Reference |
|---|---|---|---|---|
| Specific Enthalpy | $H_m$ | 0 | J kg$^{-1}$ | |
| Ice Density | $\rho_i$ | 917 | kg m$^{-3}$ | van der Veen (2013) |
| Snow Density | $\rho_s$ | 300 | kg m$^{-3}$ | |
| Water Density | $\rho_w$ | 1000 | kg m$^{-3}$ | |
| Specific Heat Capacity | $C_p$ | 2097 | J kg$^{-1}$ K$^{-1}$ | van der Veen ( 2013) |
| Latent Heat of Fusion | $L_f$ | 3.335E+5 | J kg$^{-1}$ | van der Veen (2013) |
| Thermal Conductivity of Ice | $k_i$ | 2.1 | J m$^{-1}$ K$^{-1}$ s$^{-1}$ | van der Veen (2013) |
| Moisture Diffusivity | $\nu$ | 1.045E-4 | J m$^{-1}$ s$^{-1}$ | Aschwanden et al. (2012) |