# Peer review of "S1. Physical Evidence for Subsurface Fractures"

_The Cryosphere, 2018_

## Referee Comment (RC1) · Anonymous Referee #1 · 24 Jun 2018

Review Hills et al. 2018: Processes influencing near-surface heat transfer in Greenland's ablation zone, MS No.: tc-2018-51

The authors study the near-surface thermal regime of ice in the ablation zone of Greenland (GrIS) based on observations and simulations of ice temperatures. Observations essentially demonstrate that ice temperatures at ca. 20m depth (T0) are systematically higher than mean air temperatures. This corroborates earlier observations and the related hypothesis that the thermal regime in the ablation area of ice sheets is significantly influenced by other processes than heat conduction. Such is reconsidered in model experiments addressing the relative role of seasonal changes in heat content due to heat diffusion, vertical advection due to ablation, insulation by snow cover, radiation absorption and temperature gradients in the deeper layers. Progressive model

sensitivity studies reveal that the strongest effect is related to modifications of the bottom boundary conditions, while the other (near-surface) effects tend to cancel each other. The final simulation yields T0 being significantly colder than observed ice and air temperatures. It remains somehow unclear whether this is due to principal shortcomings of the semi-idealized simulatins or having not treated process having another strong impact on the thermal regime. Some interesting observational evidence is presented indicating that episodic refreezing at depth can play a role in this context, but these processes are not treated in the simulations. The reader is left with the rather basic conclusion that in the ablation zone of ice sheets, the mean annual air temperature can not straightforwardly be used to predict the near-surface ice temperature (nor vice-versa).

General remarks: The underlying research topic is discussed since long and is still relevant in various contexts, in particular regarding the mass and energy budget of GrIS and other ice sheets and their response to regional climate changes. Experimentally guided modeling is methodically appropriate to investigate that. The demonstrated observational background is valuable and based on sound experimental methods. Presentation of measured ice temperature records and their interpretation need some reconsideration concerning coherent time and depth referencing. The meteorological data are adequate but fall a bit short due to the lack of direct measurements of solar insolation and surface temperature, both being considered as primary drivers for subsurface temperature developments. On the other hand the existing data could have been better exploited for the purpose of this study. Specifically, just one site is considered finally, while the obviously present spatial variability is poorly addressed. Supplementary information is relevant, but is not used quatitatively because the associated processes are not treated in the simulations. The 1d, enthalpy-based model approach is appropriate, but description of some important details may be improved. Specifically, this concerns the grid setup (e.g. resolution) and numerics (discretisation, solving method, initialisation, parameterization of processes related to radiation absorption and water transport incl. refreeze and runoff). The chosen simulation strat-

egy is understood as semi-idealized sensitivity studies restricted to the mentioned five processes and is principally fine. Unfortunately, however, the lacking validation of the model results leaves a rather large gap between simulations and observations, which also hampers interpretations. Judgement of the robustness of the simulation results is largely impossible and could have been supported by a few additional runs considering diverse uncertainties (of e.g. input and parameterizations) and quantitave validation with observations (which contrasting to the author opinion must not necessarily result in inappropriate tuning). New and potentially important issues could thereby have been addressed with reasonable effort as well. Such might consider the hypothesized impact of e.g. the observed transient refreezing events or of the strong temperature inversions which govern near-surface exchange processes in the atmosphere above ice sheets. The latter aspect poses the question whether the model should not be forced by a measure of surface temperature instead of air temperature. The anticipated limitations of 1d modelling are not at all discussed. The manuscript is well structured and written, figures are mostly appropriate, exept of rather inconsistent treatment of time and depths. Achievement of an overview on backgrounds would benefit from compilation of an additional table compiling meteorological and ice temperature data. Major revision concerning the above mentioned aspects is recommended.

Specific comments: P1L21: Statement "...the five heat transfer mechanisms presented here..." may be put in actual perspective that just 4 processes are quantitatively treated P2L6: "high" refers to elevation or amount of melt? P2L11: Mention importance for interpretation of ice-cores or modelling ice flow P2L14: "van Everdingen, 1998" is rather unreproducible and incompletly given reference P2L17: The metric "depth of zero amplitude" needs to be reconsidered ("zero" in true sense is rather meaningless, e.g. Hooke 1976 refers to 1% of surface amplitude, the definition T0@-20m is rather subjective and questionable in view of the model setup (identical to bottom of the domain)). In this context, please also mention that >10m temperatures principally reflect the thermal conditions during the previous year, and in case of substantial refreeze from even before (all being attenuated though). P2L21: "forcing is constant"

probably means "periodic"? P3L3: Neither insulation by winter snow, nor radiation absorption or refreezing are processes unique to ablation areas P3L12: Check spelling of "Isunnguata Sermia" P3L27: Add more details about thermistor string measurements e.g. field accuracy not only concerns calibration but also depth referencing, uppermost sensors may be affected by solar heating, cables running through strong temperature gradients may affect signals through heat conduction, how was long-term drift determined (guessing that sensors were not excavated after the 3yr measurement period?) P3L28: neither HMP60 nor NR-Lite are Campbell Sci. products, please provide some more details about uncertainties of meteorological measurements i.e. comment e.g. radiation shielding of air termperature sensor (which depending on specific device used can be substantially affected), in what extent were radiation measurements reliabale concerning horizontal alignment, rime or snow? P3L32: Please add information how the notoriously noisy SR50 signal was processes in order to derive mentioned data (filtering, independent check of accumualted amount through e.g. stakes?) P4L13: Please again clarify criteria for T0 (allowed range, at what depth, which might be site dependend?) P4L14: It is not clear whether given figures are truly comparable. Please clarify period to which they refer (mean annual air temperature is clear, but according to Fig. 2 the ice temperature records have different length, is T0 calculated for same periods each?) P4L15: Temperature gradients are calculated for what depth range? P4L21: "…strings failed…": this again points towards an inconsistent treatment of data. Question is again in what extent it makes sense to compare records of different length. Tentatively, such inconsistencies also can explain why the average profiles Fig. 2 show different gradients in the upper 10m, which despite of the partly different locations might not be expected if data refer to same period of time. Still regarding Fig. 2, please also check how in the 33km-subplotsthe max. ice temperatures can be in excess of 0°C. Please also comment this in view of uncertainties (accuracy) of measurements. Fig. 2 shall be updated accordingly and some interpertations may be adjusted then. Further a new table may be added compiling an overview on atmopsheric and subsurface conditions for the same period of time (may be simulation period).

That may be extendend by other meteorological data (humidity, wind, radiation, snow). P4L33:"...T-14, transient heating events were observed...", which refers to Fig. 3 at depth ca. 10m. Fig. 2 (lowest panel, left) also shows data from this site. One here may note the exceptionally high data (close to 0°C) at depths 13-17m, which presumably reflects the (really strong) impact of another transient heating event. Question is, why this one does not trace in Fig. 3 (as according to the annotations the entire measurement period is considered in both figures). Another question refering to Fig. 3 in this context is, how one could understand that the melt layer persists throughout the whole winter 2017, which according to the meteorological data was not exceptionally warm. Humphrey et al. 2012 put forward some interesting ideas in this context, which may be reconsidered regarding the ablation zon of the ice sheet. P5L5: NR-Lite does not measure net shortwave radiation (which btw. shall never be negative) P5L10: Numbers should be given with respect to a common period of data (see comment above) P5L13: "...measure almost no winter snowpack at sites 27-km and 46-km...": how can such be understood, as surface slope is rather flat and uniform (so wind drift may not make the difference), neither elevation nor horizontal distances between stations are large enough to explain such strong precipitation gradients. Potential measurement problems (SR50, see above) shall be explored or previous work be checked (e.g. data from GIMEX-91 or PARCA or diverse model output). P5L19: "...IMAU s6, the second year is more typical for this area..." How can this be judged from Fausto et al 2009 ? P5L30: " ...simulates ice temperature to 20 m, a depth chosen for consistency with measured data" and "$T_0$ , is output from the bottom of the domain for each model experiment...": Constraining the model domain that way may introduce artifical effects to the $T_0$ simulation results.. A study shall be performed to investigate in what extent simulated magnitude and depth of $T_0$ depends on the size (depth) of the domain. The outcome may be considered in context of redefinition of $T_0$ (see respective comments above). P5L32:" model does not, nor is it meant to, simulate the surface mass balance." Doesn't that contradict equ. 4 (temperate) and statement at P6L24? The relation between vertical velocity and omega(H) needs to be clarified (independent or

how coupled?). P6L2: Please specify grid (resolution, constant or higher resolved near the surface and the bottom, which would be reasonable?) P6L15: Equ. 1 needs some clarification resp. "temperate ice diffusivity" . Understandig that melt is treated via w and radiation absorption in the last term of equ.1, what is the nature of temperature ice diffusivity then (latent heat flux in sense of evaporation/sublimation)? If so, please justify assumption that this is order of magnitude smaller than cold ice diffusivity and what is relation to mentioned impermeability of near surface ice. Please check units in Tab. 2 given for moisture diffusivity (shall be m2/s?), and add values for diffusivity for ice and snow (instead of conductivity and Cp) P7L2: ". . .limit water content": what happens with excess water? Treatment of water transport in the model (also in snow and with respect to refreezing (super-imposed ice?) generally needs to be better described .

P7L3 and L5: "fixed to the air temperature at the surface, " this is a questionable assumption, because it neglects the existence of the quasi-persistent inversion conditions above ice sheets and glaciers and associated surface exchange processes. Surface temperatures are significantly colder than air temperature which is not accounted in the simulations presented here. The potential impact (uncertainty) may be addressed by introducing a correspondingly changed upper boundary condition (representative figures for the difference between surface and air temperature may be retrieved from on of the various GIMEX-91/K-transect papers). A more general question is whether in an enthalpy based model also the boundary conditions need to specified in terms of enthalypy. P7L22 and Fig.6 : Treatment of depths needs clarification also in this context. In case of ablation the grid will be reduced by ca. 3m. In Fig. 6 however, this can not be seen (all profiles still expand to 20m). Seemingly profiles are plotted upon each other without taking account for this. The effect increases for experiment with accumulation, when surface changes are even larger (ca. 5m in total). The issue also has implications on intercomparison of profiles from single simulations or different experiments and the determination of depth with "zero" termperature change, finally. Regarding the latter, however, it is dsuggested to use some other criterium like 1% of surface amplitude.

[Figure]

In Fig. 6a the six profiles may be colour coded according to months (presumably), as the evolution of temperature profiles is not shown elsewhere. P8L7: Please add value for $k_s$ (rho=300kgm-3) in Tab. 2 and for diffusivities, respectively. Please clarify also whether for the near-surface nodes (representing rotten ice) another value is used. P8L12: Please clarify the treatment of radiation absorption for snow (extinction coefficient, bulk approach)? P8L19 and L21: The discussed absorption coefficients are valid for shortwave radiation, while in this study treatment of radiation absorption is based on measured net radiation. It may also be considered in this context that the intensity of solar incoming radiation (mostly counting for absorption within snow and ice) is smaller than measured net radiation. As solar incoming radiation and net radiation are strongly correlated the impact of using net radiaion in the presented simulations may be tested by e.g. applying a correction factor derived from literature references (GIMEX again). P8L25: with respect "Neumann instead of Dirichlet BC" the sensitivity of simulation results (of T0 essentially) on different size of the model domain (e.g. 50m instead of 20m) shall be tested by dedicated model runs. Similar regarding different magnitude of the prescribed gradient according to the observation T-15s), which is alos valuable in oder to judge the spatial variabilty and iherent impacts on model results and interpretations. P9L5: " Because the air can exceed the melting temperature in the summer while the ice cannot". Also in this context, using surface temperature instead of air temperature to drive the model would be most interesting (see comments above). P9L20: The given numbers for T0 may be reconsidered in light of above comments (criteria for allowed variability) P9L26: Simulations are also not able to reproduce the observed rel. max. of ice temperature at ca. 10m depth. May be this is due to inconsistent periods of time (as argued above). Fig. 7 is difficult to understand, needs major clarification and revison. Questions are: how can model output yield data below -20m? What is the purpose of comparing simulation results covering one year with average profiles compiled from observational records with different lenghts and gaps? Presumably the right-most red curve refers to T16. How can this extend to -20m while according to Fig. 2 the record stops at -10m? Information about air temperature during years which are

not considered in the simulation is not useful either, in particular as long as not refering to the same July-July period (as is for the simulations). P10L7: In view of the fact that specification of the bottom BC has the strongest impact on the results the according sensitivity to observed variability should be quantified. This has not at all to do with an inappropriate tuning exercise rather helps to constrain the respective reliability of the simulation results. P10L25: "than in other areas" P10L26: "the near-surface active layer in the ablation zone is small " .. replace small by shallow P10L29: Please clarify meaning "melting dynamics are complicated by the 20-m temperature gradient" also referring to sentence before (surface processes have weaker control) P11L6: Please add other references to this recently focussed issue (e.g. Renneralm 2013, Steger 2017, Smith 2017 or Andrews et al 2014, Nature 514). Humphrey et al. 2012 appears most interesting in the overall context of the paper (although referring to firm area) and allows some aspects to be put in larger context (e.g. observations of Tair-To at ele-vatons below the ELA) P11L2: Please here also consider earlier comments (P4L33). In this context, too, please check and comment the argue that the event shown in Fig. 4a (depth 7m) corresponds to the one shown in Fig.2 (lowest panel left, depth 15m) P11L33: Also consider whether water bodies can be advected from higher elevations ? The results may generally be put in better context to related investigations in the accumulation area (e.g. Humphrey et al 2012) Table 1: please elevation of sites Table 2: H is not a constant. Please consider adding a new Table (climatology of average temperatures of air and ice, plus other meteo parameters). Data should be based on same period of time i.e., simulation period preferably)

---

## Referee Comment (RC2) · Anonymous Referee #2 · 2 Jul 2018

Hills et al. Investigate heat transfer from the atmosphere to the ice in west Greenland's ablation zone and conclude based that air temperature can not predict the near-surface ice temperature. While the topic is interesting and the data presented is valuable, the modelling part does not lead to strong conclusions. I would recommend to rework the paper, focus on the data analysis, especially the very interesting transient heating events, ideally deriving quantitative conclusions on the amount of water necessary to reproduce them and modify the modelling part significantly. Some assumptions for the model-part seem inappropriate or at least too weakly constrained in order to judge if the derived conclusions are valid. Comparison with Promice stations in the area may improve the applicability and validate some of the factors. Central is an in-depth check of the boundary (6) for the modelling part. This should be discussed thoroughly. Take a

promice station in the area, convert outgoing longwave radiation with Stefan Boltzmann to surface temperature and plot vs air temperature. This plot is necessary for the paper and will show if (6) is OK to use at all. Also, put more effort into explaining the massive local-scale variability of ice temperatures which is surprising to me. Generally, I would suggest major revisions, a change in focus of the paper and/or a substantial improvement of the modelling part.

P2 L12: add reference? L19 – if it is often used, add more references. This is potentially an important issue. P3 L1: this statement is true for all ablation areas and is not GrIS specific!? L7-8. Can observation resolve these processes quantitatively? In my view this is the 'issue' with observations, that you end up with a 'bulk' signal combining different processes. Consider rewording L13. 30 km below. . . is confusing. Rewrite. Also 1500m elevation in this area? L13: I miss info on elevation of the sites. Please add to Tab1 L24: reduce numbers of sign digits L27: how was the field calibration done? L29: near surface: how near? Did you have a radiation shield? Add a picture of the AWS L34: how often were they re-aligned? Estimating from fig 3 there was a ca 6 m surface elevation change. This pushes your air temperature quite far into the near-surface inversion. How do you account for this.

P4 L4: why five strings? L11: how does that fit with field calibration P3 L27? Fig 2 has a problem as it does not seem to account for changing surface height – I deduce this from the fact that each dot represents a sensor. It is, however, over time in different depths. Especially for the uppermost sensors this creates an issue. Suggest to correct the time-series using the known surface elevation changes. For the mean annual air temperatures. How do promice stations in the area fit to that? I am a bit worried about the height-above terrain and the radiation shield issue. Are air temperatures ventilated? Also: why don't you sort them logically, i.e. from 27, 33 to 46? Add axes descriptions. Label a-g L15: explain what positive and negative means for the gradients L15-17: unclear sentence. What do you mean? L21: makes things difficult. Consider only showing averages for concurrent times or at least full years? The rest

does not make sense to me. L26-27: this is a very large near-surface variability and not easy to understand. How about radiation errors of the uppermost sensors? Figure 3: misleading. The depth of the sensor during installation is shown and not sensor depth. This is a massive difference. Strongly recommend to correct it for that. There are some interesting features and it is impossible to tell whether these are artefacts or reality. Which ones are the warming events you refer to? how about the vertical red lines, i.e. end of 2015 or ca may 2016 further down or again some time in spring 2017. Suggest indicating the warming events you refer to later in the discussion. And what with the horizontal redish bodies in late 2014 for instance in ca 8 m? L29: refer here to something you mark in the fig. Fig. 4: do they cool off again afterwards? Why don't you show the same time-steps in c and d as you do in a and b? how does meteorology play in here? Was there a rain-event preceding this? Impossible to tell if you don't state when it happened. Fig. 5. b) consider combining b and c as surface elevation change D)If you measured with a NRlite p3, L30, you don't get net shortwave radiation to my understanding. P5 L5 see above. NOT net-shortwave L6: sounds low to me. Compare to Promice KAN stations? This could also help discriminating into SW and LW. L9: consider rewording 'warm bias' L15-20: I believe that winter 2016 was particularly snow-poor and thus not particularly 'representative'. Check if that is true on regional scale and include in discussion. Part 4 and 4.1. See issue with boundary condition (6). Boundary condition (7) is clearly not valid as yu state yourself and adapt later. But why then introducing it and calling it a boundary condition? This does not make too much sense to me. The choice of experiments seems arbitrary. How about turbulent heat exchange? P8 L9: net radiation L25: -0.05°C/m, P9 L5: also in other occasions surface temperature and air temperature can be very different L20: and P10 2-4this seems trivial. If you keep the upper part of the 'trumpet' as it is and induce a gradient in 20m you will end up colder there. Is there a value in it? L24-25. What do you mean? Unique to the ablation zone….larger than other areas? Other ablation areas? The sentence does not make sense to me L30-34. Interesting. But what is the reason for such different ice-packages coming to the same site just a few dozens of m away?

This should be better discussed and this high variability is potentially a significant result of the paper. How do satellite-derived surface temperatures vary spatially? P11: L6. Refreezing is a big topic these days. Consider adding more refs Could the amount of water be estimated that would be necessary in order to reproduce the warming caused by latent heat you observe? This would be interesting. L26-29. Check out Colgan et al. Crevasse review. There is a process mentioned of crevasses 'growing' from below to the surface.

P5-9:all that would point to ice being colder than air. But you show in fig 2 it is warmer. I doubt that the modelling serves as a base for this conclusion.

---

## Author Comment (AC1) · 30 Jul 2018

Comments and associated responses for:

**Processes influencing heat transfer in the near-surface ice of Greenland's ablation zone**

Benjamin H. Hills et al. 2018

**Anonymous Referee #1**

Author responses are in blue.

Review Hills et al. 2018: Processes influencing near-surface heat transfer in Greenland's ablation zone, MS No.: tc-2018-51

The authors study the near-surface thermal regime of ice in the ablation zone of Greenland (GrIS) based on observations and simulations of ice temperatures. Observations essentially demonstrate that ice temperatures at ca. 20m depth (T0) are systematically higher than mean air temperatures. This corroborates earlier observations and the related hypothesis that the thermal regime in the ablation area of ice sheets is significantly influenced by other processes than heat conduction. Such is reconsidered in model experiments addressing the relative role of seasonal changes in heat content due to heat diffusion, vertical advection due to ablation, insulation by snow cover, radiation absorption and temperature gradients in the deeper layers. Progressive model sensitivity studies reveal that the strongest effect is related to modifications of the bottom boundary conditions, while the other (near-surface) effects tend to cancel each other. The final simulation yields T0 being significantly colder than observed ice and air temperatures. It remains somehow unclear whether this is due to principal shortcomings of the semi-idealized simulatins or having not treated process having another strong impact on the thermal regime. Some interesting observational evidence is presented indicating that episodic refreezing at depth can play a role in this context, but these processes are not treated in the simulations. The reader is left with the rather basic conclusion that in the ablation zone of ice sheets, the mean annual air temperature can not straightforwardly be used to predict the near-surface ice temperature (nor vice-versa).

General remarks: The underlying research topic is discussed since long and is still relevant in various contexts, in particular regarding the mass and energy budget of GrIS and other ice sheets and their response to regional climate changes. Experimentally guided modeling is methodically appropriate to investigate that. The demonstrated observational background is valuable and based on sound experimental methods. Presentation of measured ice temperature records and their interpretation need some reconsideration concerning coherent time and depth referencing. The meteorological data are adequate but fall a bit short due to the lack of direct measurements of solar insolation and surface temperature, both being considered as primary drivers for subsurface temperature developments. On the other hand the existing data could have been better exploited for the purpose of this study. Specifically, just one site is considered finally, while the obviously present spatial variability is poorly addressed. Supplementary information is relevant, but is not used quatitatively because the associated processes are not treated in the simulations. The 1d, enthalpy-based model approach is appropriate, but description of some important details may be improved. Specifically, this concerns the grid setup (e.g. resolution) and numerics (discretisation, solving method, initialisation, parameterization of processes related to radiation absorption and water transport incl. refreeze and runoff). The chosen simulation strategy is understood as semi-idealized sensitivity studies restricted to the mentioned five processes and is principally fine. Unfortunately, however, the lacking validation of the model results leaves a rather large gap between simulations and observations, which also hampers interpretations. Judgement of the robustness of the simulation results is largely impossible and could have been supported by a few additional runs considering diverse uncertainties (of e.g. input and parameterizations) and quantitave validation with observations (which contrasting to the author opinion must not necessarily result in inapproriate tuning). New and potentially important issues could thereby have been addressed with reasonable effort as well. Such might consider the hypothesized impact of e.g. the observed transient refreezing events or of the strong temperature inversions which govern near-surface exchange processes in the atmosphere above ice sheets. The latter aspect poses the question whether the model should not be forced by a measure of surface temperature instead of air temperature. The anticipated limitations of 1d modelling are not at all discussed. The manuscript is well structured and written, figures

are mostly appropriate, exept of rather inconsistent treatment of time and depths. Achievement of an overview on backgrounds would benefit from compilation of an additional table compiling meteorological and ice temperature data. Major revision concerning the above mentioned aspects is recommended.

Thank you for these comments. We hope that in our revision of the manuscript we have addressed the main concerns given here. We have changed the depth referencing in all figures to be consistent, that is with depth plotted at the time of measurement rather than the time of installation. We understand that our meteorological data may fall short in terms of a mass-balance meteorology studies, but we argue that they are appropriate for our analysis of subsurface ice temperature. We have added some amount of analysis of these meteorological data in the supplement and compared them to the neighboring KAN_L station.

We appreciate the frustration in the gap between model and measured results, especially in our lack of a thorough explanation for the possible processes which could fill that gap. We have added a fairly extensive analysis with the subsurface refreezing events, showing that they could easily provide enough energy to make up for the temperature difference that we observe. We hope that this addresses your concern.

Specific comments:

P1L21: Statement ". . .the five heat transfer mechanisms presented here. . ." may be put in actual perspective that just 4 processes are quantitatively treated

We added a quantitative analysis in the discussion for the subsurface refreezing process. We believe that this is satisfactory so that we can maintain the original phrasing here.

P2L6: "high" refers to elevation or amount of melt?

Changed to:

"Bare ice regions of the Greenland ice sheet have high summer melt rates,"

P2L11: Mention importance for interpretation of ice-cores or modelling ice flow

Added:

"and is therefore important for ice flow modeling (e.g. *Meierbachtol et al.*, 2015) as well as interpretation of borehole temperature measurements (*Harrington et al.*, 2015; *Lüthi et al.*, 2015; *Hills et al.*, 2017)"

P2L14: "van Everdingen, 1998" is rather unreproducible and incompletly given reference

Changed to:

"van Everdingen, R. O. (1998), *Multi-language glossary of permafrost and related ground-ice terms*, International Permafrost Association, Calgary, Alberta,CA."

P2L17: The metric "depth of zero amplitude" needs to be reconsidered ("zero" in true sense is rather meaningless, e.g. Hooke 1976 refers to 1% of surface amplitude, the definition T0@-20m is rather subjective and questionable in view of the model setup (identical to bottom of the domain)). In this context, please also mention that >10m temperatures principally reflect the thermal conditions during the previous year, and in case of substantial refreeze from even before (all being attenuated though).

This was addressed in the line above P2L15:

"Seasonal air temperature oscillations are diminished with depth into the ice, until they are negligible (i.e. ~1%) at a 'depth of zero annual amplitude' (*van Everdingen*, 1998)."

P2L21: "forcing is constant" probably means "periodic"?

Changed to:

"and interannual climate variations are minimal"

P3L3: Neither insulation by winter snow, nor radiation absorption or refreezing are processes unique to ablation areas

Changed to:

"The contrast of a wintertime snowpack to bare ice in the summer enables an insulating effect during the winter months. The deep penetration of solar radiation into bare ice results in subsurface heating and melting (Brandt and Warren, 1993; Liston et al., 1999)."

P3L12: Check spelling of "Isunnguata Sermia"

This is the correct spelling. There are in fact multiple spellings in the literature and maps, but this is the dominant form.

P3L27: Add more details about thermistor string measurements e.g. field accuracy not only concerns calibration but also depth referencing, uppermost sensors may be affected by solar heating, cables running through strong temperature gradients may affect signals through heat conduction, how was long-term drift determined (guessing that sensors were not excavated after the 3yr measurement period?)

We changed all of the figures to be depth-referenced in the same way, with temperatures plotted at the depth of measurement rather than the depth of installation.

Added:

"Because each temperature sensor is in a black casing, measurements are discarded when the sensor lies on the surface exposed to the sun."

We argue that the lateral heat transfer into perpetually cold ice does not allow heat to move down the cable far at all. This is confirmed by the fact that the cables are frozen solidly into the ice just below the surface.

Admittedly, thermistors can have long term drift. While in most cases, our analysis is with temperature changes on the order of degrees rather than tenths of a degree (where drift would become more important), we have added a section to the supplement to address this.

P3L28: neither HMP60 nor NR-Lite are Campbell Sci. products, please provide some more details about uncertainties of meteorological measurements i.e. comment e.g. radiation shielding of air termperature sensor (which depending on specific device used can be substantially affected), in what extent were radiation measurements reliabale concerning horizontal alignment, rime or snow?

Added: "(Vaisala HMP60 with a radiation shield)"

We also added a comparison to PROMICE data (specifically for air temperature and radiation) in the supplement which should help justify the reliability of our measurements.

P3L32: Please add information how the notoriously noisy SR50 signal was processes in order to derive mentioned data (filtering, independent check of accumualted amount through e.g. stakes?)

Added:

"Data from the sonic ranger are filtered manually, removing any obvious outliers (more than 0.5 m from the surrounding measurements)."

Also, we state that all meteorological data are collapsed to a daily mean.

P4L13: Please again clarify criteria for T0 (allowed range, at what depth, which might be site dependend?)

Added:

"The mean value from the lowermost sensor (analogous to $T_0$)"

P4L14: It is not clear whether given figures are truly comparable. Please clarify period to which they refer (mean annual air temperature is clear, but according to Fig. 2 the ice temperature records have different length, is T0 calculated for same periods each?)

Figure 2 was changed. Temperature records that do not complete a full year are given transparency to make it clear that they are different from the others. Having said that, $T_0$ should be comparable because it is below any seasonal variations.

P4L15: Temperature gradients are calculated for what depth range?

Added:

"Temperature gradients are calculated by fitting a line to the mean temperature of the four lowermost sensors for each string."

P4L21: ". . .strings failed. . .": this again points towards an inconsistent treatment of data. Question is again in what extent it makes sense to compare records of different length. Tentatively, such inconsistencies also can explain why the average profiles Fig. 2 show different gradients in the upper 10m, which despite of the partly different locations might not be expected if data refer to same period of time. Still regarding Fig. 2, please also check how in the 33km-subplotsthe max. ice temperatures can be in excess of 0∘C. Please also comment this in view of uncertainties (accuracy) of measurements. Fig. 2 shall be updated accordingly and some interpertations may be adjusted then. Further a new table may be added compiling an overview on atmopsheric and subsurface conditions for the same period of time (may be simulation period). That may be extendend by other meteorological data (humidity, wind, radiation, snow).

Some of these issues had already been addressed in the original manuscript. For example, "The missing summer period explains the mean cold bias near the surface." However, seeing as the other reviewer had similar issues, much of the prose was changed for clarity, and Figure 2 has been updated in several ways. Most importantly, the three strings that did not collect an entire year of data are given some amount of transparency to show that they are different from the others, and an explanation is given that they are averaged over a shorter time period.

Seeing that we do not want the focus to be on atmospheric conditions, we have not added the requested table, although more atmospheric analysis was added to the supplement.

P4L33:" . . .T-14, transient heating events were observed. . . ", which refers to Fig. 3 at depth ca. 10m. Fig. 2 (lowest panel, left) also shows data from this site. One here may note the exceptionally high data (close to 0∘C) at depths 13-

17m, which presumably reflects the (really strong) impact of another transient heating event. Question is, why this one does not trace in Fig. 3 (as according to the annotations the entire measurement period is considered in both figures). Another question refering to Fig. 3 in this context is, how one could understand that the melt layer persists throughout the whole winter 2017, which according to the meteorological data was not exceptionally warm. Humphrey et al. 2012 put forward some interesting ideas in this context, which may be reconsidered regarding the ablation zon of the ice sheet.

Figure 3 has been changed for clarity on these issues.

We do not know how to interpret the second half of the comment: Figure 3 shows a cold surface for the winter of 2017.

P5L5: NR-Lite does not measure net shortwave radiation (which btw. shall never be negative)

Deleted "shortwave"

P5L10: Numbers should be given with respect to a common period of data (see comment above)

The measurements from the weather station and from T-14 are over the same period. As for the other temperature strings, we argue that the deeper temperatures (below ~10 m) should be comparable to the mean air temperature because they are below seasonal variations.

P5L13: ". . .measure almost no winter snowpack at sites 27-km and 46-km. . .": how can such be understood, as surface slope is rather flat and uniform (so wind drift may not make the difference), neither elevation nor horizontal distances between stations are large enough to explain such strong precipitation gradients. Potential measurement problems (SR50, see above) shall be explored or previous work be checked (e.g. data from GIMEX-91 or PARCA or diverse model output).

It is more likely that the difference in time rather than location that is making this difference.

Added:

"during the time period over which those data were collected (2011-2013)."

P5L19: ". . .IMAU s6, the second year is more typical for this area. . ." How can this be judged from Fausto et al 2009 ?

Changed to KAN_L (van As et al., 2012)

P5L30: " . . .simulates ice temperature to 20 m, a depth chosen for consistency with measured data" and "T0 , is output from the bottom of the domain for each model experiment. . .": Constraining the model domain that way may introduce artifical effects to the T0 simulation results.. A study shall be performed to investigate in what extent simulated magnitude and depth of T0 depends on the size (depth) of the domain. The outcome may be considered in context of redefinition of T0 (see respective comments above).

This is why an insulating boundary condition, equation (7), was chosen. For those experiments where there is no temperature gradient at the bottom of the domain, there is no energy flux from below, and we are capturing only the near-surface processes. It should not make a difference how far the domain extends, we are capturing all the processes at the surface.

P5L32:" model does not, nor is it meant to, simulate the surface mass balance." Doesn't that contradict equ. 4 (temperate) and statement at P6L24?

The focus of the paper is subsurface ice temperature rather than mass balance. For that reason, we feel that we can ignore some of the more complicated atmospheric processes which would be important for mass balance.

Added:

"The model, its boundary conditions, and the experiments are all designed to test heat transfer processes within the ice itself. To maintain focus on ice processes, we ignore any atmospheric effects above the ice surface such as turbulent heat fluxes."

The relation between vertical velocity and omega(H) needs to be clarified (independent or how coupled?).

We are confused by this comment: they are coupled through the advection term in equation (1), that is

$$w\partial_z H$$

As we state in section 4.2.1, the vertical velocity, $w$, is not calculated from any model output, it comes directly from the measured surface lowering.

P6L2: Please specify grid (resolution, constant or higher resolved near the surface and the bottom, which would be reasonable?)

Added:

"and 0.5-m mesh spacing refined to 2 cm near the surface."

P6L15: Equ. 1 needs some clarification resp. "temperate ice diffusivity" . Understandig that melt is treated via w and radiation absorption in the last term of equ.1, what is the nature of temperature ice diffusivity then (latent heat flux in sense of evaporation/sublimation)? If so, please justify assumption that this is order of magnitude smaller than cold ice diffusivity and what is relation to mentioned impermeability of near surface ice. Please check units in Tab. 2 given for moisture diffusivity (shall be m2/s?),

This is the standard for enthalpy models (see Aschwanden et al., 2012; Brinkerhoff and Johnson, 2013). Temperate ice diffusivity means that some amount of energy can diffuse when the enthalpy is above the temperate value, $H_m$. This is needed for numerical stability. We argue that it is much smaller than the cold ice diffusivity because the latent heat flux (water flux) should be very small.

Units are changed to kg m$^{-1}$ s$^{-1}$ which is that used in (Aschwanden et al., 2012; Brinkerhoff and Johnson, 2013).

and add values for diffusivity for ice and snow (instead of conductivity and Cp)

We prefer conductivity and heat capacity. The diffusivity is provided as a function of those constants in equation (2).

P7L2: ". . .limit water content": what happens with excess water? Treatment of water transport in the model (also in snow and with respect to refreezing (super-imposed ice?) generally needs to be better described .

Added:

"with any excess water immediately leaving the model domain as surface runoff."

P7L3 and L5: "fixed to the air temperature at the surface, " this is a questionable assumption, because it neglects the existence of the quasi-persistent inversion conditions above ice sheets and glaciers and associated surface exchange processes. Surface temperatures are significantly colder than air temperature which is not accounted in the simulations presented here. The potential impact (uncertainty) may be addressed by introducing a correspondingly changed upper boundary condition (representative figures for the difference between surface and air temperature may be retrieved from on of the various GIMEX-91/K-transect papers).

As stated above, we are testing only processes within the snow/ice itself, no atmospheric conditions. The best atmospheric measurement that we have at our site for constraining the surface boundary condition is the near-surface air temperature. While this may not be ideal for more intricate models that are attempting to constrain the surface mass balance, we argue that it suffices in our case. Our focus is on the heat transfer processes within the ice, thus a discussion of atmospheric processes is beyond the scope of this paper.

A more general question is whether in an enthalpy based model also the boundary conditions need to specified in terms of enthalypy.

For the sake of clarity, it makes sense to give values in temperature rather than enthalpy.

However, we added "Both boundary conditions are with no liquid water content, $\omega = 0$."

P7L22 and Fig.6 : Treatment of depths needs clarification also in this context. In case of ablation the grid will be reduced by ca. 3m. In Fig. 6 however, this can not be seen (all profiles still expand to 20m). Seemingly profiles are plotted upon each other without taking account for this. The effect increases for experiment with accumulation, when surface changes are even larger (ca. 5m in total). The issue also has implications on intercomparison of profiles from single simulations or different experiments and the determination of depth with "zero" termperature change, finally. Regarding the latter, however, it is dsuggested to use some other criterium like 1% of surface amplitude. In Fig. 6a the six profiles may be colour coded according to months (presumably), as the evolution of temperature profiles is not shown elsewhere.

This was explained in the text: "Because the vertical coordinate, $z$, in the model domain is treated as a distance from this moving surface, ablation brings simulated ice closer to the surface boundary."

However, through many of the comments, we have learned that our treatment of depth in this manuscript may have been confusing to some readers. For that reason, Figure 2 was changed to more closely match Figures 5 and 6 (previously 6 and 7), with a "depth below the ice surface at the time of measurement" rather than the initial ice depth.

As far as coloring the temperature profiles for a transient representation, this would take away from what we are trying to illustrate in this figure; namely, the convergent behavior near the bottom of the domain as well as the *differences* between experiments.

P8L7: Please add value for ks (rho=300kgm-3) in Tab. 2 and for diffusivities, respectively. Please clarify also whether for the near-surface nodes (representing rotten ice) another value is used.

Added $k_s$ to Table 2.

Added:

"All constants for the rotten cryoconite layer are the same as that for ice."

P8L12: Please clarify the treatment of radiation absorption for snow (extinction coefficient, bulk approach)?

This was addressed in the original manuscript:

"We assume that all this radiative energy is absorbed in the uppermost 20 cm, the rotten cryoconite layer, and if snow is present the melt production immediately drains to that cryoconite layer."

P8L19 and L21: The discussed absorption coefficients are valid for shortwave radiation, while in this study treatment of radiation absorption is based on measured net radiation. It may also be considered in this context that the intensity of solar incoming radiation (mostly counting for absorption within snow and ice) is smaller than measured net radiation. As solar incoming radiation and net radiation are strongly correlated the impact of using net

radiaion in the presented simulations may be tested by e.g. applying a correction factor derived from literature references (GIMEX again).

We only discuss the absorption coefficients for the sake of argument that all radiation will be absorbed in the rotten cryoconite layer. We did do some experimenting with a more explicit radiation model, using shortwave radiation from the nearby KAN_L station and different absorption coefficients for snow, ice, and rotten ice. We got a very similar answer to what we present here. Therefore, we choose to keep the analysis more streamlined by putting all the radiative energy in the rotten cryoconite layer. Again, this paper is about ice temperature below the surface, and we feel that getting overly detailed about radiation penetration and making an attempt to model it in detail is not essential to our conclusions and is beyond the scope of this paper.

P8L25: with respect "Neumann instead of Dirichlet BC" the sensitivity of simulation results (of T0 essentially) on different size of the model domain (e.g. 50m instead of 20m) shall be tested by dedicated model runs. Similar regarding different magnitude of the prescribed gradient according to the observation T-15s), which is alos valuable in oder to judge the spatial variabilty and iherent impacts on model results and interpretations.

This is the point of using the insulating boundary condition, it will not be any different for a simulation to 50 m. In fact, the original theory for these problems uses a semi-infinite half-space (Carslaw and Jaeger , 1959).

P9L5: " Because the air can exceed the melting temperature in the summer while the ice cannot". Also in this context, using surface temperature instead of air temperature to drive the model would be most interesting (see comments above).

We wanted to use a measured value that is independent of our ice temperature measurements, so the air temperature is what we have available.

P9L20: The given numbers for T0 may be reconsidered in light of above comments (criteria for allowed variability)

As stated above, we feel that $T_0$ is deep enough into the ice that it should be an aggregated sum of any near-surface processes and is low enough that it does not see any seasonal variations.

P9L26: Simulations are also not able to reproduce the observed rel. max. of ice temperature at ca. 10m depth. May be this is due to inconsistent periods of time (as argued above). Fig. 7 is difficult to understand, needs major clarification and revison.

In reference to the time period, this is why we are using $T_0$ and what we consider to be a 'typical' year for the meteorological data. If the meteorological data are truly representative of the interannual mean, then the $T_0$ values should be comparable because they are below any seasonal variations.

Questions are: how can model output yield data below -20m?

We changed everything in the model output to extend to 21 m, this has been addressed throughout the paper.

What is the purpose of comparing simulation results covering one year with average profiles compiled from observational records with different lenghts and gaps?

Again, $T_0$ should be an aggregated sum of all of the thermal effects over several years. We argue that the comparison is appropriate if the model input data are truly representative of a 'typical' season.

Presumably the right-most red curve refers to T16. How can this extend to -20m while according to Fig. 2 the record stops at -10m?

The right-most red curve refers to T-14 which does extend to 21 m depth at the time of installation.

Information about air temperature during years which are not considered in the simulation is not useful either, in particular as long as not refering to the same July-July period (as is for the simulations).

We disagree. It provides some context for recent seasons. The $T_0$ value that we measure will have been influenced by all the recent air temperatures. With this figure, we are trying to show that the measured ice temperatures are significantly warmer than the simulation results as well as than any of the measured air temperatures.

P10L7: In view of the fact that specification of the bottom BC has the strongest impact on the results the according sensitivity to observed variability should be quantified. This has not at all to do with an inappropriate tuning exercise rather helps to constrain the respective reliability of the simulation results.

We added two additional tests to the final experiment in our model case study (the subsurface temperature gradient experiment). In addition to the original gradient of -0.05, we add an upper and lower bound based on measured values (i.e. +/- 0.15 C/m). These results were added to Figure 5e (previously Figure 6).

P10L25: "than in other areas"

This sentence was changed to address a comment from the other reviewer.

P10L26: "the near-surface active layer in the ablation zone is small " .. replace small by shallow

Changed to "shallow"

P10L29: Please clarify meaning "melting dynamics are complicated by the 20-m temperature gradient" also referring to sentence before (surface processes have weaker control)

Changed prior sentence to:

"Therefore, the surface boundary condition has weak influence on diffusion for ice well below the surface. This is in contrast to the accumulation zone where new snow is advected downward so the surface temperature quickly influences that at depth."

P11L6: Please add other references to this recently focussed issue (e.g. Renneralm 2013, Steger 2017, Smith 2017 or Andrews et al 2014, Nature 514). Humphrey et al. 2012 appears most interesting in the overall context of the paper (although referring to firm area) and allows some aspects to be put in larger context (e.g. observations of Tair-To at elevatons below the ELA).

Based on requests from the other reviewer, we added what we thought were relevant references for large-scale latent heating in the following paragraph (e.g. Phillips et al., 2013; Poinar et al., 2016).

P11L2: Please here also consider earlier comments (P4L33). In this context, too, please check and comment the argue that the event shown in Fig. 4a (depth 7m) corresponds to the one shown in Fig.2 (lowest panel left, depth 15m)

Addressed by changing the figures for clarification.

P11L33: Also consider whether water bodies can be advected from higher elevations ? The results may generally be put in better context to related investigations in the accumulation area (e.g. Humphrey et al 2012)

We address this at the end of the discussion section:

"The only other logical mechanism for the warm offset between measurements and model results would be warming from below through a positive subsurface temperature gradient. That gradient could be created by residual heat from

the exceptionally hot summers of 2010 and 2012 (Tedesco et al., 2013), but it is unlikely because the ablation rates are so high that any ice warmed during those years has likely already melted. Alternatively, deeper latent heating from an upstream crevasse field is a plausible alternative; however, in this area full-depth temperature profiles do not show deeper ice to be anomalously warmed except in one localized case (Hills et al., 2017)."

Table 1: please elevation of sites

Added.

Table 2: H is not a constant.

Hm is the 'reference enthalpy' for melting, but that was not clear so its name was changed.

Please consider adding a new Table (climatology of average temperatures of air and ice, plus other meteo parameters). Data should be based on same period of time i.e., simulation period preferably)

We feel that anything that this table would accomplish has already been addressed in Figure 6 (previously Figure 7). Having said that, we admit that aspects of this figure were confusing, so it has been updated for clarity.

**Processes influencing  heat transfer in the **near-surface ice of** Greenland's ablation zone**

Benjamin H. Hills[1,2], Joel T. Harper[2], Toby W. Meierbachtol[2], Jesse V. Johnson[3], Neil F. Humphrey[4], Patrick J. Wright[5,2]

[1]Department of Earth and Space Sciences, University of Washington, Seattle, Washington, USA
[2]Department of Geosciences, University of Montana, Missoula, Montana, USA
[3]Department of Computer Science, University of Montana, Missoula, Montana, USA
[4]Department of Geology and Geophysics, University of Wyoming, Laramie, Wyoming, USA
[5]Inversion Labs LLC, Wilson, Wyoming, USA

*Correspondence to*: Benjamin H. Hills (bhills@uw.edu)

**Abstract.** To assess the influence of various  heat transfer mechanisms on the temperature structure of ice near the -surface  of Greenland's ablation zone, we  compare highly resolved in situ measurements  with simplified thermal modeling experiments. Seven separate temperature strings were installed at three different field sites, each with between 17 and 32 sensors and extending up to 21 meters below the surface. In one string, temperatures were measured every 30 minutes, and the record is continuous for more than three years. We use these measured ice temperatures to constrain modeling analyses focused on four isolated processes and assess the relative importance of each for the near-surface ice temperature: 1) the moving boundary of an ablating surface, 2) thermal insulation by snow, 3) radiative energy input, and 4) subsurface ice temperature gradients below the seasonally active near-surface layer. In addition to these four processes, transient heating events were observed in two of the temperature strings. Despite no observations of meltwater pathways to the subsurface, these heating events are likely the refreezing of liquid water below 5-10 meters of cold ice. Together with subsurface refreezing, the five heat transfer mechanisms presented here account for measured differences of up to 3°C between the mean annual air temperature and the ice temperature at the depth where annual temperature variability is dissipated.  Thus, in Greenland's ablation zone, the mean annual air temperature is not a reliable predict of the near-surface ice temperature, as is commonly assumed.

**1 Introduction**

~~Ice sheets are coupled to the atmosphere at their upper surface through an exchange of mass and energy. Understanding this coupling is important for knowing the ice sheet's surface mass balance and its associated contribution to sea level rise. In particular, the Greenland Ice Sheet (GrIS) has shown a change toward a more negative surface mass balance, which constitutes at least half of its contribution to recent sea level rise (van den Broeke et al., 2009; Enderlin et al., 2014). Inhigh melt,(Wharton et al., 1985)M. P.climate systems~~, we must understand the processes that control near-surface heat transfer in bare ice.

Heat transfer at the ice surface is dominated by thermal diffusion from the overlying air (Cuffey and Paterson, 2010). Seasonal air temperature oscillations are diminished with depth into the ice, until they are negligible (i.e. ~1%) at a 'depth of zero annual amplitude' (van Everdingen, 1998). The exact location of this depth is dependent on the thermal diffusivity of the material through which heat is conducted as well as the period of oscillations (Carslaw and Jaeger, 1959; pp. 64-70). In theory, the temperature at the depth of zero annual amplitude, a value we will call $T_0$, is approximately constant and equal to the mean annual air temperature. In snow and ice, the depth of zero annual amplitude is approximately 10 and 15 m respectively (Hooke, 1976). For this reason, studies in the cryosphere often use $T_0$ as a proxy for the mean air temperature, drilling to 10 or more meters to measure the snow or ice temperature at that depth (Loewe, 1970; Mock and Weeks, 1966).

In places where heat transfer is purely diffusive, the snow or ice is homogeneous, and interannual  climate variations are minimal, $T_0$ is a good approximation for the mean air temperature. However, prior studies have shown that, in many areas of glaciers and ice sheets, the relationship between air and ice temperatures can be substantially altered by additional heat transfer processes. For example, in the percolation zone, infiltration and refreezing of surface meltwater warm the subsurface (Humphrey et al., 2012; Müller, 1976). Studies have also revealed ice anomalously warmed by 5°C or more in the ablation zone (Hooke et al., 1983; Meierbachtol et al., 2015), but the mechanisms for this are unclear.

Hooke et al. (1983) explored the impacts of several heat transfer processes within near-surface ice at Storglaciären and the Barnes Ice Cap. They focused on the wintertime snowpack which acts as insulation to cold air temperatures but is permeable to meltwater percolation. Their results showed that the average ice temperature at and below the equilibrium line of those glaciers tends to be higher than the mean annual air temperature. They attributed the observed difference mainly to snow insulation because the strength of their measured offset was correlated to the thickness of the snowpack.

In this study, we expand the analysis of Hooke et al. (1983) and turn focus to the GrIS ablation zone with nearsurface temperature profiles from seven locations. We use our temperature measurements in conjunction with a one-dimensional heat transfer model to assess heat transfer processes in this area. The processes which make the ablation zone different from other areas of a glacier or ice sheet are, first, that the ice surface spends much of the summer period pinned at the melting point, despite slightly warmer air temperatures. Next, high ablation rates counter emerging ice flow, removing the ice surface and exposing deeper ice, along with its heat content, to the surface. The contrast of a wintertime snow to bare ice in the summer enables an insulat effect during the winter months. The  deep penetration of solar radiation into bare ice results in subsurface heating and melting ( Brandt and Warren, 1993; Liston et al., 1999) . Finally, surface melt can move through open fractures, carrying latent heat with it to deeper and colder ice, and upon refreezing, the meltwater warms that ice below the surface (Jarvis and Clarke, 1974; Phillips et al., 2010).

Our

 near-surface temperature observations represent an aggregated sum of the processes mentioned above. A numerical model can be used to partition the relative importance of those processes, but only with measurements in hand as validation. Therefore, confidently constraining the role of near-surface heat transfer  process requires  temperature measurements with both high temporal and spatial resolution, and records that span hours to seasons.

**2 Field Site and Instrumentation**

Field observations used in this study are from three sites in western Greenland (Figure 1). Each site is named by its location with respect to the terminus of Isunnguata Sermia, a land-terminating outlet glacier. The equilibrium line altitude is at about 1500 m elevation in this area (van de Wal et al., 2012), which is 400 m above the  furthest inland site, 46-km, ~~is ~30 km below the equilibrium line altitude which is at about 1500 m elevation [van de Wal et al., 2012],~~ so all sites are well within the ablation zone and ablation rates are high (2-3 m/yr). Solar radiation in the summer creates a layer of interconnected cryoconite holes at the ice surface, and water moving through that cryoconite layer converges into surface streams. There are no large supraglacial lakes in the immediate area of any site; all streams eventually drain from the surface through moulins. A series of dark folded layers emerge to the ice surface in this region of the ice sheet (Wientjes and Oerlemans, 2010).

At each field site, boreholes for temperature instrumentation were drilled from the surface to between 20 and 21 m depth using hot-water methods. In total, seven strings of temperature sensors were installed – one at both sites 27-km and 46-km in 2011, followed by five at site 33-km between 2014 and 2016. Strings are named by the year they were installed. Each consists of between 17 and 32 sensors spaced at 0.5-3.0 m along the cable (Table 1). In 2011 and 2014, thermistors were used as temperature sensors. The thermistors have measurement resolution of 0.02°C and accuracy of about 0.5°C after accounting for drift (Humphrey et al., 2012). In subsequent years, we used a digital temperature sensor (model DS18B20 from Maxim Integrated Products, Inc.). This sensor has resolution 0.0625°C and about the same accuracy as the thermistors. To increase accuracy, each sensor was lab-calibrated in a

0°C bath, and field-calibrated with a temperature measurement during borehole freeze-in (borehole water is exactly 0°C). Because each temperature sensor is in a black casing, measurements are discarded when the sensor lies on the surface exposed to the sun.

Meteorological variables were measured at each field site as well, using standard Campbell Scientific products. In this study, we use the near-surface (~2-m) air temperature (Vaisala HMP60 with a radiation shield), the net radiative heat flux over all wavelengths shorter than 100 μm (Kipp and Zonen NR Lite), and the change in surface elevation measured with a sonic ranger distance sensor (Campbell SR50A). Data from the sonic distance sensor are filtered manually, removing any obvious outliers (more than 0.5 m from the surrounding measurements). The filtered data are then partitioned into tTwo variables are taken from the sonic ranger, cumulative ablation during the melt season and changes in snow depth during the winter. An automated weather station with all the above instrumentation was mounted on a fixed pole frozen in the ice, with segments being removed from the mounting pole each summer so the instrumentation remains close to the surface and does not extend significantly into the air temperature inversion (Miller et al., 2013). The measurement frequency for meteorological data varies from ten minutes to an hour, but all data are collapsed to a daily mean for input to a heat transfer model.

In addition to ice temperature and meteorological measurements, investigations of the subsurface were completed at site 33-km with a borehole video camera, and with a high-frequency ground-penetrating radar survey (see supplementary). These investigations were carried out in pursuit of what we think may have been subsurface fractures that are not expressed at the ice surface (described in section 5.2). With five temperature sensor strings, an automated weather station station, and the subsurface investigations, site 33-km is by far the most thoroughly studied of the three sites. For that reason, measurements from this site serve as the foundation for thea model case study presented in section 4.

**3 Results**

**3.1 Observed Ice Temperature**

Near-surface ice temperatures were measured through time in seven shallow boreholes at three different field sites (Figure 2). Although hot-water drilling methods temporarily warm ice near the instrumentation, the ice around these shallow boreholes cools to its original temperature within days to weeks. MThe measured temperatures are spatially variable between sites, with a. The mean value from the lowermost sensor (analogous to $T_0$) ofis -3.2 at 27-km, -8.6 at 46-km, and from -9.7°C to -8.1 at 33-km. In all cases, measured $T_0$ values are warmer than the mean annual air temperature. Temperature gradients are calculated by fitting a line to the mean temperature of the four lowermost sensors for each string. at 20 mThese gradients are also variable, typically typically being between -0.15 and 0.00°C/m but +0.16°C/m at the 27-km field site (positive being increasing temperature with depth below the surface). As expected, As expected, the direction ofvariability in the temperature gradients at the bottom of the profiles measured here correlate with those measured in the uppermost ~100 me observations offor full-thickness deeper temperature profiles measured at each site (Harrington et al., 2015; Hills et al., 2017).

Even the five temperature profiles measured at site 33-km exhibit some amount of spatial variability. Three

temperature strings, T-15a, b, and c, are all similar, having strong negative temperature gradients (approximately  0.1 5°C/m), and cold $T_0$ temperatures (approximately -9.5°C). Close to the surface, these three temperature strings a cold compared to the others. However, those strings stopped collecting measurements  in May 2017 and did not yield a full year of data. The missing summer period explains the strong positive temperature gradient near the surface for those three strings. T-16 is the shortest string, extending to only 9.5 m depth. This short string exhibits the  smallest range in temperatures throughout a season with the coldest surface temperatures not even reaching -15°C. In terms of mean temperature, T-16 is similar to T-14, having a small negative temperature gradient and warm temperatures in comparison to those of T-15. Based on our observations, spatial variability in near-surface ice temperature at site 33-km is controlled on the scale of hundreds of meters. Proximal observations from the three nearby T-15 strings are similar to one another, but greater variability is observed when including the more distant strings, T-14 and T-16.

Closer inspection of the measured temperature record through time reveals the transient nature of near-surface ice temperature (Figure 3). As expected, these data show a strong seasonal oscillation near the surface. During the melt season, the ice surface quickly drops as ice is warmed to the melting point. Just below the surface, the winter cold wave persists for several weeks into the summer season. For this particular string, T-14, delayed freeze-in behavior was observed in one sensor (Figure 3b) and transient heating events were observed during the melt seasons (Figure 3c, 3d, 3e). Similar heating events were observed in the T-16 string (Figure 4), but not in any other. The events range in magnitude, but in one instance ice is warmed from -10°C to -2°C in 2 hours (Figure 3c). We can only speculate on the origins of these events, and address this below in section 5.2.

**3.2 Meteorological Data**

Meteorological data from site 33-km were observed over three years (supplementary Figure S4) . Air temperatures are normally at or above the melting temperature during the summer but fall to below -30°C in winter months. The measured ablation rate is  2-3 m/yr and maximum snow accumulation is only up to 0.5 m. Net  radiation is less than zero in the winter (net outgoing because of thermal emission in the infrared wavelengths) but over 100 $W/m^2$ (daily mean) on some days in the summer.

The mean air temperature over the entire measurement period at site 33-km (-10.5°C) is cold in comparison to measured ice temperatures at that site (Figure 2; T-14, T-15, and T-16). This warm anomaly between the ice and air temperature is also observed at sites 27-km and 46-km, where ice is warmer than the measured air temperature and significantly warmer than the reference from a regional climate model (Meierbachtol et al., 2015). Interestingly, we measure almost no winter snowpack at sites 27-km and 46-km due to low precipitation and strong winds during the time period over which those data were collected (2011-2013). Our observations are thus in contradiction to the inferences made by Hooke et al. (1983) in Arctic Canada, where  the offset between air and ice temperature appeared to be primarily a result of snow insulation. Overall, the three years for which meteorological data were collected are significantly different. The 2014-15 winter was particularly cold, bringing the mean air temperature of that year more than a degree lower than the other two

seasons. Snow accumulation was approximately doubled that winter in comparison to the other two. Also, the summer melt season is longer in 2016 than in 2015. In comparison with past trends from a nearby site, KAN_L, the second year is more typical for this area (van As et al., 2012). To model a representative season, data from that second year (July 2015 to July 2016) were chosen as annual input for the model case study.

**4 Analysis**

Our objective is now to investigate how various processes active in Greenland's ablation zone influence $T_0$. In order for model results to achieve fidelity, inputs and parameters need to be representative of actual conditions. We therefore use the  meteorological dat to constrain the modeling experiments. Our modeling is focused at field site 33-km, where we have the most data for constraining the problem.

**4.1 Model Formulation**

The foundation for quantifying impacts of near-surface heat-transfer processes is a one-dimensional thermodynamic model. We argue that the processes tested here are close enough to being homogeneous that they can be adequately assessed in one dimension. The one exception is the measured heating events which are transient and spatially discrete, these are discussed in section 5.2 and are not included in the model analysis. Our model uses measured meteorological variables as the surface boundary condition and simulates ice temperature to 20 m, a depth chosen for consistency with measured data. The ice temperature at the depth of zero annual amplitude, $T_0$, is output from the bottom of the domain for each model experiment and used as a metric to compare net temperature changes between simulations. The model, its boundary conditions, and the experiments are all designed to test heat transfer processes within the ice itself. To maintain focus on ice processes, we ignore any atmospheric effects above the ice surface such as turbulent heat fluxes. The model does not, nor is it meant to, simulate the surface mass balance. We implement an Eulerian framework, treating the $z$ dimension as depth from a moving surface boundary so that emerging ice is moving through the domain and is removed when it melts at $z = 0$. We use a finite element model with a first-order linear element and 0.5-m mesh spacing refined to 2 cm near the surface. For a seamless representation of energy across the water/ice phase boundary, we implement an advection-diffusion enthalpy formulation (i.e. Aschwanden et al., 2012; Brinkerhoff and Johnson, 2013),

[revised manuscript text omitted]

is negative (wintertime) we assume that it is controlling the air temperature, so it is already accommodated in our simulation; thus, the radiative energy input is ignored in the negative case. This radiative source term, is incorporated into equation (1) at each time step, $\phi_{rad} = \frac{Q}{20cm}$, where $Q$ is the measured radiative flux at the surface in W/m². All constants for the rotten cryoconite layer are the same as that for ice.

While some models treat the absorption of radiation in snow/ice more explicitly with a spectrally-dependent Beer-Lambert Law (Brandt and Warren, 1993), we argue that it is reasonable to assume that all wavelengths are absorbed near the surface over the length scales that we consider. The only documented value that we know of for an absorption coefficient in the cryoconite layer is 28 m⁻¹ (Lliboutry, 1965) which is close to that of snow (Perovich, 2007). If the properties are truly similar to that of snow, about 90 percent of the energy is absorbed in the uppermost 20 cm (Warren, 1982). Moreover, we argue that this is precisely the reason that the cryoconite layer only extends to a limited depth; it is a result of where radiative energy causes melting.

**4.2.4 Subsurface20 m Temperature Gradient**

Finally, in the fourth model experiment we change the boundary condition at the bottom of the domain. The free boundary is changed to a Neumann boundary with a gradient of -0.05°C/m, a value that approximately matches the measured gradient in our near-surface temperature measurementsat site 33-km at field site 33-km. Importantly, this simulated gradient is in the same direction, although with a larger magnitude, of the upper ~100 m of ice in our measurements of deep temperature profiles (Hills et al., 2017). In this case, the advective energy flux is upward, but the temperature gradient is negative, bringing colder ice to the surface. In addition, two limiting cases were tested, with gradients of +/- 0.15°C/m. This is the approximate range in measured gradients (Figure 2).

**4.3 Model Results**

The control model run of simple thermal diffusion predicts that ice temperature damps to approximately the mean annual air temperature of the study year (-9.9°C) by about 15 m below the ice surface. This result is in agreement with the analytical solution (Carslaw and Jaeger, 1959), but slightly different from the mean air temperature (-9.6°C) because the air can exceed the melting temperature in the summer while the ice cannot. Other atmospheric effects such as turbulent heat fluxes and the thermal inversion could also cause a difference between measured air temperature and ice surface temperature, but these are not considered here. For each model treatment, 1-4, the incorporation of an additional physical process changes the ice temperature. Differences between model runs are compared using $T_0$ at 210 m. Again, the model experiments are progressive, so each new experiment includes the processes from all previous experiments. Key results from each experiment are as follows (Figure 56):

1. Diffusion alone results in $T_0 = -9.9°C$, whereas observed temperatures range from -9.7°C to -8.1 at the 33-km field site.
2. Because the ablation rate is strongest in the summer, the effect of incorporating ablation is to counteract the diffusion of warm summer air temperatures. The result is a net cooling of $T_0$ from experiment (1) by -0.92°C.

3. Snow on the ice surface insulates the ice from the air temperature. In the winter, snow insulation keeps the ice warmer than the cold air, but with warm air temperatures in the spring it has the opposite effect. Because snow quickly melts in the springtime, the net effect of snow insulation is substantially more warming than cooling. $T_0$ for this experiment is +0.78°C warmer than the previous.
4. Radiative energy input mainly controls melting (van den Broeke et al., 2008), but incorporating this process does warm $T_0$ by +0.52°C.
5. Imposing a -0.05°C/m 20-m temperature gradient at the bottom of the model domain, consistent with observation, dramatically changes $T_0$ by -2.5°C.

Both ablation and the subsurfacedeep temperature gradient have a cooling effect on near-surface ice temperature. On the other hand, snow and radiative energy input have a warming effect. For this case study, the first three processes together result in almost no net change so that the modeled $T_0$ is close to the observed mean air temperature (Figure 56d). However, inclusion of the subsurfacedeep temperature gradient has a strong cooling effect on the simulated temperatures, bringing $T_0$ far from the mean measured air temperature. The limiting cases show that this bottom boundary condition strongly controls the near-surface temperature, with a range in the resulting $T_0$ values from -17.0°C to -2.0°C. In summary, measured ice temperatures are consistently warmer than both the measured air temperature and simulated ice temperature (Figure 67), except in the case of a positive subsurface gradient which is discussed below.

**5 Discussion**

Our observations show that measurements of near-surface ice in the ablation zone of western Greenland arecan be significantly warmer than would be predicted by diffusive heat exchange with the atmosphere. This is in agreement with past observations collected in other ablation zones (e.g. Hooke et al., 1983). With four experiments in a numerical model that progressively incorporate more physical complexity, we are unable to precisely match independent model output to observations. Our measurement and model output consistently point toward a disconnect between air and ice temperatures in the GrIS ablation zone, with ice temperatures being consistently warmer than the air.

**5.1 Ablation-Diffusion**

The strongest result from our model case study was a drop in $T_0$ by -2.5°C associated with the imposed subsurface deep icenegative 20-m temperature gradient. While it was important to test this scenario for one case, the temperature gradient we used was representative, but somewhat arbitrary. In reality, the observed20-m temperature gradients isare widely variable from one site to another and even within one site (Figure 2). Interestingly, full ice thickness temperature profiles show similar temperature gradients, both positive and negative, that persist for many hundreds of meters toward the bed (Harrington et al., 2015; Hills et al., 2017). Hence, the limiting cases were added to show simulation results over the range of measured gradients from our temperature strings. The resulting $T_0$ span a range of 19°C.

Our model could have tested additional temperature gradients, and those of the opposite (positive) sign likely would

have fit our measured  The majority of the subsurface  temperature gradients that we measure are negative, and theoretically the gradient should be negative . Consider that fast horizontal velocities (~100 m/yr) advect cold ice from the divide to the ablation zone, and the air temperature lapse rate couples with the relatively steep surface gradients so that the surface warms rapidly toward the terminus. These conditions lead to a vertical temperature gradient below the ice surface that is negative (Hooke, 2005; pp. 131-135), as in our model example. The one exception is in the case of deep latent heating in a crevasse field (Harrington et al., 2015; sites S3 and S4) where the deep ice temperature is warmer than the mean air temperature rather than colder.

Overall, our results demonstrate that the effect of the subsurface temperature gradient is coupled to that of surface lowering. With respect to the surface, the temperature gradient below is advected upward as ice melts. There is a competition between surface lowering and diffusion of atmospheric energy into the ice; as near-surface ice is warmed, it can be removed quickly and a new boundary set. Therefore, our conceptualization of temperature in the near-surface ice of the ablation zone should not be a seasonally oscillating upper boundary with purely diffusive heat transfer (Carslaw and Jaeger, 1959), but one with advection and diffusion (Logan and Zlotnik, 1995; Paterson, 1972). This conceptualization is unique to the ablation zone because of the rapid rate of surface lowering, whereas a diffusive model for near-surface heat transfer is much more appropriate in the accumulation zone.

 The disconnect between air and ice temperature implies that that the near-surface active layer in the ablation zone is shallow (i.e. less than 15 m) and could be skewed toward the subsurface temperature gradient. Therefore, the surface boundary condition has  weaker influence on diffusion for ice well below the surface. This is in contrast to the accumulation zone where new snow is advected downward, so the surface temperature quickly influences that at depth.  Under these conditions, it is no surprise that we see spatial variability in near-surface ice temperature even within one field site. That variability is simply an expression of the deeper ice temperature variations which are hypothesized to exist from variations in vertical advection (Hills et al., 2017), and do not have time to completely diffuse away before they are exposed at the surface.

**5.2 Subsurface Refreezing**

In two temperature strings we observe  heating events, the largest case being 8°C in 2 hours between 3 and 8 m below the ice surface (Figure 3c). These events are transient, they are spatially discrete, and they are generated at depth, all of which are most easily explained by.  the refreezing of liquid water in cold ice. Similar refreezing events have been observed in firn (Humphrey et al., 2012), where they are not only important for ice temperature but could also imply a large storage reservoir for surface meltwater (Harper et al.,

2012). However, unlike firn, solid ice is impermeable  unless fractures are present .

In Greenland's ablation zone, prior work has demonstrated the importance of  large-scale latent heating in open crevasses (Phillips et al., 2013; Poinar et al., 2016). Additionally, While ater-filled cavities have been observed in cold, near-surface ice on  a mountain glaciers (Jarvis and Clarke, 1974; Paterson and Savage, 1970). In our case, however, an explanation for refreezing water is not obvious: while the field site has occasional mm-aperture 'hairline' cracks, there are no visible open crevasses at the surface for routing water to depth.

As far as we know, this work is the first to report evidence of short-term transient latent heating events in cold ice, not obviously linked to open surface fractures.

While the hairline  Fractures could perhaps move some water ,  to permit much water to move meters through cold ice they would need to be large enough that water moves quickly, and does not instantaneously refreeze. For example,  a 1-mm wide crack in ice that is -10°C freezes shut in about 45 seconds (Alley et al., 2005; eq. 8). That amount of time could be long enough for small volumes of water to move  5-10 m below the surface, but would require a hydropotential gradient to drive water flow. Thus, top-down hairline crevassing does not seem a plausible explanation for the events we observe.

Importantly, several independent field observations in this area including hole drainage of water during hot-water drilling, ground-penetrating radar reflections, and borehole video observations, all point to the existence of subsurface air-filled and open fractures with apertures of up to a few cm (see supplement).  That they are open at depth, but are narrow or non-existent at the surface, could be linked to the colder ice at depth and its stiffer rheology. Nath and Vaughan (2003) observed similar subsurface crevasses in firn, although in their case density controls the stiffness rather than temperature.

On rare occasions, we argue that the aperture of the fractures open wider to the surface, where there is copious water stored in the cryoconite layer (Cooper et al., 2018) that can drain and refreeze at depth . While

the events seem to happen in the springtime and it would be tempting to assert that fracture opening coincides with speedup, our measurements of surface velocity at these sites show that this is not always the case. This may be due to that fact that the spring speed up coincides with early melt rather than peak melt and copious water in the cryoconite layer.

Latent heating in the form of these subsurface refreezing events is an obvious candidate for a source for the 'extra' heat  we observe in our temperature strings relative to simulations. Our data show that refreezing in subsurface fractures has the potential to warm ice substantially over short periods of time, and apparently this can occur in places where crevasses are not readily observed at the surface. Furthermore, the difference between measured and modeled temperatures (~3°C) is the equivalent of only ~1.7% water by volume. Our simplified one-dimensional model would not be well-suited to assess the influence of these latent heating events. Instead, we provide a simple calculation for energy input from the events by differencing the temperature profiles in time and integrating for total energy density (Figure 7 a-c),

$$\phi_{measured} = \frac{\rho_i C_p}{\Delta z} \int \Delta T \, dz \tag{9}$$

where $\Delta z$ is the total depth of the profile, and $\Delta T$ is the differenced temperature profile. Only sensors that are below the ice surface for the entire time period are considered. To calculate the total water content refrozen in the associated event, we remove the conductive energy fluxes from the total energy density calculated above. We do so by calculating the temperature gradients at the top and bottom of the measured temperature profile at each time step as in Cox et al. ( 2015).

$$\phi_{conductive} = \frac{-k_i}{\Delta z} \int \frac{\partial T}{\partial z}\bigg|_{top} - \frac{\partial T}{\partial z}\bigg|_{bottom} \, dt \tag{10}$$

The resulting energy sources are then converted to a volume fraction of water by

$$\omega_{measured} = \frac{\phi_{measured} - \phi_{conductive}}{\rho_w L_f} \tag{11}$$

where $\rho_w$ is the density of water. Results show that each year some fractions of a percent of water are refrozen (Figure 7 d-f). Through several seasons that amount of refreezing could easily add up to the ~3°C anomaly that we observe.

Unfortunately, without a more thorough investigation, we do not have enough evidence to show that these refreezing events are more than a local anomaly. Of our seven near-surface temperature strings, only T-14 and T-16 demonstrated refreezing events, and so we are not confident that they are temporally or spatially ubiquitous.

The only other logical Other possibilitmechanismies for the warm offset between measurements and model resultsbias would be warming from below through a positive subsurface temperature gradient. While it is tempting to associate deep warm ice with either residual heat from the exceptionally hot summers ofa strong warm event in previous years such as in 2010 and 2012 (Tedesco et al., 2013), this scenario is unlikely because the ablation rates are so high that any ice warmed during those years has likely already melted. Deeper latent heating from an upstream crevasse field is a more plausible alternative(van As et al., 2012), or possibly deeper latent heating from an upstream crevasse field. In those cases, a positive 20-m deep temperature gradient would promote warming near the surface; however, in this area full-depth temperature profiles do not show deeper ice to be anomalously warmed except in one localized case (Hills et al., 2017).

**6 Conclusion**

We observe the temperature of ice at the depth of zero annual amplitude, $T_0$, in Greenland's ablation zone to be markedly warmer than the mean annual air temperature. These findings contradict predictions from purely diffusive heat transport but are not surprising considering the processes which impact heat transfer in ice of the ablation zone. High ablation rates in this area indicate that ice temperatures below ~15 m reflect the temperature of deep ice that is emerging to the surface, confirming that the ice does not have time to equilibrate with the atmosphere. In other words, ice flow brings cold ice to the surface at a faster rate than heat from the atmosphere can diffuse into the ablating surface. The coupling between rapid ablation and the spatial variability in deep ice temperature implies there will always be a disconnect between air and ice temperatures. Additionally, we observe infrequent refreezing events below 5-10 m of cold ice. Meltwater is likely moving to that depth through subsurface fractures that are not obviously visible at the surface.

In analyzing a series of processes that control near-surface ice temperature, we find that some lead to colder ice, and others to warmer, but most are strong enough to dramatically alter the ice temperature from the purely diffusive case. With rapid ablation, a spatially variable temperature field, and subsurface refreezing events, $T_0$ in the ablation zone should not be expected to match the air temperature. That our measurements are consistently warmer, could simply be due to the limited number of observations we have, but latent heat additions are clearly measured and could be common in near-surface ice of the western Greenland ablation zone.

[Figure]

**Figure 1: A site map from southwest Greenland with field sites (red) named by their location with respect to the outlet terminus of Isunnguata Sermia. The inset shows locations of near-surface temperature strings (black) named by the year they were installed and an automated weather station  (blue). Surface elevation contours are shown at 200-m spacing (Howat et al., 2014).**

[Figure]

**Figure 2: Near-surface ice temperature measurements from seven strings: T-11a, T-11b, T-14, T-15a, T-15b, T-15c, T-14, and T-16. For each, the shaded region shows the range of measured temperatures over the entire measurement period, and the red dotsolid lines indicates a the mean value for each sensortemperature profile. Depths are plotted with respect to the surface at the time of measurement, so sensor locations move toward the surface as ice melts. Strings with less than 11 months of data are slightly more transparent. For field sites at which the air temperature was measured for at least a full year, a dashed line shows the mean air temperature.**

[Figure]

**Figure 3: Three years of ice temperature measurements from the T-14 string. While this string was initially installed to 2 m depth,** **measurements are plotted with reference to the moving surface so the sensors move up throughout the time period, revealing a gray mask. Transient features in the data include anomalously slow freeze-in behavior in one sensor (b) as well as heating events throughout the collection time period (c, d, and e). The heating events are plotted as a series of temperature profiles with the darker shades being later times and time steps between profiles of 2 hours (c), 10 hours (d), and 1 hour (e).**

[Figure]

**Figure 4: Heating events**  from temperature string T-16. Profiles are plotted as in Figure 3 c, d, and e.  **The time steps between profiles are** **2** hours s and 4 hours (b).

[Figure]

Figure 5: Meteorological data from 33-km over three years including a) air temperature, b) ice surface location, c) snow depth, and d) net shortwave radiation. All data are plotted as a daily mean. The shaded region encloses the time period that is used for the model case study.

**Figure 56**: Model results for five separate simulations. In each case, twelve simulated temperature profiles are shown from throughout the yearlong period, and control results (from (a)) are displayed for comparison (gray). Differences between the simulations are analyzed quantitatively using $T_0$, the convergent temperature at 21 m. Processes are from top to bottom: a) control  simulation of pure diffusion, b) ablation, c) snow insulation, d) radiative energy input, and finally e) subsurface temperature gradient. The two limiting cases for the subsurface temperature gradient are plotted with dashed gray lines (e).

[Figure]

**Figure 67**: A comparison of model output (gray) and data from 33-km, including mean ice temperatures (red) and mean annual air temperatures for three seasons (black dashed). Theobserved ice temperatures are plotted the same as in Figure 2.  Note that three of the temperature strings collected only ~9 months of data (transparent red). Mean temperatures from those three strings are  cold near the surface because they collected more wintertime measurements than summertime.

[Figure]

**Figure 7:  Energy source for the observed heating events. a-c) Observed energy density through time for the differenced temperature profile calculated with equation (9) (black), and conductive energy density through time calculated with equation (10) (red). d-f) Percent by volume water refrozen for the associated source in (a-c). This value is proportional to the difference between the black and red lines above.** The temperature string from which measurements were taken is labeled at the top.

**Table 1: Temperature Strings**

| String Name | Data Time Period | Time Step (hr) | Sensor | # of Sensors | Sensor Spacing (m) | Latitude | Longitude | Elevation (m) |
|---|---|---|---|---|---|---|---|---|
| T-11a | 7/5/11 – 7/15/13 | 3 | Thermistor | 32 | 0.6 | 67.195175 | -49.719515 | 848 |
| T-11b | 7/11/11 – 12/17/11 | 3 | Thermistor | 32 | 0.6 | 67.201553 | -49.289058 | 1095 |
| T-14 | 7/18/14 – 8/14/16 | 0.5 | Thermistor | 31 | < 11 m deep – 0.5
> 11 m deep – 1.0 | 67.18127 | -49.56982 | 956 |
| T-15a | 08/17/16 | N/A | DS18B20 | 17 | < 15 m deep – 1.0
> 15 m deep – 3.0 | 67.18211 | -49.568272 | 954 |
| T-15b | 08/17/16 | N/A | DS18B20 | 17 | < 15 m deep – 1.0
> 15 m deep – 3.0 | 67.182054 | -49.568059 | 954 |
| T-15c | 08/17/16 | N/A | DS18B20 | 17 | < 15 m deep – 1.0
> 15 m deep – 3.0 | 67.182114 | -49.568484 | 954 |
| T-16 | 08/17/16 | N/A | DS18B20 | 18 | 0.5 | 67.18147 | -49.57025 | 951 |

**Table 2: Constants**

| Variable | Symbol | Value | Units | Reference |
|---|---|---|---|---|
| Reference Enthalpy | $H_m$ | 0 | $J\ kg^{-1}$ | |
| Ice Density | $\rho_i$ | 917 | $kg\ m^{-3}$ | Cuffey and Paterson (2010) |
| Snow Density | $\rho_s$ | 300 | $kg\ m^{-3}$ | |
| Water Density | $\rho_w$ | 1000 | $kg\ m^{-3}$ | |
| Specific Heat Capacity | $C_p$ | 2097 | $J\ kg^{-1}\ K^{-1}$ | Cuffey and Paterson (2010) |
| Latent Heat of Fusion | $L_f$ | $3.335*10^5$ | $J\ kg^{-1}$ | Cuffey and Paterson (2010) |
| Thermal Conductivity of Ice | $k_i$ | 2.1 | $J\ m^{-1}\ K^{-1}\ s^{-1}$ | Cuffey and Paterson (2010) |
| Thermal Conductivity of Snow | $k_s$ | 0.2 | $J\ m^{-1}\ K^{-1}\ s^{-1}$ | Calonne et al. (2011) |
| Moisture Diffusivity | $v$ | $1*10^{-4}$ | $kg\ m^{-1}\ s^{-1}$ | Aschwanden et al. (2012) |

---

## Author Comment (AC2) · 30 Jul 2018

Comments and associated responses for:

**Processes influencing heat transfer in the near-surface ice of Greenland's ablation zone**

Benjamin H. Hills et al. 2018

**Anonymous Referee #2**

Author responses are in blue.

Hills et al. Investigate heat transfer from the atmosphere to the ice in west Greenland's ablation zone and conclude based that air temperature can not predict the near-surface ice temperature. While the topic is interesting and the data presented is valuable, the modelling part does not lead to strong conclusions. I would recommend to rework the paper, focus on the data analysis, especially the very interesting transient heating events, ideally deriving quantitative conclusions on the amount of water necessary to reproduce them and modify the modelling part significantly. Some assumptions for the model-part seem inappropriate or at least too weakly constrained in order to judge if the derived conclusions are valid. Comparison with Promice stations in the area may improve the applicability and validate some of the factors. Central is an in-depth check of the boundary (6) for the modelling part. This should be discussed thoroughly. Take a promice station in the area, convert outgoing longwave radiation with Stefan Boltzmann to surface temperature and plot vs air temperature. This plot is necessary for the paper and will show if (6) is OK to use at all. Also, put more effort into explaining the massive local-scale variability of ice temperatures which is surprising to me. Generally, I would suggest major revisions, a change in focus of the paper and/or a substantial improvement of the modelling part.

Thank you for these comments. We appreciate the requested change of focus from the model to the data, and have made a substantial effort to carry out this request. We have removed the emphasis on meteorological data, as both reviewers seemed to get lost in those details and our intention was always to focus on ice temperature and not meteorological processes. We have added explanations of the transient features in the data, and, as requested, have added an analysis of the subsurface refreezing events (Figure 7).

We appreciate that the modeling results are somewhat inconsistent with the data, but we would argue that this is in and of itself a worthwhile conclusion. The processes that we investigate do not account for the measured temperature discrepancy, so some other process not modeled must be important. We have added language to the discussion, emphasizing the importance of the subsurface refreezing events. We state that the measured temperature discrepancy must be either a result of those heating events, or a result of the subsurface temperature gradient being positive because of a deeper energy source.

Finally, to address some of the concerns with the meteorological data, we have added to the supplement. We compare our measurements to those of the KAN_L PROMICE station (Figure S4). We also add a surface ice temperature calculated with the Stefan-Boltzmann Law.

P2 L12: add reference?

Added Cuffey and Paterson (2010).

L19 – if it is often used, add more references. This is potentially an important issue.

Added Mock and Weeks (1966). These are the two most seminal studies (one from Greenland and the other from Antarctica) of this time period around the IGY when this method was used most. We argue that more references would be redundant.

P3 L1: this statement is true for all ablation areas and is not GrIS specific!?

Changed to "the processes which make the ablation zone different from other areas of a glacier or ice sheet"

L7-8. Can observation resolve these processes quantitatively? In my view this is the 'issue' with observations, that you end up with a 'bulk' signal combining different processes. Consider rewording

Changed paragraph:

"Our near-surface temperature observations represent an aggregated sum of the processes mentioned above. A numerical model can be used to partition the relative importance of those processes, but only with measurements in hand as validation. Therefore, confidently constraining the role of near-surface heat transfer processes requires temperature measurements with both high temporal and spatial resolution, and records that span hours to seasons."

L13. 30 km below. . . is confusing. Rewrite. Also 1500m elevation in this area?

van de Wal et al. (2012) show the ELA fluctuating around 1500 m over the last 20 years. This was confusingly worded in our case though.

Changed sentence:

"The equilibrium line altitude is at about 1500 m elevation in this area (*van de Wal et al.*, 2012), which is 400 m above the furthest inland site, 46-km, so all sites are well within the ablation zone and ablation rates are high (2-3 m/yr)."

L13: I miss info on elevation of the sites. Please add to Tab1

Added elevations.

L24: reduce numbers of sign digits

This is the number given on the datasheet for the sensor (DS18B20 from Maxim Integrated Products, Inc.)

L27: how was the field calibration done?

Changed to:

"field-calibrated with a temperature measurement during freeze-in (borehole water is exactly 0°C)."

L29: near surface: how near? Did you have a radiation shield? Add a picture of the AWS

Added:

"(~2-m) air temperature (Vaisala HMP60 with a radiation shield)"

Photo of the AWS was added to the supplement (Figure S2).

L34: how often were they re-aligned? Estimating from fig 3 there was a ca 6 m surface elevation change. This pushes your air temperature quite far into the near-surface inversion. How do you account for this.

Added: "with segments being removed from the mounting pole each summer so that the instrumentation stays close to the surface."

P4 L4: why five strings?

We are trying to asses intra-site variability in near-surface ice temperature. This is how many strings were installed at site 33-km.

L11: how does that fit with field calibration P3 L27?

This was addressed in the comment above. The borehole water should be exactly 0°C.

L15: explain what positive and negative means for the gradients

Added "(positive being increasing temperature with depth below the surface)"

L15-17: unclear sentence. What do you mean?

Changed to:

"As expected, the temperature gradients measured here correlate well with the temperature gradients in the uppermost ~100 m for full-thickness temperature profiles measured at each site (Harrington et al., 2015; Hills et al., 2017)."

L21: makes things difficult. Consider only showing averages for concurrent times or at least full years? The rest does not make sense to me.

While we agree that the interpretation is more difficult for these sensor strings, we find it important to present all of the data. The three strings that failed in the springtime still provide valuable measurements especially below ~10 m where the seasonal variations are small.

For clarification we changed Figure 2 so that the failed strings are slightly transparent, and the caption is changed accordingly.

L26-27: this is a very large near-surface variability and not easy to understand. How about radiation errors of the uppermost sensors?

Added:

"Because each temperature sensor is in a black casing, measurements are discarded when the sensor lies on the surface exposed to the sun."

L29: refer here to something you mark in the fig.

Figure 3 was significantly changed to point out the transient events.

P5 L5 see above. NOT net-shortwave

Deleted "shortwave"

L6: sounds low to me. Compare to Promice KAN stations? This could also help discriminating into SW and LW.

This was simply an error in the language. As was noticed, we repeatedly mentioned 'shortwave' radiation because of a previous draft of the manuscript, whereas we are now using net radiation. The numbers are more appropriate for mean daily net radiation.

L9: consider rewording 'warm bias'

Changed to: "This warm anomaly between the ice and air temperature is also observed…"

L15-20: I believe that winter 2016 was particularly snow-poor and thus not particularly 'representative'. Check if that is true on regional scale and include in discussion.

Looking at the data from PROMICE, KAN_L, it seems like this is in fact a representative snow year. 2015 was a much bigger snow year as can be seen in both our data and the PROMICE data.

Part 4 and 4.1. See issue with boundary condition (6). Boundary condition (7) is clearly not valid as yu state yourself and adapt later. But why then introducing it and calling it a boundary condition? This does not make too much sense to me. The choice of experiments seems arbitrary. How about turbulent heat exchange?

Boundary condition (7) is altered as one of the experiments. For this reason, an insulating boundary (7) was used for the first 4 simulations with the specific intention of isolating the processes represented in each of those 4. When a heat flux is finally introduced in 4.2.4 we see the strong effect that it has on the near-surface temperature. It is in these contrasting model results that we see the significance of that bottom temperature gradient.

We consider only processes within the snow/ice (not the air above). Hence, the inversion and turbulent heat exchange are ignored.

P8 L9: net radiation

L25: -0.05∘C/m,

P9 L5: also in other occasions surface temperature and air temperature can be very different

Added:

"Other atmospheric effects such as turbulent heat fluxes and the thermal inversion could also cause a difference between measured air temperature and ice surface temperature, but these are not considered here."

L20: and

We are not sure what is being asked for here, but changed the sentence for clarity:

"Imposing a -0.05°C/m temperature gradient at the bottom of the model domain, consistent with observation, dramatically changes $T_0$ by -2.5°C."

P10 2-4this seems trivial. If you keep the upper part of the 'trumpet' as it is and induce a gradient in 20m you will end up colder there. Is there a value in it?

We argue that our results show how important the subsurface temperature gradient is in controlling near-surface heat transfer and therefore melting in the ablation area. See P10 L26-33.

L24-25. What do you mean? Unique to the ablation zone. . ..larger than other areas? Other ablation areas? The sentence does not make sense to me

Changed to:

"This conceptualization is unique to the ablation zone because of the rapid rate of surface lowering, whereas a diffusive model for near-surface heat transfer is much more appropriate for the accumulation zone."

L30-34. Interesting. But what is the reason for such different ice-packages coming to the same site just a few dozens of m away? This should be better discussed and this high variability is potentially a significant result of the paper. How do satellite-derived surface temperatures vary spatially?

This process is discussed in detail in a previous paper that is referenced in the manuscript (Hills et al. 2017).

P11: L6. Refreezing is a big topic these days. Consider adding more refs Could the amount of water be estimated that would be necessary in order to reproduce the warming caused by latent heat you observe? This would be interesting.

Added:

"However, unlike firn, solid ice is impermeable to water unless fractures are present (Fountain et al., 2005)."

"In Greenland's ablation zone, much work is being done to assess large-scale latent heating in open crevasses (Phillips et al., 2013; Poinar et al., 2016). Additionally, water-filled cavities have been observed in cold, near-surface ice on a mountain glacier (Jarvis and Clarke, 1974; Paterson and Savage, 1970). However, we are the first to show evidence of short-term transient latent heating events in cold ice."

L26-29. Check out Colgan et al. Crevasse review. There is a process mentioned of crevasses 'growing' from below to the surface.

Added:

"Nath and Vaughan (2003) observed similar subsurface crevasses in firn, although in their case density was controlling the stiffness rather than temperature."

P5-9: all that would point to ice being colder than air. But you show in fig 2 it is warmer. I doubt that the modelling serves as a base for this conclusion.

We added more analysis to the discussion to argue that the warm ice could be the result of two processes: the observed subsurface refreezing events, or a positive subsurface temperature gradient.

Fig 2 has a problem as it does not seem to account for changing surface height – I deduce this from the fact that each dot represents a sensor. It is, however, over time in different depths. Especially for the uppermost sensors this creates an issue. Suggest to correct the time-series using the known surface elevation changes. For the mean annual air temperatures. How do promice stations in the area fit to that? I am a bit worried about the height-above terrain and the radiation shield issue. Are air temperatures ventilated? Also: why don't you sort them logically, i.e. from 27, 33 to 46? Add axes descriptions. Label a-g

We changed this figure so that temperatures are plotted at the time of measurement rather than the time of installation. We also added labels a-g and sorted based on time of installation.

We added a figure to the supplement which compares our AWS data to a nearby PROMICE station (Figure S4). Language was also added to the manuscript to address the radiation shield and height above surface:

"(Vaisala HMP60 with a radiation shield)"

"with segments being removed from the mounting pole each summer so that the instrumentation remains close enough to the surface that we are measuring ice surface conditions"

Figure 3: misleading. The depth of the sensor during installation is shown and not sensor depth. This is a massive difference. Strongly recommend to correct it for that. There are some interesting features and it is impossible to tell whether these are artefacts or reality. Which ones are the warming events you refer to? how about the vertical red lines, i.e. end of 2015 or ca may 2016 further down or again some time in spring 2017. Suggest indicating the

warming events you refer to later in the discussion. And what with the horizontal redish bodies in late 2014 for instance in ca 8 m?

We made significant changes to many of the figures including Figure 3. We hope that we have addressed some of the confusion regarding identification of the events, as well as the other transient features in the dataset; namely, the freeze-in behavior (Figure 3b) which is now addressed in the manuscript as well.

Fig. 4: do they cool off again afterwards? Why don't you show the same time-steps in c and d as you do in a and b? how does meteorology play in here? Was there a rain-event preceding this? Impossible to tell if you don't state when it happened.

We feel that the way in which we have changed Figure 3 and the addition of the new Figure 7 help to illustrate when the refreezing events happen.

Fig. 5. b) consider combining b and c as surface elevation change D)If you measured with a NRlite p3, L30, you don't get net shortwave radiation to my understanding.

This figure was moved to the supplement.

We intentionally split (b) and (c) from the measured surface elevation change so that the steps of the modeling exercise are more clear.

**Processes influencing  heat transfer in the **near-surface ice of** Greenland's ablation zone**

Benjamin H. Hills[1,2], Joel T. Harper[2], Toby W. Meierbachtol[2], Jesse V. Johnson[3], Neil F. Humphrey[4], Patrick J. Wright[5,2]

[1]Department of Earth and Space Sciences, University of Washington, Seattle, Washington, USA
[2]Department of Geosciences, University of Montana, Missoula, Montana, USA
[3]Department of Computer Science, University of Montana, Missoula, Montana, USA
[4]Department of Geology and Geophysics, University of Wyoming, Laramie, Wyoming, USA
[5]Inversion Labs LLC, Wilson, Wyoming, USA

*Correspondence to*: Benjamin H. Hills (bhills@uw.edu)

**Abstract.** To assess the influence of various  heat transfer mechanisms on the temperature structure of ice near the surface  of Greenland's ablation zone, we  compare highly resolved in situ measurements  with simplified thermal modeling experiments. Seven separate temperature strings were installed at three different field sites, each with between 17 and 32 sensors and extending up to 21 meters below the surface. In one string, temperatures were measured every 30 minutes, and the record is continuous for more than three years. We use these measured ice temperatures to constrain modeling analyses focused on four isolated processes and assess the relative importance of each for the near-surface ice temperature: 1) the moving boundary of an ablating surface, 2) thermal insulation by snow, 3) radiative energy input, and 4) subsurface ice temperature gradients below the seasonally active near-surface layer. In addition to these four processes, transient heating events were observed in two of the temperature strings. Despite no observations of meltwater pathways to the subsurface, these heating events are likely the refreezing of liquid water below 5-10 meters of cold ice. Together with subsurface refreezing, the five heat transfer mechanisms presented here account for measured differences of up to 3°C between the mean annual air temperature and the ice temperature at the depth where annual temperature variability is dissipated.  Thus, in Greenland's ablation zone, the mean annual air temperature is not a reliable predict of the near-surface ice temperature, as is commonly assumed.

**1 Introduction**

~~Ice sheets are coupled to the atmosphere at their upper surface through an exchange of mass and energy. Understanding this coupling is important for knowing the ice sheet's surface mass balance and its associated contribution to sea level rise. In particular, the Greenland Ice Sheet (GrIS) has shown a change toward a more negative surface mass balance, which constitutes at least half of its contribution to recent sea level rise (van den Broeke et al., 2009; Enderlin et al., 2014). Inhigh melt,(Wharton et al., 1985)M. P.climate systems~~, we must understand the processes that control near-surface heat transfer in bare ice.

Heat transfer at the ice surface is dominated by thermal diffusion from the overlying air (Cuffey and Paterson, 2010). Seasonal air temperature oscillations are diminished with depth into the ice, until they are negligible (i.e. ~1%) at a 'depth of zero annual amplitude' (van Everdingen, 1998). The exact location of this depth is dependent on the thermal diffusivity of the material through which heat is conducted as well as the period of oscillations (Carslaw and Jaeger, 1959; pp. 64-70). In theory, the temperature at the depth of zero annual amplitude, a value we will call $T_0$, is approximately constant and equal to the mean annual air temperature. In snow and ice, the depth of zero annual amplitude is approximately 10 and 15 m respectively (Hooke, 1976). For this reason, studies in the cryosphere often use $T_0$ as a proxy for the mean air temperature, drilling to 10 or more meters to measure the snow or ice temperature at that depth (Loewe, 1970; Mock and Weeks, 1966).

In places where heat transfer is purely diffusive, the snow or ice is homogeneous, and interannual climate variations are minimal, $T_0$ is a good approximation for the mean air temperature. However, prior studies have shown that, in many areas of glaciers and ice sheets, the relationship between air and ice temperatures can be substantially altered by additional heat transfer processes. For example, in the percolation zone, infiltration and refreezing of surface meltwater warm the subsurface (Humphrey et al., 2012; Müller, 1976). Studies have also revealed ice anomalously warmed by 5°C or more in the ablation zone (Hooke et al., 1983; Meierbachtol et al., 2015), but the mechanisms for this are unclear.

Hooke et al. (1983) explored the impacts of several heat transfer processes within near-surface ice at Storglaciären and the Barnes Ice Cap. They focused on the wintertime snowpack which acts as insulation to cold air temperatures but is permeable to meltwater percolation. Their results showed that the average ice temperature at and below the equilibrium line of those glaciers tends to be higher than the mean annual air temperature. They attributed the observed difference mainly to snow insulation because the strength of their measured offset was correlated to the thickness of the snowpack.

In this study, we expand the analysis of Hooke et al. (1983) and turn focus to the GrIS ablation zone with nearsurface temperature profiles from seven locations. We use our temperature measurements in conjunction with a one-dimensional heat transfer model to assess heat transfer processes in this area. The processes which make the ablation zone different from other areas of a glacier or ice sheet are, first, that the ice surface spends much of the summer period pinned at the melting point, despite slightly warmer air temperatures. Next, high ablation rates counter emerging ice flow, removing the ice surface and exposing deeper ice, along with its heat content, to the surface. The contrast of a wintertime snowpack to bare ice in the summer enables an insulat effect during the winter months, and the  deep penetration of solar radiation into bare ice results in subsurface heating and melting (Brandt and Warren, 1993; Liston et al., 1999) . Finally, surface melt can move through open fractures, carrying latent heat with it to deeper and colder ice, and upon refreezing, the meltwater warms that ice below the surface (Jarvis and Clarke, 1974; Phillips et al., 2010).

Our

near-surface temperature observations represent an aggregated sum of the processes mentioned above. A numerical model can be used to partition the relative importance of those processes, but only with measurements in hand as validation. Therefore, confidently constraining the role of near-surface heat transfer process requires temperature measurements with both high temporal and sp resolution, and records that span hours to seasons.

**2 Field Site and Instrumentation**

Field observations used in this study are from three sites in western Greenland (Figure 1). Each site is named by its location with respect to the terminus of Isunnguata Sermia, a land-terminating outlet glacier. The equilibrium line altitude is at about 1500 m elevation in this area (van de Wal et al., 2012), which is 400 m above the furthest inland site, 46-km, ~~is ~ 30 km below the equilibrium line altitude which is at about 1500 m elevation [van de Wal et al., 2012], ~~so all sites are well within the ablation zone and ablation rates are high (2-3 m/yr). Solar radiation in the summer creates a layer of interconnected cryoconite holes at the ice surface, and water moving through that cryoconite layer converges into surface streams. There are no large supraglacial lakes in the immediate area of any site; all streams eventually drain from the surface through moulins. A series of dark folded layers emerge to the ice surface in this region of the ice sheet (Wientjes and Oerlemans, 2010).

At each field site, boreholes for temperature instrumentation were drilled from the surface to between 20 and 21 m depth using hot-water methods. In total, seven strings of temperature sensors were installed – one at both sites 27-km and 46-km in 2011, followed by five at site 33-km between 2014 and 2016. Strings are named by the year they were installed. Each consists of between 17 and 32 sensors spaced at 0.5-3.0 m along the cable (Table 1). In 2011 and 2014, thermistors were used as temperature sensors. The thermistors have measurement resolution of 0.02°C and accuracy of about 0.5°C after accounting for drift (Humphrey et al., 2012). In subsequent years, we used a digital temperature sensor (model DS18B20 from Maxim Integrated Products, Inc.). This sensor has resolution 0.0625°C and about the same accuracy as the thermistors. To increase accuracy, each sensor was lab-calibrated in a

0°C bath, and field-calibrated with a temperature measurement during borehole freeze-in (borehole water is exactly 0°C). Because each temperature sensor is in a black casing, measurements are discarded when the sensor lies on the surface exposed to the sun.

Meteorological variables were measured at each field site as well, using standard Campbell Scientific products. In this study, we use the near-surface (~2-m) air temperature (Vaisala HMP60 with a radiation shield), the net radiative heat flux over all wavelengths shorter than 100 μm (Kipp and Zonen NR Lite), and the change in surface elevation measured with a sonic ranger distance sensor (Campbell SR50A). Data from the sonic distance sensor are filtered manually, removing any obvious outliers (more than 0.5 m from the surrounding measurements). The filtered data are then partitioned into tTwo variables are taken from the sonic ranger, cumulative ablation during the melt season and changes in snow depth during the winter. An automated weather station with all the above instrumentation was mounted on a fixed pole frozen in the ice, with segments being removed from the mounting pole each summer so the instrumentation remains close to the surface and does not extend significantly into the air temperature inversion (Miller et al., 2013). The measurement frequency for meteorological data varies from ten minutes to an hour, but all data are collapsed to a daily mean for input to a heat transfer model.

In addition to ice temperature and meteorological measurements, investigations of the subsurface were completed at site 33-km with a borehole video camera, and with a high-frequency ground-penetrating radar survey (see supplementary). These investigations were carried out in pursuit of what we think may have been subsurface fractures that are not expressed at the ice surface (described in section 5.2). With five temperature sensor strings, an automated weather station station, and the subsurface investigations, site 33-km is by far the most thoroughly studied of the three sites. For that reason, measurements from this site serve as the foundation for thea model case study presented in section 4.

**3 Results**

**3.1 Observed Ice Temperature**

Near-surface ice temperatures were measured through time in seven shallow boreholes at three different field sites (Figure 2). Although hot-water drilling methods temporarily warm ice near the instrumentation, the ice around these shallow boreholes cools to its original temperature within days to weeks. MThe measured temperatures are spatially variable between sites, with a. The mean value from the lowermost sensor (analogous to $T_0$) ofis -3.2 at 27-km, -8.6 at 46-km, and from -9.7°C to -8.1 at 33-km. In all cases, measured $T_0$ values are warmer than the mean annual air temperature. Temperature gradients are calculated by fitting a line to the mean temperature of the four lowermost sensors for each string. at 20 mThese gradients are also variable, typically typically being between -0.15 and 0.00°C/m but +0.16°C/m at the 27-km field site (positive being increasing temperature with depth below the surface). As expected, As expected, the direction ofvariability in the temperature gradients at the bottom of the profiles measured here correlate with those measured in the uppermost ~100 me observations offor full-thickness deeper temperature profiles measured at each site (Harrington et al., 2015; Hills et al., 2017).

Even the five temperature profiles measured at site 33-km exhibit some amount of spatial variability. Three

temperature strings, T-15a, b, and c, are all similar, having strong negative temperature gradients (approximately (ranging from -_0.1 to -0.045°C/m), and cold $T_0$ temperatures (approximately -9.65°C). Close to the surface, these three temperature strings areppear to be rather cold compared to the others. However, those strings stopped collecting measurements failed in May 2017 and did not yield a full year of data. The missing summer period explains the mean cold biasstrong positive temperature gradient near the surface for those three strings. T-16 is the shortest string, only string that did not reach 20 m. This short string extendextendinged to only 9.5 m depth. ;This short string exhibits the and measured the smallest range in temperatures throughout a season with the coldest surface temperatures not even reaching -15°C. In terms of mean temperature, T-16 is similar to T-14, having a small negative temperature gradient and warm temperatures in comparison to those of T-15. Based on ourthese observations, spatial variability in near-surface ice temperature at site 33-km is controlled on the scale of hundreds of meters. Proximal observations from the three nearby T-15 strings are similar to one another, but greater variability is observed when including the more distant strings, T-14 and T-16.

Closer inspection of the measured temperature record through time reveals the transient nature of near-surface ice temperature (Figure 3). As expected, these data show a strong seasonal oscillation near the surface. During the melt season, the ice surface quickly drops after near-surfaceas ice is warmed to the melting point. Just below the surface, the winter cold wave persists for several weeks into the summer season. For this particular string, T-14, delayed freeze-in behavior was observed in one sensor (Figure 3b) and transient heating events were observed during the melt seasons (Figure 3c, 3d, 3e). Similar heating events were observed in the T-16 string (Figure 4), but not in any other. The events range in magnitude, but in one instance ice is warmed from -10°C to -2°C in 2 hours (Figure 3c). We can only speculate on the origins of these events, and address this below in section 5.2.

**3.2 Meteorological Data**

Meteorological data from site 33-km were observed over three years (supplementary Figure S4) (Figure 5). Air temperatures are normally at or above the melting temperature during the summer but fall to below -30°C in winter months. The measured ablation rate is on the order of 2-3 m/yr and maximum snow accumulation is only up to 0.5 m. Net shortwave radiation is less than zero in the winter (net outgoing because of thermal emission in the infrared wavelengths) but over 100 W/m$^2$ (daily mean) on some days in the summer.

The mean air temperature over the entire measurement period at site 33-km (-10.5°C) is cold in comparison to measured ice temperatures at that site (Figure 2; T-14, T-15, and T-16). This warm anomaly between the ice and air temperaturebias in the near-surface ice temperature is also observed at sites 27-km and 46-km, where ice is warmer than the measured air temperature and significantly warmer than the reference from a regional climate model (Meierbachtol et al., 2015). Interestingly, we measure almost no winter snowpack at sites 27-km and 46-km due to low precipitation and strong winds during the time period over which those data were collected (2011-2013). Our observations are thus in contradiction to the inferences made by Hooke et al. (1983) in Arctic Canada, wherewho said that the offset between air and ice temperature appeared to beis primarily a result of snow insulation. Overall, the three years for which meteorological data were collected are significantly different. The 2014-15 winter was particularly cold, bringing the mean air temperature of that year more than a degree lower than the other two

seasons. Snow accumulation was approximately doubled that winter in comparison to the other two. Also, the summer melt season is longer in 2016 than in 2015. In comparison with past trends from a nearby site, KAN_L, the second year is more typical for this area (van As et al., 2012). To model a representative season, data from that second year (July 2015 to July 2016) were chosen as annual input for the model case study.

**4 Analysis**

Our objective is now to investigate how various processes active in Greenland's ablation zone influence $T_0$. In order for model results to achieve fidelity, inputs and parameters need to be representative of actual conditions. We therefore use the  meteorological dat to constrain the modeling experiments. Our modeling is focused at field site 33-km, where we have the most data for constraining the problem.

**4.1 Model Formulation**

The foundation for quantifying impacts of near-surface heat-transfer processes is a one-dimensional thermodynamic model. We argue that the processes tested here are close enough to being homogeneous that they can be adequately assessed in one dimension. The one exception is the measured heating events which are transient and spatially discrete, these are discussed in section 5.2 and are not included in the model analysis. Our model uses measured meteorological variables as the surface boundary condition and simulates ice temperature to 20 m, a depth chosen for consistency with measured data. The ice temperature at the depth of zero annual amplitude, $T_0$, is output from the bottom of the domain for each model experiment and used as a metric to compare net temperature changes between simulations. The model, its boundary conditions, and the experiments are all designed to test heat transfer processes within the ice itself. To maintain focus on ice processes, we ignore any atmospheric effects above the ice surface such as turbulent heat fluxes. The model does not, nor is it meant to, simulate the surface mass balance. We implement an Eulerian framework, treating the $z$ dimension as depth from a moving surface boundary so that emerging ice is moving through the domain and is removed when it melts at $z = 0$. We use a finite element model with a first-order linear element and 0.5-m mesh spacing refined to 2 cm near the surface. For a seamless representation of energy across the water/ice phase boundary, we implement an advection-diffusion enthalpy formulation (i.e. Aschwanden et al., 2012; Brinkerhoff and Johnson, 2013),

$$(\partial_t + w\partial_z)H = \partial_z(\alpha\partial_z H) + \phi/\rho_i \tag{1}$$

Here, $\partial$ is a partial derivative, $t$ is time, $w$ is the vertical ice velocity with respect to the lowering ice surface, $z$ is depth, $H$ is specific enthalpy, $\alpha$ is diffusivity, $\phi$ is any added energy source, and $\rho_i$ is the density of ice. The diffusivity term is enthalpy-dependent,

$$\alpha(H) = \begin{bmatrix} k_i/\rho_i C_p & cold, H < H_m \\ v/\rho_i & temperate, H > H_m \end{bmatrix} \tag{2}$$

where $k_i$ is the thermal conductivity of ice which we assume is constant over the small temperature range in this

study (~25°C), $C_p$ is the specific heat capacity which is again assumed constant, $v$ is the moisture diffusivity in temperate ice, and $H_m$ is the reference enthalpy at the melting point (all constants are shown in Table 2). Aschwanden et al. (2012) include a thermally diffusive component in temperate ice (i.e. $k_i \partial_z^2 T_m(P)$). However, since we consider only near-surface ice, where pressures ($P$) are low, this term reduces to zero. Using this formulation, energy moves by a sensible heat flux in cold ice and a latent heat flux in temperate ice. We assume that the latent heat flux, prescribed by a temperate ice diffusivity ($v/\rho_i$), is an order of magnitude smaller than the cold ice diffusivity ($k_i/\rho_i C_p$). We argue that this is representative of the near-surface ice when cold ice is impermeable to meltwater.

The desired model output is ice temperature. It has been argued that temperature is related to enthalpy through a continuous function, where the transition between cold and temperate ice is smooth over some 'cold-temperate transition surface' (M. Lüthi et al., 2002). On the other hand, we argue that cold ice is  impermeable to water except in open fractures (which we do not include in these simulations), so we use a stepwise transition,

$$T(H) = \begin{cases} (H - H_m)\big/C_p + T_m & cold \\ T_m & temperate \end{cases} \tag{3}$$

Additional enthalpy above the reference increases the water content in ice,

$$\omega(H) = \begin{cases} 0 & cold \\ (H - H_m)\big/L_f & temperate \end{cases} \tag{4}$$

[revised manuscript text omitted]

is negative (wintertime) we assume that it is controlling the air temperature, so it is already accommodated in our simulation; thus, the radiative energy input is ignored in the negative case. This radiative source term , is incorporated into equation (1) at each time step , $\phi_{rad} = \frac{Q}{20cm}$, where $Q$ is the measured radiative flux at the surface in W/m². All constants for the rotten cryoconite layer are the same as that for ice.

While some models treat the absorption of radiation in snow/ice more explicitly with a spectrally-dependent Beer-Lambert Law (Brandt and Warren, 1993), we argue that it is reasonable to assume that all wavelengths are absorbed near the surface over the length scales that we consider. The only documented value that we know of for an absorption coefficient in the cryoconite layer is 28 m⁻¹ (Lliboutry, 1965) which is close to that of snow (Perovich, 2007). If the properties are truly similar to that of snow, about 90 percent of the energy is absorbed in the uppermost 20 cm (Warren, 1982). Moreover, we argue that this is precisely the reason that the cryoconite layer only extends to a limited depth: it is a result of where radiative energy causes melting.

**4.2.4 Subsurface20 m Temperature Gradient**

Finally, in the fourth model experiment we change the boundary condition at the bottom of the domain. The free boundary is changed to a Neumann boundary with a gradient of -0.05°C/m , a value that approximately matches the measured gradient in our near-surface temperature measurementsat site 33-km at field site 33-km. Importantly, this simulated gradient is in the same direction, although with a larger magnitude, of the upper ~100 m of ice in our measurements of deep temperature profiles (Hills et al., 2017). In this case, the advective energy flux is upward, but the temperature gradient is negative, bringing colder ice to the surface. In addition, two limiting cases were tested, with gradients of +/- 0.15°C/m. This is the approximate range in measured gradients (Figure 2).

**4.3 Model Results**

The control model run of simple thermal diffusion predicts that ice temperature damps to approximately the mean annual air temperature of the study year (-9.9°C) by about 15 m below the ice surface. This result is in agreement with the analytical solution (Carslaw and Jaeger, 1959), but slightly different from the mean air temperature (-9.6°C) because the air can exceed the melting temperature in the summer while the ice cannot. Other atmospheric effects such as turbulent heat fluxes and the thermal inversion could also cause a difference between measured air temperature and ice surface temperature, but these are not considered here. For each model treatment, 1-4, the incorporation of an additional physical process changes the ice temperature. Differences between model runs are compared using $T_0$ at 210 m. Again, the model experiments are progressive, so each new experiment includes the processes from all previous experiments. Key results from each experiment are as follows (Figure 56):

1. Diffusion alone results in $T_0 = -9.9°C$, whereas observed temperatures range from -9.7°C to -8.1 at the 33-km field site.
2. Because the ablation rate is strongest in the summer, the effect of incorporating ablation is to counteract the diffusion of warm summer air temperatures. The result is a net cooling of $T_0$ from experiment (1) by -0.92°C.

3. Snow on the ice surface insulates the ice from the air temperature. In the winter, snow insulation keeps the ice warmer than the cold air, but with warm air temperatures in the spring it has the opposite effect. Because snow quickly melts in the springtime, the net effect of snow insulation is substantially more warming than cooling. $T_0$ for this experiment is +0.78°C warmer than the previous.

4. Radiative energy input mainly controls melting (van den Broeke et al., 2008), but incorporating this process does warm $T_0$ by +0.52°C.

5. Imposing a -0.05°C/m 20 m temperature gradient at the bottom of the model domain, consistent with observation, dramatically changes $T_0$ by -2.5°C.

Both ablation and the subsurfacedeep temperature gradient have a cooling effect on near-surface ice temperature. On the other hand, snow and radiative energy input have a warming effect. For this case study, the first three processes together result in almost no net change so that the modeled $T_0$ is close to the observed mean air temperature (Figure 56d). However, inclusion of the subsurfacedeep temperature gradient has a strong cooling effect on the simulated temperatures, bringing $T_0$ far from the mean measured air temperature. The limiting cases show that this bottom boundary condition strongly controls the near-surface temperature, with a range in the resulting $T_0$ values from -17.0°C to -2.0°C. In summary, measured ice temperatures are consistently warmer than both the measured air temperature and simulated ice temperature (Figure 67), except in the case of a positive subsurface gradient which is discussed below.

**5 Discussion**

Our observations show that measurements of near-surface ice in the ablation zone of western Greenland arecan be significantly warmer than would be predicted by diffusive heat exchange with the atmosphere. This is in agreement with past observations collected in other ablation zones (e.g. Hooke et al., 1983). With four experiments in a numerical model that progressively incorporate more physical complexity, we are unable to precisely match independent model output to observations. Our measurement and model output consistently point toward a disconnect between air and ice temperatures in the GrIS ablation zone, with ice temperatures being consistently warmer than the air.

**5.1 Ablation-Diffusion**

The strongest result from our model case study was a drop in $T_0$ by -2.5°C associated with the imposed subsurface deep icenegative 20 m temperature gradient. While it was important to test this scenario for one case, the temperature gradient we used was representative, but somewhat arbitrary. In reality, the observed20 m temperature gradients isare widely variable from one site to another and even within one site (Figure 2). Interestingly, full ice thickness temperature profiles show similar temperature gradients, both positive and negative, that persist for many hundreds of meters toward the bed (Harrington et al., 2015; Hills et al., 2017). Hence, the limiting cases were added to show simulation results over the range of measured gradients from our temperature strings. The resulting $T_0$ span a range of 19°C.

Our model could have tested additional temperature gradients, and those of the opposite (positive) sign likely would

 $T_0$  The majority of the subsurface temperature gradient that we measure are negative, and theoretically the gradient should be negative . Consider that fast horizontal velocities (~100 m/yr) advect cold ice from the divide to the ablation zone, and the air temperature lapse rate couples with the relatively steep surface gradients so that the surface warms rapidly toward the terminus. These conditions lead to a vertical temperature gradient below the ice surface that is negative (Hooke, 2005; pp. 131-135), as in our model example. The one exception is in the case of deep latent heating in a crevasse field (Harrington et al., 2015; sites S3 and S4) where the deep ice temperature is warmer than the mean air temperature rather than colder.

Overall, our results demonstrate that the effect of the subsurface temperature gradient is coupled to that of surface lowering. With respect to the surface, the temperature gradient below is advected upward as ice melts. There is a competition between surface lowering and diffusion of atmospheric energy into the ice; as near-surface ice is warmed, it can be removed quickly and a new boundary set. Therefore, our conceptualization of temperature in the near-surface ice of the ablation zone should not be a seasonally oscillating upper boundary with purely diffusive heat transfer (Carslaw and Jaeger, 1959), but one with advection and diffusion (Logan & Zlotnik, 1995; Paterson, 1972). This conceptualization is unique to the ablation zone because of the rapid rate of surface lowering, whereas a diffusive model for near-surface heat transfer is much more appropriate in the accumulation zone.

The disconnect between air and ice temperature implies that that the near-surface active layer in the ablation zone is shallow (i.e. less than 15 m) and could be skewed toward the subsurface temperature gradient. Therefore, the surface boundary condition has  weaker influence on diffusion for ice well below the surface. This is in contrast to the accumulation zone where new snow is advected downward, so the surface temperature quickly influences that at depth .  Under these conditions, it is no surprise that we see spatial variability in near-surface ice temperature even within one field site. That variability is simply an expression of the deeper ice temperature variations which are hypothesized to exist from variations in vertical advection (Hills et al., 2017), and do not have time to completely diffuse away before they are exposed at the surface.

**5.2 Subsurface Refreezing**

In two temperature strings we observe  heating events, the largest case being  8°C in 2 hours between 3 and 8 m below the ice surface (Figure 3c). These events are transient, they are spatially discrete, and they are generated at depth, all of which are most easily explained by  the refreezing of liquid water in cold ice. Similar refreezing events have been observed in firn (Humphrey et al., 2012), where they are not only important for ice temperature but could also imply a large storage reservoir for surface meltwater (Harper et al.,

2012). However, unlike firn, solid ice is impermeable to water unless fractures are present (Fountain et al., 2005).

In Greenland's ablation zone, much work is being done toprior work has demonstrated the importance of assess large-scale latent heating in open crevasses (Phillips et al., 2013; Poinar et al., 2016). Additionally, wWhile water-filled cavities have been observed in cold, near-surface ice on at least one a mountain glaciers (Jarvis and& Clarke, 1974; Paterson and& Savage, 1970). In our case, however, a mechanism for sudden water movement to depth isan explanation for refreezing water is not obvious: while the field site has occasional mm-aperture 'hairline' cracks, there arebut no visible open crevasses at the surface for routing water to depth.

However,As far as we know, this work iswe are the first to reportshow evidence of short-term transient latent heating events in cold ice, not obviously linked to open surface fractures.

-While the hairline and Fountain et al. (2005) suggest that fractures provide the main pathway for liquid water to move through temperate ice, a mechanism for sudden water movement to depth in the ablation zone of Greenland is not obvious. fFractures could perhaps move someare the most likely pathway to move water to depth, but to permitmove much water to move meters through cold ice they would need to be large enough that water moves quickly, and does not instantaneously refreeze. For example, A a 1-mm wide crack in ice that is -10°C freezes shut in about 45 seconds (Alley et al., 2005; eq. 8). T Assuming that there is a hydraulic potential gradient to drive water flow, that amount of time could beis long enough for small volumes of water to move at least 5-10 m below the surface, but would require a hydropotential gradient to drive water flowthrough cold ice. Thus, top-down hairline crevassing does not seem a plausible explanation for the events we observe.

A mechanism for sudden water movement to depth at our field site, 33 km, is not obvious. The field site at 33 km has no visible open crevasses at the surface, but does have occasional mm- aperture cracks.

Nevertheless,Importantly, several independent field observations in this area including hole drainage of water during hot-water drilling, ground-penetrating radar reflections, and borehole video observations, all pointing to the existence of subsurface air-filled and open fractures with apertures of up to a few cm (see supplementary). We suggest that these features occasionally move water to -10 m below the ice surface, where it refreezes and warms the ice as we have observed. That they are open at depth, but are narrow or non-existent at the surface, could be linked to the colder ice at depth and its stiffer rheology. Nath and Vaughan (2003) observed similar subsurface crevasses in firn, although in their case density controls the stiffness rather than temperature. We suggest that these features occasionally move water to -10 m below the ice surface, where it refreezes and warms the ice as we have observed.

On rare occasions, we argue that the aperture of the fractures open wider to the surface, where there is copious water stored in the cryoconite layer (Cooper et al., 2018) that can drain and refreeze at depth (Cooper et al., 2018). -While

the events seem to happen in the springtime and it would be tempting to assert that fracture opening coincides with speedup, our measurements of surface velocity at these sites show that this is not always the case. This may be due to that fact that the spring speed up coincides with early melt rather than peak melt and copious water in the cryoconite layer.

Latent heating in the form of these subsurface refreezing events is an obvious candidate for a source for the 'extra' heat  we observe in our temperature strings relative to simulations. Our data show that refreezing in subsurface fractures has the potential to warm ice substantially over short periods of time, and apparently this can occur in places where crevasses are not readily observed at the surface. Furthermore, the difference between measured and modeled temperatures (~3°C) is the equivalent of only ~1.7% water by volume. Our simplified one-dimensional model would not be well-suited to assess the influence of these latent heating events. Instead, we provide a simple calculation for energy input from the events by differencing the temperature profiles in time and integrating for total energy density (Figure 7 a-c),

$$\phi_{measured} = \frac{\rho_i C_p}{\Delta z} \int \Delta T \, dz \tag{9}$$

where $\Delta z$ is the total depth of the profile, and $\Delta T$ is the differenced temperature profile. Only sensors that are below the ice surface for the entire time period are considered. To calculate the total water content refrozen in the associated event, we remove the conductive energy fluxes from the total energy density calculated above. We do so by calculating the temperature gradients at the top and bottom of the measured temperature profile at each time step as in Cox et al. ( 2015).

$$\phi_{conductive} = \frac{-k_i}{\Delta z} \int \left. \frac{\partial T}{\partial z} \right|_{top} - \left. \frac{\partial T}{\partial z} \right|_{bottom} dt \tag{10}$$

The resulting energy sources are then converted to a volume fraction of water by

$$\omega_{measured} = \frac{\phi_{measured} - \phi_{conductive}}{\rho_w L_f} \tag{11}$$

where $\rho_w$ is the density of water. Results show that each year some fractions of a percent of water are refrozen (Figure 7 d-f). Through several seasons that amount of refreezing could easily add up to the ~3°C anomaly that we observe.

Unfortunately, without a more thorough investigation, we do not have enough evidence to show that these refreezing events are more than a local anomaly. Of our seven near-surface temperature strings, only T-14 and T-16 demonstrated refreezing events, and so we are not confident that they are temporally or spatially ubiquitous.

The only other logical mechanism for the warm offset between measurements and model results would be warming from below through a positive subsurface temperature gradient.

While it is tempting to associate deep warm ice with  residual heat from the exceptionally hot summers of 2010 and 2012 (Tedesco et al., 2013), this scenario is unlikely because the ablation rates are so high that any ice warmed during those years has likely already melted. Deeper latent heating from an upstream crevasse field is a more plausible alternative; however, in this area full-depth temperature profiles do not show deeper ice to be anomalously warmed except in one localized case (Hills et al., 2017).

**6 Conclusion**

We observe the temperature of ice at the depth of zero annual amplitude, $T_0$, in Greenland's ablation zone to be markedly warmer than the mean annual air temperature. These findings contradict predictions from purely diffusive heat transport but are not surprising considering the processes which impact heat transfer in ice of the ablation zone. High ablation rates in this area indicate that ice temperatures below ~15 m reflect the temperature of deep ice that is emerging to the surface, confirming that the ice does not have time to equilibrate with the atmosphere. In other words, ice flow brings cold ice to the surface at a faster rate than heat from the atmosphere can diffuse into the ablating surface. The coupling between rapid ablation and the spatial variability in deep ice temperature implies there will always be a disconnect between air and ice temperatures. Additionally, we observe  refreezing events below 5-10 m of cold ice. Meltwater is likely moving to that depth through subsurface fractures that are not obviously visible at the surface.

In analyzing a series of processes that control near-surface ice temperature, we find that some lead to colder ice, and others to warmer, but most are strong enough to dramatically alter the ice temperature from the purely diffusive case. With rapid ablation, a spatially variable temperature field, and subsurface refreezing events, $T_0$ in the ablation zone should not be expected to match the air temperature. That our measurements are consistently warmer, could simply be due to the limited number of observations we have, but latent heat additions are clearly measured and could be common in near-surface ice of the western Greenland ablation zone.

[Figure]

**Figure 1: A site map from southwest Greenland with field sites (red) named by their location with respect to the outlet terminus of Isunnguata Sermia. The inset shows locations of near-surface temperature strings (black) named by the year they were installed and a**n automated weather station  **(blue). Surface elevation contours are shown at 200-m spacing (Howat et al., 2014).**

[Figure]

**Figure 2: Near-surface ice temperature measurements from seven strings: T-11a, T-11b, T-14, T-15a, T-15b, T-15c,  and T-16. For each, the shaded region shows the range of measured temperatures over the entire measurement period, and the solid lines indicate the mean temperature profile. Depths are plotted with respect to the surface at the time of measurement, so sensor locations move toward the surface as ice melts. Strings with less than 11 months of data are slightly more transparent. For field sites at which the air temperature was measured for at least a full year, a dashed line shows the mean air temperature.**

[Figure]

**Figure 3: Three years of ice temperature measurements from the T-14 string. While this string was initially installed to 2 m depth,**  measurements are plotted with reference to the moving surface so the sensors move up throughout the time period, revealing a gray mask**. Transient features in the data include anomalously slow freeze-in behavior in one sensor (b) as well as heating events throughout the collection time period (c, d, and e). The heating events are plotted as a series of temperature profiles with the darker shades being later times and time steps between profiles of 2 hours (c), 10 hours (d), and 1 hour (e).**

[Figure]

**Figure 4: Heating events** from temperature string T-16. Profiles are plotted as in Figure 3 c, d, and e.  **The time step**s **between profiles** are **2** hours s and 4 hours (b).

[Figure]

Figure 5: Meteorological data from 33-km over three years including a) air temperature, b) ice surface location, c) snow depth, and d) net shortwave radiation. All data are plotted as a daily mean. The shaded region encloses the time period that is used for the model case study.

**Figure 56: Model results for fivesix separate simulations. In each case, twelve simulated temperature profiles are shown from throughout the yearlong period, and control results (from (a)) are displayed for comparison (gray). Differences between the simulations are analyzed quantitatively using $T_0$, the convergent temperature at 21 m. Processes are from top to bottom: a) control model run simulation of pure diffusion, b) ablation, c) snow insulation, d) radiative energy input, and finally e) subsurface20 m temperature gradient. The two limiting cases for the subsurface temperature gradient are plotted with dashed gray lines (e).**

[Figure]

**Figure 67: A comparison of model output (gray) and data from 33-km, including mean ice temperatures (red) and mean annual air temperatures for three seasons (black dashed). The observed measuredice temperatures are plotted differently fromthe same as in Figure-ure 2. . Instead of fixed sensor locations, the depth here is plotted at a distance relative to a melting surface (the same way as the model results). Note that three of the temperature strings failed before running for an entire yearcollected only ~9 months of data (transparent red). Mean temperatures from tThose three strings are biased cold near the surface because they collected more wintertime measurements than summertime.**

[Figure]

**Figure 7: Energy source for the observed heating events. a-c) Observed energy density through time for the differenced temperature profile calculated with equation (9) (black), and conductive energy density through time calculated with equation (10) (red). d-f) Percent by volume water refrozen for the associated source in (a-c). This value is proportional to the difference between the black and red lines above.** The temperature string from which measurements were taken is labeled at the top.

**Table 1: Temperature Strings**

| String Name | Data Time Period | Time Step (hr) | Sensor | # of Sensors | Sensor Spacing (m) | Latitude | Longitude | Elevation (m) |
|---|---|---|---|---|---|---|---|---|
| T-11a | 7/5/11 – 7/15/13 | 3 | Thermistor | 32 | 0.6 | 67.195175 | -49.719515 | 848 |
| T-11b | 7/11/11 – 12/17/11 | 3 | Thermistor | 32 | 0.6 | 67.201553 | -49.289058 | 1095 |
| T-14 | 7/18/14 – 8/14/16 | 0.5 | Thermistor | 31 | < 11 m deep – 0.5
> 11 m deep – 1.0 | 67.18127 | -49.56982 | 956 |
| T-15a | 08/17/16 | N/A | DS18B20 | 17 | < 15 m deep – 1.0
> 15 m deep – 3.0 | 67.18211 | -49.568272 | 954 |
| T-15b | 08/17/16 | N/A | DS18B20 | 17 | < 15 m deep – 1.0
> 15 m deep – 3.0 | 67.182054 | -49.568059 | 954 |
| T-15c | 08/17/16 | N/A | DS18B20 | 17 | < 15 m deep – 1.0
> 15 m deep – 3.0 | 67.182114 | -49.568484 | 954 |
| T-16 | 08/17/16 | N/A | DS18B20 | 18 | 0.5 | 67.18147 | -49.57025 | 951 |

**Table 2: Constants**

| Variable | Symbol | Value | Units | Reference |
|---|---|---|---|---|
| Reference Enthalpy | $H_m$ | 0 | J kg$^{-1}$ | |
| Ice Density | $\rho_i$ | 917 | kg m$^{-3}$ | Cuffey and Paterson (2010) |
| Snow Density | $\rho_s$ | 300 | kg m$^{-3}$ | |
| Water Density | $\rho_w$ | 1000 | kg m$^{-3}$ | |
| Specific Heat Capacity | $C_p$ | 2097 | J kg$^{-1}$ K$^{-1}$ | Cuffey and Paterson (2010) |
| Latent Heat of Fusion | $L_f$ | $3.335*10^5$ | J kg$^{-1}$ | Cuffey and Paterson (2010) |
| Thermal Conductivity of Ice | $k_i$ | 2.1 | J m$^{-1}$ K$^{-1}$ s$^{-1}$ | Cuffey and Paterson (2010) |
| Thermal Conductivity of Snow | $k_s$ | 0.2 | J m$^{-1}$ K$^{-1}$ s$^{-1}$ | Calonne et al. (2011) |
| Moisture Diffusivity | $v$ | $1*10^{-4}$ | kg m$^{-1}$ s$^{-1}$ | Aschwanden et al. (2012) |

---

## Author Response (AR2)

Comments and associated responses for:

**Processes influencing heat transfer in the near-surface ice of Greenland's ablation zone**
Benjamin H. Hills et al. 2018

**Minor Revisions**

Author responses are in blue.

Hills et al. 2018: Processes influencing heat transfer in the near-surface ice of Greenland's ablation zone

The main concerns of both reviewers addressed the definition of the metric T0, consistent depth referencing of observed and calculated vertical profiles, aspects of forcing data and lower boundary condition and enhanced discussion of subsurface refreezing processes. Overall, the the the manuscript was substantially improved with respect to these issues, thanks for that effort. The following comments address some remaining issues which I suggest to reconsider in more detail.

Thank you for this review. We agree that the manuscript was substantially improved from the previous revisions, and hope that the changes addressed below satisfy your final concerns. As you state above, the primary concerns of consistent depth referencing and subsurface refreezing were a focus of the previous revisions. The remaining concerns seem to focus on the boundary conditions at the surface and the bottom of our model domain. We hope that the response below gives some validation in our choice for boundary conditions.

1) Line above P2L15: I am still not confident regarding "T0", which is the essential metric used in this work. Here the authors state "Seasonal air temperature oscillations are diminished with depth into the ice, until they are negligible (i.e. ~1%) at a 'depth of zero annual amplitude'. On the other hand it is stated that "The mean value from the lowermost sensor (analogous to T0 ) is -3.2 at 27-km, … (P4L19, which refers to actual evaluation practice) ". Methodically seen this leaves kind of gap i.e., to show that at this lowermost sensor postion the 1% criterium is acutally matched (on average at least). Additionally this issue also questions in what extent the pragmatic choice of the lowermost sensor position is justified in view of the fact that right this level is intrinsically influenced by the lower boundary condition (where no temperature variability can occur by definition). Hence this question indirectly leads back to the one whether the simulations should not build on a deeper domain. I still recommend performing a sensitivity study addressing these issues and related uncertainties.

We assess the ~1% criterion in a section added to the supplement.

The lower boundary is a Neumann boundary condition, rather than a Dirichlet boundary condition as suggested in the comment above ("where no temperature variability can occur by definition"). The temperature at the bottom of the domain can therefore change in time, but we run the model "with a one-day time step until ice temperature at the bottom of the domain converges to a steady temperature" (as stated on P8L8).

In order to address concerns with the lower boundary, we have run additional tests presented in a section added to the supplement.

2) I recognize the added details concerning the meteorological data. Concerning the ice temperature measurements close to the surface some critical remarks may be given, though. If thermistors were housed in a "black casing" (as stated now, P3L30), than not only sensors lying at the surface may have been affected by solar induced heating. Dark cables/casing can experience significantly enhanced temperature and effect sensors (also by conduction within cables due to temperature gradients). Glendinning and Morris (1999) demonstrated for snow that corresponding effects can be of order >2°C @70cm depth. The indicated "discarding" procedure helps identifying problems close to 0°C, but does not identify/correct effects on sensors having negative temperatures. Overall one might expect that thermistors in the upper ca. 50cm below the surface are prone to a warm bias during summer (which can not be excluded by observations that at a certain point of time cables were found frozen into the ice.

We do not agree that the discarding procedure is only satisfactory for measurements close to 0°C. All values, no matter the temperature measured, are discarded after the sensor lies on the surface (as stated at P3L30).

We argue that the results presented here and the story given in the discussion are illustrated by the entire temperature profile measured to 20 m and are mostly independent of any small errors within the uppermost ~0.5 m. Having said that, we have modified the statement to convey a loss of confidence in the measurements as they approach the surface:
"Because each temperature sensor is in a black casing, measurement error surely increases due to solar heating as sensors move into the rotten cryoconite layer (~0.2 m depth), and we completely discard any measurement taken after the sensor is exposed at the surface."

3) The authors multiply mention that the paper is not focussing on meteorological aspects, which is fine. However, this not acceptable regarding obseved air temperature which constitutes essential model input and is used in context of interpretation of the results. I reckognize more detailed information on used instruments and their setup. But the potential influence of a likely inefficient radiation shield on the temperature measurements is still understated. According to the now given Fig. S2, a rather ineffective shield was used and significant radiation errors may be expected (also because being mounted close i.e., ca. 0.5m to the strongly reflecting surface and the low incidence angle of solar radiaton during transitional seasons). There are several studies incl. manufacturer statements, that this kind of screen can induce significant errors in temperature measurements (several °C depending on wind speed, too). Unfortunately, these effects are hard to quantify or to correct. At least, however, one expects some more critical comments that such uncertainties are inherent in the data and were not corrected. The currently used air temperature data are likely to be too high, which shall be discussed in the interpretation of the results, too.
Comparison to PROMICE data is ok (Fig. S4) is not really valuable in this context (due large distance between sites and respective need to correct for elevation differences). Regarding calculation of surface temperatures, the used emissivity shall be specified.

We appreciate these concerns with the air temperature measurement. We do agree that there are potential problems, especially in that the sensor location remains fixed as the surface melts instead of maintaining at 2-m distance. The most appropriate way to address this is through a comparison to what is considered a more sophisticated AWS station.

We added a statement in the manuscript pointing out this comparison:

"Out of concern for error in our air temperature measurement, we offer a comparison (Figure S4) to the nearby weather station monitored by the Programme for Monitoring of the Greenland Ice Sheet (PROMICE) (van As et al., 2012)."

Figure S4 was added to address a concern from the original review. We argue that while it is not a perfect comparison, it gives some justification in using our measured air temperature as a surface boundary condition.

Added to supplement:

"and considering ice a black body radiator."

4) Revision of Fig. 2 is acknowledged, however, I still can not fully agree to the argument that different i.e. inconsistent record lengths do not affect calculation of T0 ("… should be comparable because it is below any seasonal variation ..")

We address the robustness of $T_0$ in the section added to the supplement.

Further the caption mentions " For field sites at which the air temperature was measured for at least a full year, a dashed line shows the mean air temperature". Why then still showing dashed lines for b, de and f, which do not cover a full year?

The statement is true as it stands in the manuscript. The *air temperature* was measured at those sites for over one year. For these strings that you mention, the *ice temperature* was not measured for more than one year, but that has already been addressed at several places in the manuscript including in this figure caption.

Overall, I do not fully support diverse argumetns why uncertainties in air temperature measurements and its use as model input is not an issue in context of this investigation. But I also see the weak point that the vailable data hardly allow a better approach. In this perspective, corresponding sensitivity studies could have been valuable. This is menat in sense of disturbing input (i.e., air temperature) and investigate the impact on simulatin results (T0 basically).

As we stated in the original review, this study is focused on processes within the ice. We argue that the requested sensitivity studies for the surface boundary condition would detract from the overall story presented here. As stated above, we tried to ease some of these concerns about the surface boundary condition by comparing our measurements to the nearby KAN_L weather station. A thorough investigation of the atmospheric processes effecting this surface boundary condition is best left for a separate study. We acknowledge that our meteorological instruments, while appropriate for this study of heat transfer in ice, are not state-of-the-art and therefore leave room for error in the atmospheric interpretation. We have added statements in the manuscript addressing the concerns with our instrumentation (see comments above), and suggest that if an

intensive study of the atmospheric processes is to be done it should be done by a group that focuses primarily on meteorological data.

5) P5L16: …"Net radiation is less than zero in the winter (net outgoing because of thermal emission in the infrared wavelengths)", may be reformulated to account for the fact that not only emission counts, but that this component emission prevails over atmospheric input)

Changed to:
"(net outgoing because thermal emission in the infrared wavelengths dominates over atmospheric inputs)"

6) P6L8: "…. model uses measured meteorological variables as the surface boundary condition and simulates ice temperature to 21 m, a depth chosen for consistency with measured data. The ice temperature at the depth of zero annual amplitude, T0, is output from the bottom of the domain for each model experiment and used as a metric…." Admittently, I am still not convinced about several aspects in this context as long as respective uncertainties are not addressed quantitatively. In particular this concerns use of air temperature as forcing at the upper boundary (ignoring measurements uncertainties and stratification effects) and the implementation of the lower boundary condition at a depth close to the depth of average T0. Both is still rather superficial treated.

We have addressed these issues in our revision and throughout this response. Our approach to air temperature is not fundamentally different than many other studies in Greenland. We utilize modern and high quality instrumentation for measurement, and the paper acknowledges potential shortcomings of the data and methods. We provide a comparison to the nearby KAN_L station, which does not reveal an intrinsic flaw with our data.

We show in the section added to the supplement that our modeling procedure of the lower boundary condition does in fact, have no impact on our results (see below).

P9L4: I need some help how of Phi(rad) in given dimensions is compatible with equ.1

Changed to $\phi_{rad} = \frac{Q}{0.2m}$ (instead of $cm$) for clarity. The units are
$$[Q] = W/m^2$$
$$[\phi_{rad}] = W/m^3$$
$$\left[\frac{\phi_{rad}}{\rho_i}\right] = \frac{J}{kg\ s}$$
Which is the time rate of change of the specific enthalpy (as a source term).

P10L15: "The limiting cases show that this bottom boundary condition strongly controls the near-surface temperature, with a range in the resulting T0 values from -17.0°C to -2.0°C. In summary, measured ice temperatures are consistently warmer than both the measured air temperature and simulated ice temperature …" This is an expected and most important result, which has to be re-emphasized in the discussions, too. And again, it would be most interesting to know in what extent this issue depends on alternative depth of the model domain and corresponding specification of the lower boundary condition.

We feel that this result has been given appropriate emphasis in the discussion. The subsurface gradient is the focus of section 5.1.

We address problems with the lower boundary condition in the section added to the supplement. Unfortunately, we cannot include these limiting gradients in the deeper tests, because the gradients are specifically prescribed based on measured values at ~20 m depth. Setting a Neumann boundary with the specified gradient at 50 m instead would give a different result, not because the model is wrong but because that is a completely different scenario.

**Processes influencing heat transfer in the near-surface ice of Greenland's ablation zone**

Benjamin H. Hills[1,2], Joel T. Harper[2], Toby W. Meierbachtol[2], Jesse V. Johnson[3], Neil F. Humphrey[4], Patrick J. Wright[5,2]

[1]Department of Earth and Space Sciences, University of Washington, Seattle, Washington, USA
[2]Department of Geosciences, University of Montana, Missoula, Montana, USA
[3]Department of Computer Science, University of Montana, Missoula, Montana, USA
[4]Department of Geology and Geophysics, University of Wyoming, Laramie, Wyoming, USA
[5]Inversion Labs LLC, Wilson, Wyoming, USA

*Correspondence to*: Benjamin H. Hills (bhills@uw.edu)

**Abstract.** To assess the influence of various heat transfer processes on the thermal structure of ice near the surface of Greenland's ablation zone, we compare in-situ measurements with thermal modeling experiments. A total of seven temperature strings were installed at three different field sites, each with between 17 and 32 sensors and extending up to 21 meters below the surface. In one string, temperatures were measured every 30 minutes, and the record is continuous for more than three years. We use these measured ice temperatures to constrain our modeling experiments, focusing on four isolated processes and assessing the relative importance of each for the near-surface ice temperature: 1) the moving boundary of an ablating surface, 2) thermal insulation by snow, 3) radiative energy input, and 4) subsurface ice temperature gradients below the seasonally active near-surface layer. In addition to these four processes, transient heating events were observed in two of the temperature strings. Despite no observations of meltwater pathways to the subsurface, these heating events are likely the refreezing of liquid water below 5-10 meters of cold ice. Together with subsurface refreezing, the five heat transfer mechanisms presented here account for measured differences of up to 3°C between the mean annual air temperature and the ice temperature at the depth where annual temperature variability is dissipated. Thus, in Greenland's ablation zone, the mean annual air temperature is not a reliable predictor of the near-surface ice temperature, as is commonly assumed.

**1 Introduction**

Bare ice regions of the Greenland ice sheet have high summer melt rates. Here, the surface ice temperature is important to ablation processes such as melt, water storage, runoff, and albedo modifications associated with the surface cryoconite layer. The ice surface temperature also acts as an essential boundary condition for the transfer of heat into deeper ice below, and is therefore important for ice flow modeling (e.g. Meierbachtol et al., 2015) as well as interpretation of borehole temperature measurements (Harrington et al., 2015; Hills et al., 2017; Lüthi et al., 2015). In order to constrain the rate of ice melting, and more generally to understand the mechanisms which move energy between the ice and the atmosphere above, we must understand the processes that control near-surface heat transfer in bare ice.

Heat transfer at the ice surface is dominated by thermal diffusion from the overlying air (Cuffey and Paterson, 2010). Seasonal air temperature oscillations are diminished with depth into the ice, until they are negligible (i.e. ~1%) at a 'depth of zero annual amplitude' (van Everdingen, 1998). The exact location of this depth is dependent on the thermal diffusivity of the material through which heat is conducted as well as the period of oscillations (Carslaw and Jaeger, 1959; pp. 64-70). In theory, the temperature at the depth of zero annual amplitude, a value we will call $T_0$, is approximately constant and equal to the mean annual air temperature. In snow and ice, the depth of zero annual amplitude is approximately 10 and 15 m respectively (Hooke, 1976). For this reason, studies in the cryosphere often use $T_0$ as a proxy for the mean air temperature, drilling to 10 or more meters to measure the snow or ice temperature at that depth (Loewe, 1970; Mock and Weeks, 1966).

In places where heat transfer is purely diffusive, the snow or ice is homogeneous, and interannual climate variations are minimal, $T_0$ is a good approximation for the mean air temperature. However, prior studies have shown that, in many areas of glaciers and ice sheets, the relationship between air and ice temperatures can be substantially altered by additional heat transfer processes. For example, in the percolation zone, infiltration and refreezing of surface meltwater warm the subsurface (Humphrey et al., 2012; Müller, 1976). Studies have also revealed ice anomalously warmed by 5°C or more in the ablation zone (Hooke et al., 1983; Meierbachtol et al., 2015), but the mechanisms for this are unclear.

Hooke et al. (1983) explored the impacts of several heat transfer processes within near-surface ice at Storglaciären and the Barnes Ice Cap. They focused on the wintertime snowpack which acts as insulation to cold air temperatures but is permeable to meltwater percolation. Their results showed that the average ice temperature at and below the equilibrium line of those glaciers tends to be higher than the mean annual air temperature. They attributed the observed difference mainly to snow insulation because the strength of their measured offset was correlated to the thickness of the snowpack.

In this study, we expand the analysis of Hooke et al. (1983) and turn focus to the GrIS ablation zone with near-surface temperature profiles from seven locations. We use our temperature measurements in conjunction with a one-dimensional model to assess heat transfer processes in this area. The processes which make the ablation zone different from other areas of a glacier or ice sheet are, first, that the ice surface spends much of the summer period pinned at the melting point, despite slightly warmer air temperatures. Next, high ablation rates counter emergent ice flow, removing the ice surface and exposing deeper ice, along with its heat content, to the surface. The contrast of a

wintertime snowpack to bare ice in the summer enables an insulating effect during winter months. The deep penetration of solar radiation into bare ice results in subsurface heating and melting (Brandt and Warren, 1993; Liston et al., 1999). Finally, surface melt can move through open fractures, carrying latent heat with it to deeper and colder ice, and upon refreezing, the meltwater warms that ice below the surface (Jarvis and Clarke, 1974; Phillips et al., 2010).

Our near-surface temperature observations represent an aggregated sum of the processes mentioned above. A numerical model can be used to partition the relative importance of those processes, but only with measurements in hand as validation. Therefore, confidently constraining the role of near-surface heat transfer processes requires temperature measurements with both high temporal and spatial resolution, and records that span hours to seasons.

**2 Field Site and Instrumentation**

Field observations used in this study are from three sites in western Greenland (Fig. 1). Each site is named by its location with respect to the terminus of Isunnguata Sermia, a land-terminating outlet glacier. The equilibrium line altitude is at about 1500 m elevation in this area (van de Wal et al., 2012), which is 400 m above the furthest inland site, 46-km, so all sites are well within the ablation zone and ablation rates are high (2-3 m/yr). Solar radiation in the summer creates a layer of interconnected cryoconite holes at the ice surface, and water moving through that cryoconite layer converges into surface streams. There are no large supraglacial lakes in the immediate area of any site; all streams eventually drain from the surface through moulins. A series of dark folded layers emerge to the ice surface in this region of the ice sheet (Wientjes and Oerlemans, 2010).

At each field site, boreholes for temperature instrumentation were drilled from the surface to between 10 and 21 m depth using hot-water methods. In total, seven strings of temperature sensors were installed – one at both sites 27-km and 46-km in 2011, followed by five at site 33-km between 2014 and 2016. Strings are named by the year they were installed. Each consists of between 17 and 32 sensors spaced at 0.5-3.0 m along the cable (Table 1). In 2011 and 2014, thermistors were used as temperature sensors. The thermistors have measurement resolution of 0.02°C and accuracy of about 0.5°C after accounting for drift (Humphrey et al., 2012). In subsequent years, we used a digital temperature sensor (model DS18B20 from Maxim Integrated Products, Inc.). This sensor has resolution 0.0625°C and about the same accuracy as the thermistors. To increase accuracy, each sensor was lab-calibrated in a 0°C bath, and field-calibrated with a temperature measurement during freeze-in (borehole water is exactly 0°C). Because each temperature sensor is in a black casing, measurement error surely increases due to solar heating as sensors move into the rotten cryoconite layer (~0.2 m depth), and we completely discard any measurement taken after the sensor is exposed at the surface.

Meteorological variables were measured at each field site as well. In this study, we use the near-surface (~2-m) air temperature (Vaisala HMP60 with a radiation shield), the net radiative heat flux over all wavelengths shorter than 100 μm (Kipp and Zonen NR Lite), and the change in surface elevation measured with a sonic distance sensor (Campbell SR50A). Data from the sonic distance sensor are filtered manually, removing any obvious outliers (more than 0.5 m from the surrounding measurements). The filtered data are then partitioned into two variables,

cumulative ablation during the melt season and changes in snow depth during the winter. An automated weather station with all the above instrumentation was mounted on a fixed pole frozen in the ice, with segments being removed from the mounting pole each summer so the instrumentation remains close to the surface and does not extend significantly into the air temperature inversion (Miller et al., 2013). Out of concern for error in our air temperature measurement, we offer a comparison (Fig. S4) to the nearby weather station monitored by the Programme for Monitoring of the Greenland Ice Sheet (PROMICE) (van As et al., 2012). The measurement frequency for meteorological data varies from ten minutes to an hour, but all data are collapsed to a daily mean for input to a heat transfer model.

In addition to ice temperature and meteorological measurements, investigations of the subsurface were completed at site 33-km with a borehole video camera and a high-frequency ground-penetrating radar survey (see Supplement).

These investigations were carried out in pursuit of what we think may have been subsurface fractures that are not expressed at the ice surface (described in section 5.2). With five temperature sensor strings, an automated weather station, and the subsurface investigations, site 33-km is by far the most thoroughly studied of the three sites. For that reason, measurements from this site serve as the foundation for the model case study presented in section 4.

**3 Results**

**3.1 Observed Ice Temperature**

Near-surface ice temperatures were measured through time in seven shallow boreholes at three different field sites (Fig. 2). Although hot-water drilling methods temporarily warm ice near the instrumentation, the ice around these

shallow boreholes cools to its original temperature within days to weeks. Measured temperatures are spatially variable between sites. The mean value from the lowermost sensor (analogous to $T_0$) is -3.2 at 27-km, -8.6 at 46-km, and from -9.7°C to -8.1 at 33-km. In all cases, measured $T_0$ values are warmer than the mean annual air temperature. Temperature gradients are calculated by fitting a line to the mean temperature of the four lowermost sensors for each string. These gradients are also variable, typically being between -0.15 and 0.0°C/m but +0.16°C/m at the 27-km field site (positive being increasing temperature with depth below the surface). As expected, the direction of temperature gradients measured here correlate with those measured in the uppermost ~100 m for full-thickness temperature profiles (Harrington et al., 2015; Hills et al., 2017).

Even the five temperature profiles measured at site 33-km exhibit some amount of spatial variability. Three temperature strings, T-15a, b, and c, are all similar, having strong negative temperature gradients (approximately -0.14°C/m), and cold $T_0$ temperatures (-9.6°C). Close to the surface, these three temperature strings are cold

compared to the others. However, those strings stopped collecting measurements in May 2017 and did not yield a full year of data. The missing summer period explains the strong positive temperature gradient near the surface for those three strings. T-16 is the shortest string, extending to only 9.5 m depth. This short string exhibits the smallest range in temperatures throughout a season with the coldest surface temperatures not even reaching -15°C. In terms of mean temperature, T-16 is similar to T-14, having a small negative temperature gradient and warm temperatures in comparison to those of T-15a, b, and c. Based on our observations, spatial variability in near-surface ice

temperature at site 33-km is controlled on the scale of hundreds of meters. Proximal observations from the nearby T-15a, b, and c strings are similar to one another, but greater variability is observed when including the more distant strings, T-14 and T-16.

Closer inspection of the measured temperature record through time reveals the transient nature of near-surface ice temperature (Fig. 3). As expected, these data show a strong seasonal oscillation near the surface. During the melt season, the ice surface quickly drops as ice is warmed to the melting point. Just below the surface, the winter cold wave persists for several weeks into the summer season. In string T-14 we observe delayed freeze-in behavior in one sensor (Fig. 3b) and transient heating events during the melt seasons (Fig. 3c, 3d, 3e). Similar heating events were observed in string T-16 (Fig. 4), but not in any other. The events range in magnitude, but in one instance ice is warmed from -10°C to -2°C in 2 hours (Fig. 3c). We can only speculate on the origins of these events, and address this below in section 5.2.

**3.2 Meteorological Data**

Meteorological data from site 33-km were observed over three years (Supplement Fig. S4). Air temperatures are normally at or above the melting temperature during the summer but fall to below -30°C in winter months. The measured ablation rate is 2-3 m/yr and maximum snow accumulation is only up to 0.5 m. Net radiation is less than zero in the winter (net outgoing because thermal emission in the infrared wavelengths dominates over atmospheric inputs) but over 100 W/m$^2$ (daily mean) on some days in the summer.

The mean air temperature over the entire measurement period at site 33-km (-10.5°C) is cold in comparison to measured ice temperatures at that site (Fig. 2; T-14, T-15a, T-15b, T-15c, and T-16). This warm anomaly between the ice and air temperature is also observed at sites 27-km and 46-km, where ice is warmer than the measured air temperature and significantly warmer than the reference from a regional climate model (Meierbachtol et al., 2015). Interestingly, we measure almost no winter snowpack at sites 27-km and 46-km due to low precipitation and strong winds during the time period over which those data were collected (2011-2013). Our observations are thus in contradiction to the inferences made by Hooke et al. (1983) in Arctic Canada, where the offset between air and ice temperature appeared to be primarily a result of snow insulation.

[revised manuscript text omitted]

**4.2.4 Subsurface Temperature Gradient**

Finally, in the fourth model experiment we change the boundary condition at the bottom of the domain. The free boundary is changed to a Neumann boundary with a gradient of -0.05°C/m, a value that approximately matches the measured gradient at site 33-km. Importantly, this simulated gradient is in the same direction, although with a larger magnitude, of the upper ~100 m of ice in our measurements of deep temperature profiles (Hills et al., 2017). In this case, the advective energy flux is upward, but the temperature gradient is negative, bringing colder ice to the surface. In addition, two limiting cases were tested, with gradients of +/- 0.15°C/m. This is the approximate range in measured gradients (Fig. 2).

**4.3 Model Results**

The control model run of simple thermal diffusion predicts that ice temperature converges to approximately the mean annual air temperature of the study year (-9.9°C) by about 15 m below the ice surface. This result is in agreement with the analytical solution (Carslaw and Jaeger, 1959), but slightly different from the mean air temperature (-9.6°C) because the air can exceed the melting temperature in the summer while the ice cannot. Other atmospheric effects such as turbulent heat fluxes and the thermal inversion could also cause a difference between measured air temperature and ice surface temperature, but these are not considered here. For each model treatment, 1-4, the incorporation of an additional physical process changes the ice temperature. Differences between model runs are compared using $T_0$ at 21 m. Again, the model experiments are progressive, so each new experiment includes the processes from all previous experiments. Key results from each experiment are as follows (Fig. 5):

1. Diffusion alone results in $T_0 = -9.9°C$, whereas observed temperatures range from -9.7°C to -8.1 at the 33-km field site.
2. Because the ablation rate is strongest in the summer, the effect of incorporating ablation is to counteract the diffusion of warm summer air temperatures. The result is a net cooling of $T_0$ from experiment (1) by -0.92°C.
3. Snow on the ice surface insulates the ice from the air temperature. In the winter, snow insulation keeps the ice warmer than the cold air, but with warm air temperatures in the spring it has the opposite effect. Because snow quickly melts in the springtime, the net effect of snow insulation is substantially more warming than cooling. $T_0$ for this experiment is +0.78°C warmer than the previous.
4. Radiative energy input mainly controls melting (van den Broeke et al., 2008), but incorporating this process does warm $T_0$ by +0.52°C.
5. Imposing a -0.05°C/m temperature gradient at the bottom of the model domain, consistent with observation, dramatically changes $T_0$ by -2.5°C.

Both ablation and the subsurface temperature gradient have a cooling effect on near-surface ice temperature. On the other hand, snow and radiative energy input have a warming effect. For this case study, the first three processes

together result in almost no net change so that the modeled $T_0$ is close to the observed mean air temperature (Fig. 5d). However, inclusion of the subsurface temperature gradient has a strong cooling effect on the simulated temperatures, bringing $T_0$ far from the mean measured air temperature. The limiting cases show that this bottom boundary condition strongly controls the near-surface temperature, with a range in the resulting $T_0$ values from -17.0°C to -2.0°C. In summary, measured ice temperatures are consistently warmer than both the measured air temperature and simulated ice temperature (Fig. 6), except in the case of a positive subsurface gradient which is discussed below.

**5 Discussion**

Our observations show that measurements of near-surface ice in the ablation zone of western Greenland are significantly warmer than would be predicted by diffusive heat exchange with the atmosphere. This is in agreement with past observations collected in other ablation zones (Hooke et al., 1983). With four experiments in a numerical model that progressively incorporate more physical complexity, we are unable to precisely match independent model output to observations. Our measurement and model output point toward a disconnect between air and ice temperatures in the GrIS ablation zone, with ice temperatures being consistently warmer than the air.

**5.1 Ablation-Diffusion**

The strongest result from our model case study was a drop in $T_0$ by -2.5°C associated with the imposed subsurface temperature gradient. While it was important to test this scenario for one case, the temperature gradient we used was representative but somewhat arbitrary. In reality, the observed temperature gradients are widely variable from one site to another and even within one site (Fig. 2). Interestingly, full ice thickness temperature profiles show similar temperature gradients, both positive and negative (Harrington et al., 2015; Hills et al., 2017). Hence, the limiting cases were added to show simulation results over the range of measured gradients from our temperature strings. The resulting $T_0$ span a range of 19°C.

The majority of the subsurface temperature gradients that we measure are negative, and theoretically the gradient should be negative. Consider that fast horizontal velocities (~100 m/yr) advect cold ice from the divide to the ablation zone, and the air temperature lapse rate couples with the relatively steep surface gradients so that the surface warms rapidly toward the terminus. These conditions lead to a vertical temperature gradient below the ice surface that is negative (Hooke, 2005; pp. 131-135), as in our model example. The one exception is in the case of deep latent heating in a crevasse field (Harrington et al., 2015; sites S3 and S4) where the deep ice temperature is warmer than the mean air temperature rather than colder.

Overall, our results demonstrate that the effect of the subsurface temperature gradient is coupled to that of surface lowering. With respect to the surface, the temperature gradient below is advected upward as ice melts. There is a competition between surface lowering and diffusion of atmospheric energy into the ice; as near-surface ice gets warmer, it can be removed quickly and a new boundary set. Therefore, our conceptualization of temperature in the near-surface ice of the ablation zone should not be a seasonally oscillating upper boundary with purely diffusive heat transfer (Carslaw and Jaeger, 1959), but one with advection and diffusion (Logan and Zlotnik, 1995; Paterson,

1972). This conceptualization is unique to the ablation zone because of the rapid rate of surface lowering, whereas a diffusive model for near-surface heat transfer is much more appropriate in the accumulation zone.

The disconnect between air and ice temperature implies that that the near-surface active layer in the ablation zone is shallow (i.e. less than 15 m) and could be skewed toward the subsurface temperature gradient. Therefore, the surface boundary condition has weak influence on diffusion for ice well below the surface. This is in contrast to the accumulation zone where new snow is advected downward, so the surface temperature quickly influences that at depth. Under these conditions, it is no surprise that we see spatial variability in near-surface ice temperature even within one field site. That variability is simply an expression of the deeper ice temperature variations which are hypothesized to exist from variations in vertical advection (Hills et al., 2017), and do not have time to completely diffuse away before they are exposed at the surface.

**5.2 Subsurface Refreezing**

We observe heating events in two temperature strings, the largest case being 8°C in 2 hours between 3 and 8 m below the ice surface (Fig. 3c). These events are transient, they are spatially discrete, and they are generated at depth, all of which are most easily explained by the refreezing of liquid water in cold ice. Similar refreezing events have been observed in firn (Humphrey et al., 2012), where they are not only important for ice temperature but could also imply a large storage reservoir for surface meltwater (Harper et al., 2012). However, unlike firn, solid ice is impermeable to water unless fractures are present (Fountain et al., 2005). Two persistently warm features are also observed between 5 and 10 m depth into the winters of 2015 and 2017 (Fig. 3a). We interpret these as a nearby latent heat source, either with running or ponded water that does not freeze for an extended time.

In Greenland's ablation zone, prior work has demonstrated the importance of large-scale latent heating in open crevasses (Phillips et al., 2013; Poinar et al., 2016). Additionally, water-filled cavities have been observed in cold, near-surface ice on mountain glaciers (Jarvis and Clarke, 1974; Paterson and Savage, 1970). In our case, however, an explanation for refreezing water is not obvious. While the field site has occasional mm-aperture 'hairline' cracks, there are no visible open crevasses at the surface for routing water to depth. As far as we know, this work is the first to report evidence of short-term transient latent heating events in cold ice, not obviously linked to open surface fractures.

While the hairline fractures could perhaps move some water, to permit much water to move meters through cold ice they would need to be large enough that water moves quickly and does not instantaneously refreeze. For example,  a 1-mm wide crack in ice that is -10°C freezes shut in about 45 seconds (Alley et al., 2005; eq. 8). That amount of time could be long enough for small volumes of water to move 5-10 m below the surface but would require a hydropotential gradient to drive water flow. Thus, top-down hairline crevassing does not seem a plausible explanation for the events we observe.

Importantly, several independent field observations in this area including hole drainage of water during hot-water drilling, ground-penetrating radar reflections, and borehole video observations, all point to the existence of subsurface air-filled and open fractures with apertures of up to a few cm (see Supplement). That they are open at depth, but are narrow or non-existent at the surface, could be linked to the colder ice at depth and its stiffer rheology.

Nath and Vaughan (2003) observed similar subsurface fractures in firn, although in their case density controls the stiffness rather than temperature. On rare occasions, we argue that the aperture of the fractures open wider to the surface, where there is copious water stored in the cryoconite layer (Cooper et al., 2018) that can drain and refreeze at depth. While the events seem to happen in the springtime and it would be tempting to assert that fracture opening coincides with speedup, our measurements of surface velocity at these sites show that this is not always the case. This may be due to that fact that the spring speedup coincides with early melt rather than peak melt and copious water in the cryoconite layer.

Latent heating in the form of these subsurface refreezing events is an obvious candidate for a source for the 'extra' heat we observe in our temperature strings relative to simulations. Our data show that refreezing in subsurface fractures has the potential to warm ice substantially over short periods of time, and apparently this can occur in places where open crevasses are not readily observed at the surface. Furthermore, the difference between measured and modeled temperatures (up to 3°C) is the equivalent of only ~1.7% water by volume. Our simplified one-dimensional model would not be well-suited to assess the influence of these latent heating events. Instead, we provide a simple calculation for energy input from the events by differencing the temperature profiles in time and integrating for total energy density (Fig. 7 a-c),

$$\phi_{measured} = \frac{\rho_i C_p}{\Delta z} \int \Delta T \, dz \qquad (9)$$

where $\Delta z$ is the total depth of the profile, and $\Delta T$ is the differenced temperature profile. Only sensors that are below the ice surface for the entire time period are considered. To calculate the total water content refrozen in the associated event, we remove the conductive energy fluxes from the total energy density calculated above. We do so by calculating the temperature gradients at the top and bottom of the measured temperature profile at each time step as in Cox et al. (2015).

$$\phi_{conductive} = \frac{-k_i}{\Delta z} \int \frac{\partial T}{\partial z}_{top} - \frac{\partial T}{\partial z}_{bottom} \, dt \qquad (10)$$

The resulting energy sources are then converted to a volume fraction of water by

$$\omega_{measured} = \frac{\phi_{measured} - \phi_{conductive}}{\rho_w L_f} \qquad (11)$$

where $\rho_w$ is the density of water. Results show that each year some fractions of a percent of water are refrozen (Fig. 7). Through several seasons that amount of refreezing could easily add up to the ~3°C anomaly that we observe. Unfortunately, without a more thorough investigation, we do not have enough evidence to show that these refreezing events are more than a local anomaly. Of our seven near-surface temperature strings, only T-14 and T-16 demonstrated refreezing events, so we are not confident that they are temporally or spatially ubiquitous.

The only other logical mechanism for the warm offset between measurements and model results would be warming from below through a positive subsurface temperature gradient. While it is tempting to associate deep warm ice with residual heat from the exceptionally hot summers of 2010 and 2012 (Tedesco et al., 2013), this scenario is unlikely because the ablation rates are so high that any ice warmed during those years has likely already melted. Deeper

latent heating from an upstream crevasse field is a more plausible alternative; however, in this area full-depth temperature profiles do not show deeper ice to be anomalously warmed except in one localized case (Hills et al., 2017).

**6 Conclusion**

We observe the temperature of ice at the depth of zero annual amplitude, $T_0$, in Greenland's ablation zone to be markedly warmer than the mean annual air temperature. These findings contradict predictions from purely diffusive heat transport but are not surprising considering the processes which impact heat transfer in ice of the ablation zone. High ablation rates in this area indicate that ice temperatures below 15 m reflect the temperature of deep ice that is emerging to the surface, confirming that the ice does not have time to equilibrate with the atmosphere. In other words, ice flow brings cold ice to the surface at a faster rate than heat from the atmosphere can diffuse into the ablating surface. The coupling between rapid ablation and the spatial variability in deep ice temperature implies there will always be a disconnect between air and ice temperatures. Additionally, we observe refreezing events below 5-10 m of cold ice. Meltwater is likely moving to that depth through subsurface fractures that are not obviously visible at the surface.

In analyzing a series of processes that control near-surface ice temperature, we find that some lead to colder ice, and others to warmer, but most are strong enough to dramatically alter the ice temperature from the purely diffusive case. With rapid ablation, a spatially variable temperature field, and subsurface refreezing events, $T_0$ in the ablation zone should not be expected to match the air temperature. That our measurements are consistently warmer, could simply be due to the limited number of observations we have, but latent heat additions are clearly measured and could be common in near-surface ice of the western Greenland ablation zone.

**Figure 1: A site map from southwest Greenland with field sites (red) named by their location with respect to the outlet terminus of Isunnguata Sermia. The inset shows locations of near-surface temperature strings (black) named by the year they were installed and an automated weather station (blue). Surface elevation contours are shown at 200-m spacing (Howat et al., 2014).**

**Figure 2: Near-surface ice temperature measurements from seven strings: T-11a, T-11b, T-14, T-15a, T-15b, T-15c, and T-16. For each, the shaded region shows the range of measured temperatures over the entire measurement period, and the solid line indicates the mean temperature profile. Depths are plotted with respect to the surface at the time of measurement, so sensor locations move toward the surface as ice melts. Strings with less than 11 months of data are slightly more transparent. For field sites at which the air temperature was measured for at least a full year, a dashed line shows the mean air temperature.**

**Figure 3: Three years of ice temperature measurements from the T-14 string. While this string was initially installed to 21-m depth, measurements are plotted with reference to the moving surface, so the sensors move up throughout the time period, revealing a gray mask. Transient features in the data include anomalously slow freeze-in behavior in one sensor (b) as well as heating events throughout the collection time period (c, d, and e). The heating events are plotted as a series of temperature profiles with the darker shades being later times and time steps between profiles of 2 hours (c), 10 hours (d), and 1 hour (e).**

**Figure 4: Heating events from temperature string T-16. Profiles are plotted as in Figure 3 c, d, and e. The time steps between profiles are 2 hours (a) and 4 hours (b).**

**Figure 5: Model results for five separate simulations. In each case, twelve simulated temperature profiles are shown from throughout the year-long period, and control results (from (a)) are displayed for comparison (gray). Differences between the simulations are analyzed quantitatively using $T_0$, the convergent temperature at 21 m. Processes are, from top to bottom: a) control simulation of pure diffusion, b) ablation, c) snow insulation, d) radiative energy input, and finally e) subsurface temperature gradient. The two limiting cases for the subsurface temperature gradient are plotted with dashed gray lines (e).**

**Figure 6: A comparison of model output (gray) and data from 33-km, including mean ice temperatures (red) and**

mean annual air temperatures for three seasons (black dashed). The observed ice temperatures are plotted the same as in Figure 2. Note that three of the temperature strings collected only ~9 months of data (transparent red). Mean temperatures from those three strings are cold near the surface because they collected more wintertime measurements than summertime.

Figure 7:  Energy source for the observed heating events. a-c) Observed energy density through time for the differenced temperature profile calculated with equation (9) (black), and conductive energy density through time calculated with equation (10) (red). d-f) Percent by volume water refrozen for the associated source in (a-c). This value is proportional to the difference between the black and red lines above. The temperature string from which measurements were taken is labeled at the top.

**Table 1: Temperature Strings**

| String Name | Data Time Period | Time Step (hr) | Sensor | # of Sensors | Sensor Spacing (m) | Latitude | Longitude | Elevation (m) |
|---|---|---|---|---|---|---|---|---|
| T-11a | 7/5/11 – 7/15/13 | 3 | Thermistor | 32 | 0.6 | 67.195175 | -49.719515 | 848 |
| T-11b | 7/11/11 – 12/17/11 | 3 | Thermistor | 32 | 0.6 | 67.201553 | -49.289058 | 1095 |
| T-14 | 7/18/14 – 6/23/17 | 0.5 | Thermistor | 31 | < 11 m deep – 0.5
> 11 m deep – 1.0 | 67.18127 | -49.56982 | 956 |
| T-15a | 8/17/16 - 5/20/17 | 0.5 | DS18B20 | 17 | < 15 m deep – 1.0
> 15 m deep – 3.0 | 67.18211 | -49.568272 | 954 |
| T-15b | 8/17/16 - 5/20/17 | 0.5 | DS18B20 | 17 | < 15 m deep – 1.0
> 15 m deep – 3.0 | 67.182054 | -49.568059 | 954 |
| T-15c | 8/17/16 - 5/20/17 | 0.5 | DS18B20 | 17 | < 15 m deep – 1.0
> 15 m deep – 3.0 | 67.182114 | -49.568484 | 954 |
| T-16 | 8/17/16 - 7/22/17 | 0.5 | DS18B20 | 18 | 0.5 | 67.18147 | -49.57025 | 951 |

**Table 2: Constants**

| Variable | Symbol | Value | Units | Reference |
|---|---|---|---|---|
| Reference Enthalpy | $H_m$ | 0 | J kg$^{-1}$ | |
| Ice Density | $\rho_i$ | 917 | kg m$^{-3}$ | Cuffey and Paterson (2010) |
| Snow Density | $\rho_s$ | 300 | kg m$^{-3}$ | |
| Water Density | $\rho_w$ | 1000 | kg m$^{-3}$ | |
| Specific Heat Capacity | $C_p$ | 2097 | J kg$^{-1}$ K$^{-1}$ | Cuffey and Paterson (2010) |
| Latent Heat of Fusion | $L_f$ | $3.335*10^5$ | J kg$^{-1}$ | Cuffey and Paterson (2010) |
| Thermal Conductivity of Ice | $k_i$ | 2.1 | J m$^{-1}$ K$^{-1}$ s$^{-1}$ | Cuffey and Paterson (2010) |
| Thermal Conductivity of Snow | $k_s$ | 0.2 | J m$^{-1}$ K$^{-1}$ s$^{-1}$ | Calonne et al. (2011) |
| Moisture Diffusivity | $v$ | $1*10^{-4}$ | kg m$^{-1}$ s$^{-1}$ | Aschwanden et al. (2012) |